# Transformers as Decision Makers: Provable In-Context Reinforcement Learning via Supervised Pretraining

**Licong Lin**
University of California, Berkeley
liconglin@berkeley.edu

**Yu Bai**
Salesforce AI Research
yu.bai@salesforce.com

**Song Mei**
University of California, Berkeley
songmei@berkeley.edu

## ABSTRACT

Large transformer models pretrained on offline reinforcement learning datasets have demonstrated remarkable in-context reinforcement learning (ICRL) capabilities, where they can make good decisions when prompted with interaction trajectories from unseen environments. However, when and how transformers can be trained to perform ICRL have not been theoretically well-understood. In particular, it is unclear which reinforcement-learning algorithms transformers can perform in context, and how distribution mismatch in offline training data affects the learned algorithms. This paper provides a theoretical framework that analyzes supervised pretraining for ICRL. This includes two recently proposed training methods — algorithm distillation and decision-pretrained transformers. First, assuming model realizability, we prove the supervised-pretrained transformer will imitate the conditional expectation of the expert algorithm given the observed trajectory. The generalization error will scale with model capacity and a distribution divergence factor between the expert and offline algorithms. Second, we show transformers with ReLU attention can efficiently approximate near-optimal online reinforcement learning algorithms like LinUCB and Thompson sampling for stochastic linear bandits, and UCB-VI for tabular Markov decision processes. This provides the first quantitative analysis of the ICRL capabilities of transformers pretrained from offline trajectories.

## 1 INTRODUCTION

The transformer architecture (Vaswani et al., 2017) for sequence modeling has become a key weapon for modern artificial intelligence, achieving success in language (Devlin et al., 2018; Brown et al., 2020; OpenAI, 2023) and vision (Dosovitskiy et al., 2020). Motivated by these advances, the research community has actively explored how to best harness transformers for reinforcement learning (RL) (Chen et al., 2021; Janner et al., 2021; Lee et al., 2022; Reed et al., 2022; Laskin et al., 2022; Lee et al., 2023; Yang et al., 2023). While promising empirical results have been demonstrated, the theoretical understanding of transformers for RL remains limited.

This paper provides theoretical insights into in-context reinforcement learning (ICRL)—an emerging approach that utilizes sequence-to-sequence models like transformers to perform reinforcement learning in newly encountered environments. In ICRL, the model takes as input the current state and past interaction history with the environment (the *context*), and outputs an action. The key hypothesis in ICRL is that pretrained transformers can act as *RL algorithms*, progressively improving their policy based on past observations. Approaches such as Algorithm Distillation (Laskin et al., 2022) and Decision-Pretrained Transformers (Lee et al., 2023) have demonstrated early successes, finding that *supervised pretraining* can produce good ICRL performance. However, many concrete theoretical questions remain open about the ICRL capabilities of transformers, including but not limited to (1) what RL algorithms can transformers implement in-context; (2) what performance guarantees (e.g. regret bounds) can such transformers achieve when used iteratively as an online RL

algorithm; and (3) when can supervised pretraining find such a good transformer. Specifically, this paper investigates the following open question:

*How can supervised pretraining on Transformers learn in-context reinforcement learning?*

In this paper, we initiate a theoretical study of the ICRL capability of transformers under supervised pretraining to address the open questions outlined above. We show that (1) Transformers can implement prevalent RL algorithms, including LinUCB and Thompson sampling for stochastic linear bandits, and UCB-VI for tabular Markov decision processes; (2) The algorithms learned by transformers achieve near-optimal regret bounds in their respective settings; (3) Supervised pretraining find such algorithms as long as the sample size scales with the covering number of transformer class and distribution ratio between expert and offline algorithms.

**Summary of contributions and paper outline**

- We propose a general framework for supervised pretraining approaches to meta-reinforcement learning (Section 2). This framework encompasses existing methods like Algorithm Distillation (Laskin et al., 2022), where the expert and context algorithms are identical, as well as Decision-Pretrained Transformers (Lee et al., 2023), where the expert generates optimal actions for the MDP. It also includes approximate DPT variants where the expert estimates optimal actions from full interaction trajectories.

- We prove that the supervised-pretrained transformer will imitate the conditional expectation of the expert algorithm given the observed trajectory (Section 3). The generalization error scales with both model capacity and a distribution ratio measuring divergence between the expert algorithm and the algorithm that generated offline trajectories.

- We demonstrate that transformers can effectively approximate several near-optimal reinforcement learning algorithms by taking observed trajectories as context inputs (Section 4). Specifically, we show transformers can approximate LinUCB (Section 4.1) and Thompson sampling algorithms (Section 4.2) for stochastic linear bandit problems, and UCB-VI (Section 4.3) for tabular Markov decision processes. Combined with the generalization error bound from supervised pretraining and regret bounds of these RL algorithms, this provides regret bounds for supervised-pretrained transformers.

- Preliminary experiments validate that transformers can perform ICRL in our setup (Section 5).

- Technically, we prove efficient approximation of LinUCB by showing transformers can implement accelerated gradient descent for solving ridge regression (Appendix E.4), enabling fewer attention layers than the vanilla gradient descent approach in Bai et al. (2023). To enable efficient Thompson sampling implementation, we prove transformers can compute matrix square roots through the Pade decomposition (Appendix F.3). These approximation results are interesting in their own right.

**Related work** Our work is intimately related to the lines of work on meta-reinforcement learning, in-context learning, transformers for decision-making, and the approximation theory of transformers. Due to limited space, we discuss these related works in Appendix A.

## 2 FRAMEWORK FOR IN-CONTEXT REINFORCEMENT LEARNING

Let $\mathcal{M}$ be the space of decision-making environments, where each environment $\mathsf{M} \in \mathcal{M}$ shares the same number of rounds $T$ and state-action-reward spaces $\{\mathcal{S}_t, \mathcal{A}_t, \mathcal{R}_t\}_{t \in [T]}$. Each $\mathsf{M} = \{\mathbb{T}_\mathsf{M}^{t-1}, \mathbb{R}_\mathsf{M}^t\}_{t \in [T]}$ has its own transition model $\mathbb{T}_\mathsf{M}^t : \mathcal{S}_t \times \mathcal{A}_t \to \Delta(\mathcal{S}_{t+1})$ (with $\mathcal{S}_0, \mathcal{A}_0 = \{\emptyset\}$ so $\mathbb{T}_\mathsf{M}^0(\cdot) \in \Delta(\mathcal{S}_1)$ gives the initial state distribution) and reward functions $\mathbb{R}_\mathsf{M}^t : \mathcal{S}_t \times \mathcal{A}_t \to \Delta(\mathcal{R}_t)$. We equip $\mathcal{M}$ with a distribution $\Lambda \in \Delta(\mathcal{M})$, the environment prior. While this setting is general, we later give concrete examples taking $\mathcal{M}$ as $T$ rounds of bandits or $K$ episodes of $H$-step MDPs with $T = KH$.

**Distributions of offline trajectories** We denote a partial interaction trajectory, consisting of observed state-action-reward tuples, by $D_t = \{(s_1, a_1, r_1), \ldots, (s_t, a_t, r_t)\} \in \mathcal{T}_t = \prod_{s \leq t}(\mathcal{S}_s \times \mathcal{A}_s \times \mathcal{R}_s)$ and write $D = D_T$ for short. A standard algorithm Alg maps a partial trajectory $D_{t-1} \in \mathcal{T}_{t-1}$ and state $s_t \in \mathcal{S}_t$ to a distribution over the actions $\mathsf{Alg}(\cdot|D_{t-1}, s_t) \in \Delta(\mathcal{A}_t)$. Given an environment

M and algorithm Alg, the distribution over a full trajectory $D_T$ is fully specified:

$$\mathbb{P}_{\mathsf{M}}^{\mathsf{Alg}}(D_T) = \prod_{t=1}^{T} \mathbb{T}_{\mathsf{M}}^{t-1}(s_t|s_{t-1}, a_{t-1})\mathsf{Alg}(a_t|D_{t-1}, s_t)\mathbb{R}_{\mathsf{M}}^{t}(r_t|s_t, a_t).$$

In supervised pretraining, we use a *context algorithm* $\mathsf{Alg}_0$ (which we also refer to as the offline algorithm) to collect the offline trajectories $D_T$. For each trajectory $D_T$, we also assume access to expert actions $\overline{a} = (\overline{a}_t \in \mathcal{A}_t)_{t \in T} \sim \mathsf{Alg}_E(\cdot|D_T, \mathsf{M})$, sampled from an expert algorithm $\mathsf{Alg}_E$: $\mathcal{T}_T \times \mathsf{M} \to \prod_{t \in [T]} \Delta(\mathcal{A}_t)$. This expert could omnisciently observe the full trajectory $D_T$ and environment M to recommend actions. Let $\overline{D}_T = D_T \cup \{\overline{a}\}$ be the augmented trajectory. Then we have

$$\mathbb{P}_{\mathsf{M}}^{\mathsf{Alg}_0, \mathsf{Alg}_E}(\overline{D}_T) = \mathbb{P}_{\mathsf{M}}^{\mathsf{Alg}_0}(D_T) \prod_{t=1}^{T} \mathsf{Alg}_E^t(\overline{a}_t|D_T, \mathsf{M}).$$

We denote $\mathbb{P}_{\Lambda}^{\mathsf{Alg}_0, \mathsf{Alg}_E}$ as the joint distribution of $(\mathsf{M}, \overline{D}_T)$ where $\mathsf{M} \sim \Lambda$ and $\overline{D}_T \sim \mathbb{P}_{\mathsf{M}}^{\mathsf{Alg}_0, \mathsf{Alg}_E}$, and $\mathbb{P}_{\Lambda}^{\mathsf{Alg}_0}$ as the joint distribution of $(\mathsf{M}, D_T)$ where $\mathsf{M} \sim \Lambda$ and $D_T \sim \mathbb{P}_{\mathsf{M}}^{\mathsf{Alg}_0}$.

**Three special cases of expert algorithms**   We consider three special cases of the expert algorithm $\mathsf{Alg}_E$, corresponding to three supervised pretraining setups:

(a) *Algorithm distillation (Laskin et al., 2022).* The algorithm depends only on the partial trajectory $D_{t-1}$ and current state $s_t$: $\mathsf{Alg}_E^t(\cdot|D_T, \mathsf{M}) = \mathsf{Alg}_E^t(\cdot|D_{t-1}, s_t)$. For example, $\mathsf{Alg}_E$ could be a bandit algorithm like the Upper Confidence Bound (UCB).

(b) *Decision pretrained transformer (DPT) (Lee et al., 2023).* The algorithm depends on the environment M and the current state $s_t$: $\mathsf{Alg}_E^t(\cdot|D_T, \mathsf{M}) = \mathsf{Alg}_E^t(\cdot|s_t, \mathsf{M})$. For example, $\mathsf{Alg}_E$ could output the optimal action $a_t^*$ in state $s_t$ for environment M.

(c) *Approximate DPT.* The algorithm depends on the full trajectory $D_T$ but not the environment M: $\mathsf{Alg}_E^t(\cdot|D_T, \mathsf{M}) = \mathsf{Alg}_E^t(\cdot|D_T)$. For example, $\mathsf{Alg}_E$ could estimate the optimal action $\widehat{a}_t^*$ from the entire trajectory $D_T$.

For any expert algorithm $\mathsf{Alg}_E$, we define its reduced algorithm where the $t$-th step is

$$\overline{\mathsf{Alg}}_E(\cdot|D_{t-1}, s_t) := \mathbb{E}_{\Lambda}^{\mathsf{Alg}_0}[\mathsf{Alg}_E^t(\cdot|D_T, \mathsf{M})|D_{t-1}, s_t].$$

The expectation on the right is over $\mathbb{P}_{\Lambda}^{\mathsf{Alg}_0}(D_T, \mathsf{M}|D_{t-1}, s_t) = \Lambda(\mathsf{M}) \cdot \mathbb{P}_{\mathsf{M}}^{\mathsf{Alg}_0}(D_T)/\mathbb{P}_{\mathsf{M}}^{\mathsf{Alg}_0}(D_{t-1}, s_t)$. Note that the reduced expert algorithm $\overline{\mathsf{Alg}}_E$ generally depends on the context algorithm $\mathsf{Alg}_0$. However, for cases (a) and (b), $\overline{\mathsf{Alg}}_E$ is independent of the context algorithm $\mathsf{Alg}_0$. Furthermore, in case (a), we have $\overline{\mathsf{Alg}}_E^t = \mathsf{Alg}_E^t$.

**Transformer architecture**   We consider a sequence of $N$ input vectors $\{\mathbf{h}_i\}_{i=1}^{N} \subset \mathbb{R}^D$, compactly written as an input matrix $\mathbf{H} = [\mathbf{h}_1, \ldots, \mathbf{h}_N] \in \mathbb{R}^{D \times N}$, where each $\mathbf{h}_i$ is a column of $\mathbf{H}$ (also a *token*). Throughout this paper, we define $\sigma(t) := \mathrm{ReLU}(t) = \max\{t, 0\}$ as the standard relu activation function.

**Definition 1** (Masked attention layer). *A masked attention layer with $M$ heads is denoted as* $\mathrm{Attn}_{\boldsymbol{\theta}}(\cdot)$ *with parameters* $\boldsymbol{\theta} = \{(\mathbf{V}_m, \mathbf{Q}_m, \mathbf{K}_m)\}_{m \in [M]} \subset \mathbb{R}^{D \times D}$. *On any input sequence* $\mathbf{H} \in \mathbb{R}^{D \times N}$, *we have* $\overline{\mathbf{H}} = \mathrm{Attn}_{\boldsymbol{\theta}}(\mathbf{H}) = [\overline{\mathbf{h}}_1, \ldots, \overline{\mathbf{h}}_N] \in \mathbb{R}^{D \times N}$, *where*

$$\overline{\mathbf{h}}_i = [\mathrm{Attn}_{\boldsymbol{\theta}}(\mathbf{H})]_i = \mathbf{h}_i + \sum_{m=1}^{M} \frac{1}{i} \sum_{j=1}^{i} \sigma(\langle \mathbf{Q}_m \mathbf{h}_i, \mathbf{K}_m \mathbf{h}_j \rangle) \cdot \mathbf{V}_m \mathbf{h}_j \in \mathbb{R}^D.$$

We remark that the use of ReLU attention layers is for technical reasons. In practice, both ReLU attention and softmax attention layers should perform well. Indeed, several studies have shown that ReLU transformers achieve comparable performance to softmax transformers across a variety of tasks (Wortsman et al., 2023; Shen et al., 2023; Bai et al., 2023).

**Definition 2** (MLP layer). *An MLP layer with hidden dimension $D'$ is denoted as* $\mathrm{MLP}_{\boldsymbol{\theta}}(\cdot)$ *with parameters* $\boldsymbol{\theta} = (\mathbf{W}_1, \mathbf{W}_2) \in \mathbb{R}^{D' \times D} \times \mathbb{R}^{D \times D'}$. *On any input sequence* $\mathbf{H} \in \mathbb{R}^{D \times N}$, *we have* $\overline{\mathbf{H}} = \mathrm{MLP}_{\boldsymbol{\theta}}(\mathbf{H}) = [\overline{\mathbf{h}}_1, \ldots, \overline{\mathbf{h}}_N] \in \mathbb{R}^{D \times N}$, *where*

$$\overline{\mathbf{h}}_i = \mathbf{h}_i + \mathbf{W}_2 \cdot \sigma(\mathbf{W}_1 \mathbf{h}_i) \in \mathbb{R}^D.$$

We next define $L$-layer decoder-based transformers. Each layer consists of a masked attention layer (see Definition 1) followed by an MLP layer (see Definition 2) and a clip operation.

**Definition 3** (Decoder-based Transformer). *An $L$-layer decoder-based transformer, denoted as $\mathrm{TF}_{\boldsymbol{\theta}}^{\mathsf{R}}(\cdot)$, is a composition of $L$ masked attention layers, each followed by an MLP layer and a clip operation: $\mathrm{TF}_{\boldsymbol{\theta}}^{\mathsf{R}}(\mathbf{H}) = \mathbf{H}^{(L)} \in \mathbb{R}^{D \times N}$, where $\mathbf{H}^{(L)}$ is defined iteratively by taking $\mathbf{H}^{(0)} = \mathsf{clip}_{\mathsf{R}}(\mathbf{H}) \in \mathbb{R}^{D \times N}$, and for $\ell \in [L]$,*

$$\mathbf{H}^{(\ell)} = \mathsf{clip}_{\mathsf{R}}\left(\mathrm{MLP}_{\boldsymbol{\theta}_{\mathtt{mlp}}^{(\ell)}}\left(\mathrm{Attn}_{\boldsymbol{\theta}_{\mathtt{mattn}}^{(\ell)}}\left(\mathbf{H}^{(\ell-1)}\right)\right)\right) \in \mathbb{R}^{D \times N}, \quad \mathsf{clip}_{\mathsf{R}}(\mathbf{H}) = [\mathrm{proj}_{\|\mathbf{h}\|_2 \leq \mathsf{R}}(\mathbf{h}_i)]_i.$$

*Above, the parameter $\boldsymbol{\theta} = (\boldsymbol{\theta}_{\mathtt{mattn}}^{(1:L)}, \boldsymbol{\theta}_{\mathtt{mlp}}^{(1:L)})$ consists of $\boldsymbol{\theta}_{\mathtt{mattn}}^{(\ell)} = \{(\mathbf{V}_m^{(\ell)}, \mathbf{Q}_m^{(\ell)}, \mathbf{K}_m^{(\ell)})\}_{m \in [M]} \subset \mathbb{R}^{D \times D}$ and $\boldsymbol{\theta}_{\mathtt{mlp}}^{(\ell)} = (\mathbf{W}_1^{(\ell)}, \mathbf{W}_2^{(\ell)}) \in \mathbb{R}^{D' \times D} \times \mathbb{R}^{D \times D'}$. We define the parameter class of transformers as $\Theta_{D,L,M,D',B} := \{\boldsymbol{\theta} = (\boldsymbol{\theta}_{\mathtt{attn}}^{(1:L)}, \boldsymbol{\theta}_{\mathtt{mlp}}^{(1:L)}) : \|\boldsymbol{\theta}\| \leq B\}$, where the norm of a transformer $\mathrm{TF}_{\boldsymbol{\theta}}^{\mathsf{R}}$ is denoted as*

$$\|\boldsymbol{\theta}\| := \max_{\ell \in [L]}\left\{ \max_{m \in [M]}\left\{\|\mathbf{Q}_m^{(\ell)}\|_{\mathrm{op}}, \|\mathbf{K}_m^{(\ell)}\|_{\mathrm{op}}\right\} + \sum_{m=1}^{M}\|\mathbf{V}_m^{(\ell)}\|_{\mathrm{op}} + \|\mathbf{W}_1^{(\ell)}\|_{\mathrm{op}} + \|\mathbf{W}_2^{(\ell)}\|_{\mathrm{op}}\right\}. \quad (1)$$

We introduced clipped operations in transformers for technical reasons. For brevity, we will write $\mathrm{TF}_{\boldsymbol{\theta}} = \mathrm{TF}_{\boldsymbol{\theta}}^{\mathsf{R}}$ when there is no ambiguity. We will set the clipping value $\mathsf{R}$ to be sufficiently large so that the clip operator does not take effect in any of our approximation results.

**Algorithm induced by Transformers** We equip the transformer with an embedding mapping $\mathtt{h} : \cup_{t \in [T]}\mathcal{S}_t \cup \cup_{t \in [T]}(\mathcal{A}_t \times \mathcal{R}_t) \to \mathbb{R}^D$. This assigns any state $s_t \in \mathcal{S}_t$ a $D$-dimensional embedding vector $\mathtt{h}(s_t) \in \mathbb{R}^D$, and any action-reward pair $(a_t, r_t) \in \mathcal{A}_t \times \mathcal{R}_t$ a $D$-dimensional embedding $\mathtt{h}(a_t, r_t) \in \mathbb{R}^D$. The embedding function $\mathtt{h}$ should encode the time step $t$ of the state, action, and reward. With abuse of notation, we denote $\mathtt{h}(D_{t-1}, s_t) = [\mathtt{h}(s_1), \mathtt{h}(a_1, r_1), \ldots, \mathtt{h}(a_{t-1}, r_{t-1}), \mathtt{h}(s_t)]$. We define a concatenation operator $\mathtt{cat} : \mathbb{R}^{D \times *} \to \mathbb{R}^{D \times *}$ that concatenates its inputs $\mathtt{cat}(\mathbf{h}_1, \ldots, \mathbf{h}_N) = [\mathbf{h}_1, \ldots, \mathbf{h}_N]$ in most examples, but it could also insert special tokens at certain positions (in MDPs we add an additional token at the end of each episode). For a partial trajectory and current state $(D_{t-1}, s_t)$, we input $\mathbf{H} = \mathtt{cat}(\mathtt{h}(s_1), \mathtt{h}(a_1, r_1), \ldots, \mathtt{h}(a_{t-1}, r_{t-1}), \mathtt{h}(s_t)) \in \mathbb{R}^{D \times *}$ into the transformer. This produces output $\overline{\mathbf{H}} = \mathrm{TF}_{\boldsymbol{\theta}}^{\mathsf{R}}(\mathbf{H}) = [\overline{\mathbf{h}}_1, \overline{\mathbf{h}}_2 \ldots, \overline{\mathbf{h}}_{-2}, \overline{\mathbf{h}}_{-1}]$ with the same shape as $\mathbf{H}$. To extract a distribution over the action space $\mathcal{A}_t$ with $|\mathcal{A}_t| = A$ actions, we assume a fixed linear extraction mapping $\mathtt{A} \in \mathbb{R}^{A \times D}$. The induced algorithm is then defined as: $\mathsf{Alg}_{\boldsymbol{\theta}}(\cdot|D_{t-1}, s_t) = \mathrm{softmax}(\mathtt{A} \cdot \overline{\mathbf{h}}_{-1})$. The overall algorithm induced by the transformer is:

$$\mathsf{Alg}_{\boldsymbol{\theta}}(\cdot|D_{t-1}, s_t) = \mathrm{softmax}(\mathtt{A} \cdot \mathrm{TF}_{\boldsymbol{\theta}}^{\mathsf{R}}(\mathtt{cat}(\mathtt{h}(D_{t-1}, s_t)))_{-1}). \quad (2)$$

We will always choose a proper concatenation operator $\mathtt{cat}$ in examples, so that in the pretraining phase, all the algorithm outputs $\{\mathsf{Alg}_{\boldsymbol{\theta}}(\cdot|D_{t-1}, s_t)\}_{t \leq T}$ along the trajectory can be computed in a single forward propagation.

## 3 STATISTICAL ANALYSIS OF SUPERVISED PRETRAINING

In supervised pretraining, we are given $n$ i.i.d offline trajectories $\{D_T^i = (s_1^i, a_1^i, r_1^i, \ldots, s_T^i, a_T^i, r_T^i)\}_{i=1}^n \sim_{iid} \mathbb{P}_{\Lambda}^{\mathsf{Alg}_0}$ from the interaction of $\mathsf{M}^i \sim_{iid} \Lambda$ with an offline algorithm $\mathsf{Alg}_0$. Given an expert algorithm $\mathsf{Alg}_E$, we augment each trajectory $D_T^i$ by $\{\overline{a}_t^i \sim_{iid} \mathsf{Alg}_E(\cdot|D_{t-1}^i, s_t^i)\}_{t \in [T]}$. Supervised pretraining maximizes the log-likelihood over the algorithm class $\{\mathsf{Alg}_{\boldsymbol{\theta}}\}_{\boldsymbol{\theta} \in \Theta}$

$$\widehat{\boldsymbol{\theta}} = \arg\max_{\boldsymbol{\theta} \in \Theta} \frac{1}{n}\sum_{i=1}^n \sum_{t=1}^T \log \mathsf{Alg}_{\boldsymbol{\theta}}(\overline{a}_t^i|D_{t-1}^i, s_t^i). \quad (3)$$

This section discusses the statistical properties of the algorithm learned via supervised pretraining.

### 3.1 MAIN RESULT

Our main result demonstrates that the algorithm maximizing the supervised pretraining loss will imitate $\overline{\mathsf{Alg}}_E(\cdot|D_{t-1}, s_t) = \mathbb{E}_{\mathsf{M} \sim \Lambda, D_T \sim \mathsf{Alg}_0}[\mathsf{Alg}_E^t(\cdot|D_T, \mathsf{M})|D_{t-1}, s_t]$, the conditional expectation of the expert algorithm $\mathsf{Alg}_E$ given the observed trajectory. The imitation error bound will scale with the covering number of the algorithm class and a distribution ratio factor, defined as follows.

**Definition 4** (Covering number). *For a class of algorithms $\{\mathsf{Alg}_\theta, \theta \in \Theta\}$, we say $\Theta_0 \subseteq \Theta$ is an $\rho$-cover of $\Theta$, if $\Theta_0$ is a finite set such that for any $\theta \in \Theta$, there exists $\theta_0 \in \Theta_0$ such that*

$$\| \log \mathsf{Alg}_{\theta_0}(\cdot | D_{t-1}, s_t) - \log \mathsf{Alg}_\theta(\cdot | D_{t-1}, s_t) \|_\infty \leq \rho, \quad \text{for all } D_{t-1}, s_t, t \in [T].$$

*The covering number $\mathcal{N}_\Theta(\rho)$ is the minimal cardinality of $\Theta_0$ such that $\Theta_0$ is a $\rho$-cover of $\Theta$.*

**Definition 5** (Distribution ratio). *We define the distribution ratio of two algorithms $\mathsf{Alg}_1, \mathsf{Alg}_2$ by*

$$\mathcal{R}_{\mathsf{Alg}_1, \mathsf{Alg}_2} := \mathbb{E}_{\mathsf{M} \sim \Lambda, D_T \sim \mathbb{P}_{\mathsf{M}}^{\mathsf{Alg}_1}} \Big[ \prod_{s=1}^{T} \frac{\mathsf{Alg}_1(a_s | D_{s-1}, s_s)}{\mathsf{Alg}_2(a_s | D_{s-1}, s_s)} \Big] = 1 + \chi^2 \Big( \mathbb{P}_\Lambda^{\mathsf{Alg}_1}; \mathbb{P}_\Lambda^{\mathsf{Alg}_2} \Big).$$

Our main result requires the realizability assumption of algorithm class $\{\mathsf{Alg}_\theta\}_{\theta \in \Theta}$ with respect to the conditional expectation of the expert algorithm.

**Assumption A** (Approximate realizability). *There exists $\theta^* \in \Theta$ and $\varepsilon_{\mathsf{real}} > 0$ such that for all $t \in [T]$,*

$$\log \mathbb{E}_{\mathsf{M} \sim \Lambda, \overline{D}_T \sim \mathbb{P}_{\mathsf{M}}^{\mathsf{Alg}_0, \mathsf{Alg}_E}} \Big[ \frac{\overline{\mathsf{Alg}}_E(\overline{a}_t | D_{t-1}, s_t)}{\mathsf{Alg}_{\theta^*}(\overline{a}_t | D_{t-1}, s_t)} \Big] \leq \varepsilon_{\mathsf{real}}. \tag{4}$$

We aim to bound the performance gap between $\mathsf{Alg}_{\widehat{\theta}}$ and $\overline{\mathsf{Alg}}_E$ in terms of expected cumulative rewards, where the expected cumulative reward is defined as

$$\mathfrak{R}_{\Lambda, \mathsf{Alg}}(T) := \mathbb{E}_{\mathsf{M} \sim \Lambda} \big[ \mathfrak{R}_{\mathsf{M}, \mathsf{Alg}}(T) \big], \qquad \mathfrak{R}_{\mathsf{M}, \mathsf{Alg}}(T) = \mathbb{E}_{D_T \sim \mathbb{P}_{\mathsf{M}}^{\mathsf{Alg}}} \big[ \textstyle\sum_{t=1}^{T} r_t \big].$$

An intermediate step of the result is controlling the expected Hellinger distance between two algorithms, where for distributions $p, q$, we have $\mathrm{D}_{\mathrm{H}}^2(p, q) = \int ( \sqrt{p(x)} - \sqrt{q(x)} )^2 dx$.

**Theorem 6** (Performance gap between expected cumulative rewards). *Let Assumption A hold and let $\widehat{\theta}$ be a solution to Eq. (3). Take $\mathcal{R} = \mathcal{R}_{\overline{\mathsf{Alg}}_E, \mathsf{Alg}_0}$ as defined in Definition 5, and $\mathcal{N}_\Theta = \mathcal{N}_\Theta((nT)^{-2})$ as defined in Definition 4. Then for some universal constant $c > 0$, with probability at least $1 - \delta$, we have*

$$\mathbb{E}_{D_T \sim \mathbb{P}_\Lambda^{\mathsf{Alg}_E}} \Big[ \sum_{t=1}^{T} \mathrm{D}_{\mathrm{H}} \big( \mathsf{Alg}_{\widehat{\theta}}(\cdot | D_{t-1}, s_t), \overline{\mathsf{Alg}}_E(\cdot | D_{t-1}, s_t) \big) \Big] \leq cT\sqrt{\mathcal{R}} \Big( \sqrt{\frac{\log [\mathcal{N}_\Theta \cdot T/\delta]}{n}} + \sqrt{\varepsilon_{\mathsf{real}}} \Big). \tag{5}$$

*Further assume that $|r_t| \leq 1$ almost surely. Then with probability at least $1 - \delta$, the difference of the expected cumulative rewards between $\mathsf{Alg}_{\widehat{\theta}}$ and $\overline{\mathsf{Alg}}_E$ satisfies*

$$\Big| \mathfrak{R}_{\Lambda, \mathsf{Alg}_{\widehat{\theta}}}(T) - \mathfrak{R}_{\Lambda, \overline{\mathsf{Alg}}_E}(T) \Big| \leq cT^2 \sqrt{\mathcal{R}} \Big( \sqrt{\frac{\log [\mathcal{N}_\Theta \cdot T/\delta]}{n}} + \sqrt{\varepsilon_{\mathsf{real}}} \Big). \tag{6}$$

The proof of Theorem 6 is contained in Section D.1.

We remark that when the expectation on the left-hand-side of (5) is with respect to the measure $\mathbb{P}_\Lambda^{\mathsf{Alg}_0}$, standard MLE analysis will provide a bound without the distribution ratio factor $\mathcal{R} = \mathcal{R}_{\overline{\mathsf{Alg}}_E, \mathsf{Alg}_0}$ in the right-hand side. The distribution ratio factor arises from the distribution shift between trajectories generated by the expert algorithm $\mathsf{Alg}_E$ versus the context algorithm $\mathsf{Alg}_0$. In addition, it should be noted that the result in Theorem 6 holds generally provided Assumption A is satisfied, which does not require that the algorithm class is induced by transformers.

## 3.2 IMPLICATIONS IN SPECIAL CASES

**Algorithm Distillation** When we set $\mathsf{Alg}_E = \mathsf{Alg}_0$, the supervised pretraining approach corresponds to the Algorithm Distillation method introduced in Laskin et al. (2022). In this case, it suffices to set $\overline{a}^i = a^i$ for every pretraining trajectory, eliminating the need to sample additional expert actions. The conditional expectation of the expert algorithm is given by $\overline{\mathsf{Alg}}_E = \mathsf{Alg}_0$, and the distribution ratio $\mathcal{R}_{\mathsf{Alg}_E, \mathsf{Alg}_0} = 1$. Under these conditions, Theorem 6 ensures that $\mathsf{Alg}_{\widehat{\theta}}$ imitates $\mathsf{Alg}_0$ with a reward difference bounded by

$$\Big| \mathfrak{R}_{\Lambda, \mathsf{Alg}_{\widehat{\theta}}}(T) - \mathfrak{R}_{\Lambda, \mathsf{Alg}_0}(T) \Big| \leq cT^2 \Big( \sqrt{\frac{\log [\mathcal{N}_\Theta \cdot T/\delta]}{n}} + \sqrt{\varepsilon_{\mathsf{real}}} \Big).$$

If the context algorithm $\mathsf{Alg}_0$ does not perform well, we cannot expect the learned algorithm $\mathsf{Alg}_{\widehat{\theta}}$ to have good performance, regardless of the number of offline trajectories.

**Decision Pretrained Transformer**    When we set $\mathsf{Alg}_E^t = \mathsf{Alg}_E^t(s_t, \mathsf{M}) = a_t^*$ to be the optimal action at time $t$, the supervised pretraining approach corresponds to Decision-Pretrained Transformers (DPT) proposed in Lee et al. (2023). In this case, the conditional expectation of the expert algorithm $\overline{\mathsf{Alg}}_E(\cdot|D_{t-1}, s_t) = \mathbb{E}[\mathsf{Alg}_E(\cdot|s_t, \mathsf{M})|D_{t-1}, s_t] = \mathsf{Alg}_{\mathrm{TS}}(\cdot|D_{t-1}, s_t)$ is the Thompson sampling algorithm (Lee et al., 2023, Theorem 1), which samples from the posterior distribution of the optimal action $a_t^*$ given by $\mathbb{P}(a_t^*(\mathsf{M})|D_{t-1}, s_t) \propto \Lambda(\mathsf{M}) \cdot \mathbb{P}_{\mathsf{M}}^{\mathsf{Alg}_0}(D_{t-1}, s_t)$. This implies that learning from optimal actions effectively learns to imitate Thompson sampling. Furthermore, the context algorithm is not required to perform well for the learned algorithm to be consistent with Thompson sampling. However, a high-quality context algorithm $\mathsf{Alg}_0$ may help reduce the distribution ratio $\mathcal{R}$, thereby learning Thompson sampling with fewer samples.

**Approximate DPT**    In practical scenarios, the learner may not have access to the optimal action $a_t^*$ of the environment $\mathsf{M}$ during pretraining. Instead, they might rely on an estimated optimal action $\widehat{a}_t^* \sim \mathsf{Alg}_E^t(\cdot|D_T)$, derived from the entire trajectory $D_T$. We can offer a guarantee analogous to Theorem 6, provided the distribution of the estimated action closely aligns with its posterior distribution:

$$\mathbb{E}_{D_T \sim \mathbb{P}_\Lambda^{\mathsf{Alg}_0}} \mathrm{KL}(\mathsf{Alg}_E^t(\cdot|D_T) \,\|\, \mathbb{P}_{\mathrm{TS},t}(\cdot|D_T)) \le \varepsilon_{\mathsf{approx}}, \quad \forall t \in [T]. \tag{7}$$

Here, $\mathbb{P}_{\mathrm{TS},t}(\cdot|D_T)$ represents the posterior distribution of the optimal action $a_t^* = a_t^*(\mathsf{M})$ at time $t$, given the observation $D_T$, where $(\mathsf{M}, D_T) \sim \mathbb{P}_\Lambda^{\mathsf{Alg}_0}$.

**Proposition 7.** *Let Assumption A hold and let $\widehat{\theta}$ be the solution to Eq. (3). Take $\mathcal{R} = \mathcal{R}_{\mathsf{Alg}_{\mathrm{TS}}, \mathsf{Alg}_0}$ as defined in Definition 5, and $\mathcal{N}_\Theta = \mathcal{N}_\Theta((nT)^{-2})$ as defined in Definition 4. Assume that for each trajectory, an estimated optimal action is provided $\widehat{a}_t^* \sim \mathsf{Alg}_E^t(\cdot|D_T)$ at each time $t \in [T]$ satisfying Eq. (7). Assume that the rewards $|r_t| \le 1$ almost surely. Then for some universal constant $c > 0$, with probability at least $1 - \delta$, the difference of the expected cumulative rewards between $\mathsf{Alg}_{\widehat{\theta}}$ and $\mathsf{Alg}_{\mathrm{TS}}$ satisfies*

$$|\mathfrak{R}_{\Lambda, \mathsf{Alg}_{\widehat{\theta}}}(T) - \mathfrak{R}_{\Lambda, \mathsf{Alg}_{\mathrm{TS}}}(T)| \le c\sqrt{\mathcal{R}} \cdot T^2 \Big( \sqrt{\frac{\log[\mathcal{N}_\Theta \cdot T/\delta]}{n}} + \sqrt{\varepsilon_{\mathsf{real}}} + \sqrt{\varepsilon_{\mathsf{approx}}} \Big).$$

The proof of Proposition 7 is contained in Appendix D.2.

## 4    APPROXIMATION BY TRANSFORMERS

In this section, we demonstrate the capability of transformers to implement prevalent reinforcement learning algorithms that produce near-optimal regret bounds. Specifically, we illustrate the implementation of LinUCB for stochastic linear bandits in Section 4.1, Thompson sampling for stochastic linear bandits in Section 4.2, and UCB-VI for tabular Markov decision process in Section 4.3.

### 4.1    LINUCB FOR LINEAR BANDITS

A stochastic linear bandit environment is defined by $\mathsf{M} = (\mathbf{w}^*, \mathcal{E}, \mathbb{A}_1, \dots, \mathbb{A}_T)$. For each time step $t \in [T]$, the learner chooses an action $a_t \in \mathbb{R}^d$ from a set of actions $\mathbb{A}_t = \{\mathbf{a}_{t,1}, \dots, \mathbf{a}_{t,A}\}$, which consists of $A$ actions and may vary over time. Upon this action selection, the learner receives a reward $r_t = \langle a_t, \mathbf{w}^* \rangle + \varepsilon_t$. Here, $\{\varepsilon_t\} \sim_{iid} \mathcal{E}$ are zero-mean noise variables, and $\mathbf{w}^* \in \mathbb{R}^d$ represents an unknown parameter vector. Stochastic linear bandit can be cast into our general framework by setting $s_t = \mathbb{A}_t$ and adopting a deterministic transition where $s_t$ transits to $s_{t+1}$ deterministically regardless of the chosen action.

We assume the context algorithm $\mathsf{Alg}_0$ is the soft LinUCB (Chu et al., 2011). Specifically, for each time step $t \in [T]$, the learner estimates the parameter $\mathbf{w}^*$ using linear ridge regression $\mathbf{w}_{\mathrm{ridge},\lambda}^t := \arg\min_{\mathbf{w} \in \mathbb{R}^d} \sum_{j=1}^{t-1}(r_j - \langle \mathbf{a}_j, \mathbf{w} \rangle)^2 + \lambda\|\mathbf{w}\|_2^2$. Subsequently, the learner calculates the upper confidence bounds for the reward of each action as $v_{tk}^* := \langle \mathbf{a}_{t,k}, \mathbf{w}_{\mathrm{ridge},\lambda}^t \rangle + \alpha \cdot (\mathbf{a}_{t,k}^\top(\lambda\mathbf{I}_d + \sum_{j=1}^{t-1} a_j a_j^\top)^{-1}\mathbf{a}_{t,k})^{1/2}$. Finally, the learner selects an action $a_t$ according to probability $\{p_{t,j}^*\}_{j \in [A]} = \mathrm{softmax}(\{v_{tj}^*/\tau\}_{j \in [A]})$ for some sufficiently small $\tau > 0$. Note that the soft LinUCB $\mathsf{Alg}_{\mathrm{sLinUCB}(\tau)}$ recovers LinUCB as $\tau \to 0$.

We further assume the existence of constants $\sigma, b_a, B_a, B_w > 0$ such that the following conditions hold: $|\varepsilon_t| \le \sigma$, $b_a \le \|\mathbf{a}_{t,k}\|_2 \le B_a$, and $\|\mathbf{w}^*\|_2 \le B_w$ for all $t \in [T], k \in [A]$. Given these, the confidence parameter is defined as: $\alpha = \sqrt{\lambda}B_w + \sigma\sqrt{2\log(2B_a B_w T) + d\log((d\lambda + TB_a^2)/(d\lambda))} =$

$\tilde{\mathcal{O}}(\sqrt{d})$. The following result shows that the soft LinUCB algorithm can be efficiently approximated by transformers, for which the proof is contained in Appendix E.4.

**Theorem 8** (Approximating the soft LinUCB). *Consider the embedding mapping* h*, extraction mapping* A*, and concatenation operator* cat *as in E.1. For any small* $\varepsilon, \tau > 0$*, there exists a transformer* $\mathrm{TF}_{\boldsymbol{\theta}}^{\mathsf{R}}(\cdot)$ *with* $\log \mathsf{R} = \tilde{\mathcal{O}}(1)$,

$$D \leq \mathrm{O}(dA), \ L = \tilde{\mathcal{O}}(\sqrt{T}), \ M \leq 4A, \ D' = \tilde{\mathcal{O}}(d + A\sqrt{Td/(\tau\varepsilon)}), \ \|\boldsymbol{\theta}\| = \tilde{\mathcal{O}}(A + T\sqrt{d}/(\tau\varepsilon^{1/4})), \tag{8}$$

*such that taking* $\mathrm{Alg}_{\boldsymbol{\theta}}$ *as defined in Eq. (2), we have*

$$\left| \log \mathrm{Alg}_{\mathrm{sLinUCB}(\tau)}(\mathbf{a}_{t,k}|D_{t-1}, s_t) - \log \mathrm{Alg}_{\boldsymbol{\theta}}(\mathbf{a}_{t,k}|D_{t-1}, s_t) \right| \leq \varepsilon, \quad \forall t \in [T], k \in [A].$$

*Here* $\mathrm{O}(\cdot)$ *hides some absolute constant, and* $\tilde{\mathcal{O}}(\cdot)$ *additionally hides polynomial terms in* $(\sigma, b_a^{-1}, B_a, B_w, \lambda^{\pm 1})$*, and poly-logarithmic terms in* $(T, A, d, 1/\varepsilon, 1/\tau)$.

A key component in proving Theorem 8 is demonstrating that the transformer can approximate the accelerated gradient descent algorithm for solving linear ridge regression (Lemma 21), a result of independent interest. Leveraging Theorem 8, we can derive the following regret bound for the algorithm obtained via Algorithm Distillation, with the proof provided in Appendix E.5.

**Theorem 9** (Regret of LinUCB and ICRL). *Let* $\Theta = \Theta_{D,L,M,D',B}$ *be the class of transformers satisfying Eq. (8) with* $\varepsilon = 1/T^3$ *and* $\tau = 1/\log(4TAB_a(B_w + 2\alpha/\sqrt{\lambda}))/\sqrt{4T} = \tilde{\mathcal{O}}(T^{-1/2})$*, and choose the clip value* $\log \mathsf{R} = \tilde{\mathcal{O}}(1)$*. Let both the context algorithm* $\mathrm{Alg}_0$ *and the expert algorithm* $\mathrm{Alg}_E$ *coincide with the soft LinUCB algorithm* $\mathrm{Alg}_{\mathrm{sLinUCB}(\tau)}$ *with parameter* $\tau$ *during supervised pretraining. Then with probability at least* $1 - \delta$*, the learned algorithm* $\mathrm{Alg}_{\hat{\boldsymbol{\theta}}}$*, a solution to Eq. (3), entails the regret bound*

$$\mathbb{E}_{\mathsf{M} \sim \Lambda}\Big[ \sum_{t=1}^{T} \max_k \langle \mathbf{a}_{t,k}, \mathbf{w}^* \rangle - \mathfrak{R}_{\mathsf{M},\mathrm{Alg}_{\hat{\boldsymbol{\theta}}}}(T) \Big] \leq \mathcal{O}\Big( d\sqrt{T}\log(T) + T^2 \sqrt{\frac{\log(\mathcal{N}_{\Theta} \cdot T/\delta)}{n}} \Big),$$

*where* $\log \mathcal{N}_{\Theta} \leq \tilde{\mathcal{O}}(L^2 D(MD + D')\log n) \leq \tilde{\mathcal{O}}(T^{3.5}d^2 A^3 \log n)$*. Here* $\mathcal{O}$ *hides polynomial terms in* $(\sigma, b_a^{-1}, B_a, B_w, \lambda^{\pm 1})$*, and* $\tilde{\mathcal{O}}$ *additionally hides poly-logarithmic terms in* $(T, A, d, 1/\varepsilon, 1/\tau)$.

### 4.2 THOMPSON SAMPLING FOR LINEAR BANDIT

We continue to examine the stochastic linear bandit framework of Section 4.1, now assuming a Gaussian prior $\mathbf{w}^\star \sim \mathcal{N}(0, \lambda \mathbf{I}_d)$ and Gaussian noises $\{\varepsilon_t\}_{t \geq 0} \sim_{iid} \mathcal{N}(0, r)$. Additionally, we assume existence of $(b_a, B_a)$ such that $b_a \leq \|\mathbf{a}_{t,k}\|_2 \leq B_a$. In this model, Thompson sampling also utilizes linear ridge regression. Subsequently, we establish that transformers trained under the DPT methodology can learn Thompson sampling algorithms. We state the informal theorem in Theorem 10 below, where its formal statement and proof are contained in Appendix F.

**Theorem 10** (Approximating the Thompson sampling, Informal). *Consider the embedding mapping* h*, extraction mapping* A*, and concatenation operator* cat *as in E.1. Under Assumption B, C, for sufficiently small* $\varepsilon$*, there exists a transformer* $\mathrm{TF}_{\boldsymbol{\theta}}^{\mathsf{R}}(\cdot)$ *with* $\log \mathsf{R} = \tilde{\mathcal{O}}(1)$,

$$D = \tilde{\mathcal{O}}(AT^{1/4}d), \quad L = \tilde{\mathcal{O}}(\sqrt{T}), \quad M = \tilde{\mathcal{O}}(AT^{1/4}), \\ \|\boldsymbol{\theta}\| = \tilde{\mathcal{O}}(T + AT^{1/4} + \sqrt{A}), \quad D' = \tilde{\mathcal{O}}(AT^{1/4}d), \tag{9}$$

*such that taking* $\mathrm{Alg}_{\boldsymbol{\theta}}$ *as defined in Eq. (2), with probability at least* $1 - \delta_0$ *over* $(\mathsf{M}, D_T) \sim \mathbb{P}_{\Lambda}^{\mathrm{Alg}}$ *for any* Alg*, we have*

$$\log \mathrm{Alg}_{\mathrm{TS}}(\mathbf{a}_{t,k}|D_{t-1}, s_t) - \log \mathrm{Alg}_{\boldsymbol{\theta}}(\mathbf{a}_{t,k}|D_{t-1}, s_t) \leq \varepsilon, \quad \forall t \in [T], k \in [A].$$

*Here,* $\tilde{\mathcal{O}}(\cdot)$ *hides polynomial terms in* $(\mathsf{M}_0, \mathsf{C}_0, \lambda^{\pm 1}, r^{\pm 1}, b_a^{-1}, B_a)$*, and poly-logarithmic terms in* $(T, A, d, 1/\varepsilon, 1/\delta_0)$*, where* $(\mathsf{M}_0, \mathsf{C}_0)$ *are parameters in Assumption B and C.*

Central to proving Theorem 10 is establishing that the transformer can approximate matrix square roots via Pade decomposition (Appendix F.3), a result of independent interest. Theorem 10 thereby implies the subsequent regret bound for transformers trained under DPT.

**Theorem 11** (Regret of Thompson sampling and ICRL). *Follow the assumptions of Theorem 10. Let* $\Theta = \Theta_{D,L,M,D',B}$ *be the class of transformers satisfying Eq. (9) with* $\varepsilon = 1/(\mathcal{R}T^3)$, $\delta_0 = \delta/(2n)$, *and choose the clip value* $\log \mathsf{R} = \tilde{\mathcal{O}}(1)$. *Assume the trajectories are collected by some context algorithm* $\mathsf{Alg}_0$, *and we choose the expert algorithm* $\mathsf{Alg}_E(s_t, \mathsf{M}) = a_t^* = \arg\max_{\mathbf{a} \in \mathbb{A}_t} \langle \mathbf{a}, \mathbf{w}^* \rangle$ *to be the optimal action of the bandit instance* $\mathsf{M}$ *for each trajectory. Then with probability at least* $1 - \delta$, *the learned algorithm* $\mathsf{Alg}_{\widehat{\theta}}$, *a solution to Eq. (3), entails regret bound*

$$\mathbb{E}_{\mathsf{M} \sim \Lambda}\Big[\sum_{t=1}^{T} \max_k \langle \mathbf{a}_{t,k}, \mathbf{w}^* \rangle - \mathfrak{R}_{\mathsf{M}, \mathsf{Alg}_{\widehat{\theta}}}(T)\Big] \leq \mathcal{O}\bigg(d\sqrt{T} \log(Td) + \sqrt{\mathcal{R}} \cdot T^2 \sqrt{\frac{\log(\mathcal{N}_\Theta T/\delta)}{n}}\bigg),$$

*where* $\mathcal{R} = \mathcal{R}_{\mathsf{Alg}_{\mathrm{TS}}, \mathsf{Alg}_0}$, *and* $\log \mathcal{N}_\Theta \leq \tilde{\mathcal{O}}(L^2 D(MD + D') \log n) \leq \tilde{\mathcal{O}}(T^{5/4} A^2 d(\mathsf{M}_0 + A\sqrt{T}d) \log n)$. *Here* $\mathcal{O}$ *hides polynomial terms in* $(\lambda^{\pm 1}, r^{\pm 1}, b_a^{-1}, B_a)$, *and* $\tilde{\mathcal{O}}$ *additionally hides poly-logarithmic terms in* $(\mathsf{M}_0, \mathsf{C}_0, T, A, d, 1/\varepsilon, 1/\delta_0)$.

## 4.3 UCB-VI FOR TABULAR MDPS

A finite-horizon tabular MDP is specified by $\mathsf{M} = (\mathcal{S}, \mathcal{A}, H, \{P_h\}_{h \in [H]}, \{R_h\}_{h \in [H]}, \mu_1)$, with $H$ being the time horizon, $\mathcal{S}$ the state space of size $S$, $\mathcal{A}$ the action space of size $A$, and $\mu_1 \in \Delta(\mathcal{S})$ defining the initial state distribution. At each time step $h \in [H]$, $P_h : \mathcal{S} \times \mathcal{A} \to \Delta(\mathcal{S})$ denotes the state transition dynamics and $R_h : \mathcal{S} \times \mathcal{A} \to [0, 1]$ gives the reward function. A policy $\pi := \{\pi_h : (\mathcal{S} \times \mathcal{A} \times \mathbb{R})^{h-1} \times \mathcal{S} \to \Delta(\mathcal{A})\}_{h \in [H]}$ maps history and state to a distribution over actions. The value of policy $\pi$ interacting with environment $\mathsf{M}$ is defined as the expected cumulative reward $V_\mathsf{M}(\pi) = \mathbb{E}_{\mathsf{M}, \pi}[\sum_{h=1}^{H} R_h(s_h, a_h)]$. A policy $\pi^*$ is said to be optimal if $\pi^* = \arg\max_{\pi \in \Delta(\Pi)} V_\mathsf{M}(\pi)$.

We let the context algorithm $\mathsf{Alg}_0$ interact with an MDP instance $\mathsf{M}$ to generate $K$ episodes, each consisting of $H$ horizon sequences $(s_{k,h}, a_{k,h}, r_{k,h})_{k \in [K], h \in [H]}$. These can be reindexed into a single trajectory $D_T = \{(s_t, a_t, r_t)\}_{t \in [T]}$ with $t = H(k-1) + h$ and $T = KH$. The Bayes regret of any algorithm $\mathsf{Alg}$ gives $\mathbb{E}_{\mathsf{M} \sim \Lambda}[KV_\mathsf{M}(\pi^*) - \mathfrak{R}_{\mathsf{M}, \mathsf{Alg}}(T)]$.

Near minimax-optimal regret for tabular MDPs can be attained through the UCB-VI algorithm (Azar et al., 2017). We demonstrate that transformers are capable of approximating the soft UCB-VI algorithm $\mathsf{Alg}_{\mathrm{sUCBVI}(\tau)}$, a slight modification of UCB-VI formalized in Appendix G.1.

**Theorem 12** (Approximating the soft UCB-VI). *Consider the embedding mapping* h, *extraction mapping* A, *and concatenation operator* cat *as in Appendix G.1. There exists a transformer* $\mathrm{TF}_{\theta}^{\mathsf{R}}(\cdot)$ *with* $\log \mathsf{R} = \tilde{\mathcal{O}}(1)$,

$$\begin{aligned} D &= \mathrm{O}(HS^2A), \quad L = 2H + 8, \quad M = \mathrm{O}(HS^2A), \\ D' &= \mathrm{O}(K^2HS^2A), \quad \|\boldsymbol{\theta}\| \leq \tilde{\mathcal{O}}(K^2HS^2A + K^3 + 1/\tau), \end{aligned} \tag{10}$$

*such that* $\mathsf{Alg}_{\mathrm{sUCBVI}(\tau)}(a|D_{t-1}, s_t) = \mathsf{Alg}_{\boldsymbol{\theta}}(a|D_{t-1}, s_t)$ *for all* $t \in [T], a \in \mathcal{A}$. *Here* $\mathrm{O}(\cdot)$ *hides universal constants and* $\tilde{\mathcal{O}}(\cdot)$ *hides poly-logarithmic terms in* $(H, K, S, A, 1/\tau)$.

Leveraging Theorem 12, we can derive the following regret bound for the algorithm obtained via Algorithm Distillation.

**Theorem 13** (Regret of UCB-VI and ICRL). *Let* $\Theta = \Theta_{D,L,M,D',B}$ *be the class of transformers satisfying Eq. (10) with* $\tau = 1/K$, *and choose the clip value* $\log \mathsf{R} = \tilde{\mathcal{O}}(1)$. *Let both the context algorithm* $\mathsf{Alg}_0$ *and the expert algorithm* $\mathsf{Alg}_E$ *coincide with the soft UCB-VI algorithm* $\mathsf{Alg}_{\mathrm{sUCBVI}(\tau)}$ *during supervised pretraining. Then with probability at least* $1 - \delta$, *the learned algorithm* $\mathsf{Alg}_{\widehat{\theta}}$, *a solution to Eq. (3), entails regret bound*

$$\mathbb{E}_{\mathsf{M} \sim \Lambda}[KV_\mathsf{M}(\pi^*) - \mathfrak{R}_{\mathsf{M}, \mathsf{Alg}_{\widehat{\theta}}}(T)] \leq \tilde{\mathcal{O}}\bigg(H^2\sqrt{SAK} + H^3S^2A + T^2\sqrt{\frac{\log(\mathcal{N}_\Theta T/\delta)}{n}}\bigg),$$

*where* $\log \mathcal{N}_\Theta \leq \tilde{\mathcal{O}}(L^2 D(MD + D') \log n) = \tilde{\mathcal{O}}(H^4S^4A^3(K^2 + HS^2A) \log n)$, *and* $\tilde{\mathcal{O}}(\cdot)$ *hides poly-logarithmic terms in* $(H, K, S, A)$.

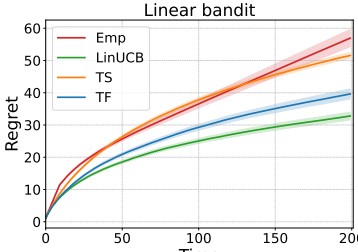 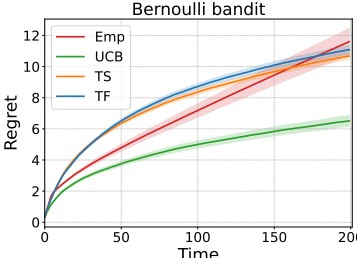

Figure 1: Regrets of transformer (TF), empirical average (Emp), Thompson sampling (TS) and LinUCB or UCB (LinUCB reduces to UCB for Bernoulli bandits). Left: linear bandit with $d = 5$, $A = 10$, $\sigma = 1.5$, $\mathsf{Alg}_0 = \mathsf{Alg}_E = \mathrm{LinUCB}$. Right: Bernoulli bandit with $d = 5$, $\mathsf{Alg}_0 = (\mathsf{Alg}_{\mathrm{unif}} + \mathsf{Alg}_{\mathrm{TS}})/2$ and $\mathsf{Alg}_E = a^*$. The simulation is repeated 500 times. Shading displays the standard deviation of the regret estimates.

## 5 EXPERIMENTS

In this section, we perform preliminary simulations to demonstrate the ICRL capabilities of transformers and validate our theoretical findings. We remark that while similar experiments have been conducted in existing works (Laskin et al., 2022; Lee et al., 2023), our setting differs in several aspects such as imitating the entire interaction trajectory in our pretrain loss (3) as opposed to on the last (query) state only as in Lee et al. (2023). The code is available at https://github.com/licong-lin/in-context-rl.

We compare pretrained transformers against empirical average, LinUCB (or UCB), and Thompson sampling. We use a GPT-2 model Garg et al. (2022); Lee et al. (2023) with $L = 8$ layers, $M = 4$ heads, and embedding dimension $D = 32$. We utilize ReLU attention layers, aligning with our theoretical construction. We pretrain the transformer with two setups: (1) Both context algorithm $\mathsf{Alg}_0$ and expert algorithm $\mathsf{Alg}_E$ use LinUCB (the Algorithm Distillation approach); (2) Context algorithms $\mathsf{Alg}_0$ mixes uniform policy and Thompson sampling, while expert $\mathsf{Alg}_E = a_t^*$ provides optimal actions (DPT). See Appendix C for further experimental details.

In the first setup, we consider stochastic linear bandits with $d = 5$ and $A = 10$. At each $t \in [200]$, the agent chooses an action $a_t$ and receives reward $r_t = \langle a_t, \mathbf{w}^* \rangle + \varepsilon_t$ where $\varepsilon_t \sim \mathcal{N}(0, 1.5^2)$. The parameter $\mathbf{w}^*$ is from $\mathrm{Unif}([0, 1]^d)$. The action set $\mathbb{A}_t = \mathbb{A}$ is fixed over time with actions i.i.d. from $\mathrm{Unif}([-1, 1]^d)$. We generate 100K trajectories using $\mathsf{Alg}_0 = \mathsf{Alg}_E = \mathrm{LinUCB}$ and train transformer $\mathrm{TF}_{\hat{\theta}}(\cdot)$ via Eq. (3). Figure 1 (left) shows regrets of the transformer (TF), empirical average (Emp), LinUCB, and Thompson sampling (TS). The transformer outperforms Thompson sampling and empirical average, and is comparable to LinUCB, agreeing with Theorem 9. The small regret gap between TF and LinUCB may stem from the limited capacity of the GPT2 model.

In the second setup, we consider multi-armed Bernoulli bandits with $d = 5$. The parameter $\mathbf{w}^*$ is from $\mathrm{Unif}([0, 1]^d)$. The fixed action set $\mathbb{A}_t = \mathbb{A}$ contains one-hot vectors $\{\mathbf{e}_i\}_{i=1}^d$ (multi-armed bandits). At each $t \in [200]$, the agent selects $a_t$ receives reward $r_t \sim \mathrm{Bern}(\langle a_t, \mathbf{w}^* \rangle)$. Let $\mathsf{Alg}_{\mathrm{unif}}$ be the uniform policy. We use $\mathsf{Alg}_{\mathrm{unif}}$ and $\mathsf{Alg}_{\mathrm{TS}}$ as context algorithms to generate 50K trajectories each. The expert is fixed as $\mathsf{Alg}_E = a^*$. We train transformer $\mathrm{TF}_{\hat{\theta}}(\cdot)$ via Eq. (3). Figure 1 (right) shows regrets for the pretrained transformer (TF), empirical average (Emp), UCB, and Thompson sampling (TS). The transformer aligns with Thompson sampling, validating Theorem 11. However, TS underperforms UCB for Bernoulli bandits, as shown.

## 6 CONCLUSIONS

This paper theoretically investigates the ICRL capability of supervised-pretrained transformers. We demonstrate how transformers can efficiently implement prevalent RL algorithms including LinUCB, Thompson sampling, and UCB-VI, achieving near-optimal regrets in respective settings. We also provide sample complexity guarantees for the supervised pretraining approach to learning these algorithms. The generalization error scales with the covering number of the transformer class as well as the distribution ratio between the expert and offline algorithms. Simulations validate our theoretical findings. Finally, we discuss the limitations of our results and provide additional discussions in Appendix A.1.

ACKNOWLEDGMENTS

The authors would like to thank Peter Bartlett for the valuable discussions, and Jonathan Lee for the valuable discussions regarding Decision-Pretrained Transformers as well as providing an early version of its implementation. This work is supported by NSF CCF-2315725, DMS-2210827, NSF Career award DMS-2339904, and an Amazon Research Award.

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

## CONTENTS

## A   RELATED WORK

**Meta-learning and meta-reinforcement learning**   In-context reinforcement learning can be cast into the framework of meta-learning and meta-reinforcement learning (Schmidhuber, 1987; 1992; Bengio et al., 1990; Naik & Mammone, 1992; Ishii et al., 2002; Schaul & Schmidhuber, 2010; Thrun & Pratt, 2012). More recently, a line of work focuses on meta-learn certain shared structures such

as the dynamics of the shared tasks (Fu et al., 2016; Nagabandi et al., 2018), a task context identifier (Rakelly et al., 2019; Humplik et al., 2019; Zintgraf et al., 2019), exploration strategies (Gupta et al., 2018), or the initialization of the network policy (Finn et al., 2017; Hochreiter et al., 2001; Nichol et al., 2018; Rothfuss et al., 2018). Theories for this last approach of model-agnostic meta-learning have been explored by Wang et al. (2020).

Our work focuses on a more agnostic approach to learning the learning algorithm itself (Wang et al., 2016; Duan et al., 2016; Dorfman et al., 2021; Mitchell et al., 2021; Li et al., 2020; Pong et al., 2022; Laskin et al., 2022; Lee et al., 2023). Among these works, Wang et al. (2016); Duan et al. (2016) focus on the online meta-RL setting with the training objective to be the total reward. Furthermore, Dorfman et al. (2021); Mitchell et al. (2021); Li et al. (2020); Pong et al. (2022) focus on offline meta-RL, but their training objectives differ from the cross entropy loss used here, requiring explicit handling of distribution shift. The supervised pretraining approach we consider is most similar to the algorithm distillation methods of Laskin et al. (2022) and the decision-pretrained transformers of Lee et al. (2023). We provide quantitative sample complexity guarantees and transformer constructions absent from previous work.

**In-context learning**  The in-context learning (ICL) capability of pretrained transformers has gained significant attention since being demonstrated on GPT-3 Brown et al. (2020). Recent work investigates why and how pretrained transformers perform ICL (Garg et al., 2022; Li et al., 2023; Von Oswald et al., 2023; Akyürek et al., 2022; Xie et al., 2021; Bai et al., 2023; Zhang et al., 2023; Ahn et al., 2023; Raventós et al., 2023). In particular, Xie et al. (2021) propose a Bayesian framework explaining how ICL works. Garg et al. (2022) show transformers can be trained from scratch to perform ICL of simple function classes. Von Oswald et al. (2023); Akyürek et al. (2022); Bai et al. (2023) demonstrate transformers can implement in-context learning algorithms via in-context gradient descent, with Bai et al. (2023) showing transformers can perform in-context algorithm selection. Zhang et al. (2023) studied training dynamics of a single attention layer for in-context learning of linear functions. Our work focuses on the related but distinct capability of in-context decision-making for pretrained transformers.

**Transformers for decision making**  Besides the ICRL approach, recent work has proposed goal-conditioned supervised learning (GCSL) for using transformers to make decisions (Chen et al., 2021; Janner et al., 2021; Lee et al., 2022; Reed et al., 2022; Brohan et al., 2022; Shafiullah et al., 2022; Yang et al., 2023). In particular, Decision Transformer (DT) (Chen et al., 2021; Janner et al., 2021) uses transformers to autoregressively model action sequences from offline data, conditioned on the achieved return. During inference, one queries the model with a desired high return. Limitations and modifications of GCSL have been studied in Yang et al. (2022); Paster et al. (2022); Strupl et al. (2022); Brandfonbrener et al. (2022). A key distinction between GCSL and ICRL is that GCSL treats the transformer as a policy, whereas ICRL treats it as an algorithm for improving the policy based on observed trajectories.

**Expressivity of transformers**  The transformer architecture, introduced by Vaswani et al. (2017), has revolutionized natural language processing and is used in most recently developed large language models like BERT and GPT (Devlin et al., 2018; Brown et al., 2020). The expressivity of transformers has been extensively studied (Yun et al., 2019; Pérez et al., 2019; Hron et al., 2020; Yao et al., 2021; Bhattamishra et al., 2020; Zhang et al., 2022; Liu et al., 2022; Wei et al., 2022; Fu et al., 2023; Bai et al., 2023; Akyürek et al., 2022; Von Oswald et al., 2023). Deep neural networks such as ResNets and transformers have been shown to efficiently approximate various algorithms, including automata (Liu et al., 2022), Turing machines (Wei et al., 2022), variational inference (Mei & Wu, 2023), and gradient descent (Bai et al., 2023; Akyürek et al., 2022; Von Oswald et al., 2023). Our work provides efficient transformer constructions that implement accelerated gradient descent and matrix square root algorithms, complementing existing expressivity results.

**Statistical theories of imitation learning**  Our generalization error analysis adapts classical analysis of maximum-likelihood estimator (Geer, 2000). The error compounding analysis for imitation learning appeared in early works (Ross et al., 2011; Ross & Bagnell, 2010). More recent theoretical analyses of imitation learning also appear in Rajaraman et al. (2020; 2021); Rashidinejad et al. (2021).

### A.1 LIMITATION AND DISCUSSION

In this section, we discuss some limitations of our work and some potential future directions.

**Distribution ratio** In Theorem 6, our regret bound of the learned algorithms $\mathsf{Alg}_{\widehat{\theta}}$ depends on the distribution ratio $\mathcal{R}_{\mathsf{Alg}_E, \mathsf{Alg}_0}$. While in cases like algorithm distillation (Laskin et al., 2022) the distribution ratio equals one since the offline algorithm matches the expert algorithm, in the worst case, the ratio can exponentially depend on $T$ or even become arbitrarily large. To control the distribution ratio in practice, one approach is to augment the offline trajectory dataset with a portion of trajectories generated by an expert algorithm or no-regret algorithms resembling the expert algorithm. On the other hand, further research could investigate structural assumptions of decision-making problems that avoid pessimistic dependence on the distribution ratio in regret bounds.

**Guarantee of pretrained transformer** Our statistical result (Theorem 6) only guarantees that the pretrained transformer learns an "algorithm" matching the expert algorithm under the pre-training distribution, even though our approximation results (Theorem 8, 10, 12) show the existence of a transformer approximating the expert algorithm over the entire input space. In our early experiments, we noticed the learned transformers do not generalize well on out-of-distribution instances, such as with shifted reward distributions or increased number of runs $T$. Similar phenomena occur in other in-context learning problems (e.g. Garg et al. (2022)). Understanding the actual algorithm implemented by the pretrained transformer through theoretical and empirical analysis is an interesting question for future work.

**Alternative pretraining methods** Our theoretical results study pretraining the transformer by maximizing the log-likelihood of i.i.d. offline trajectories as in Eq. (3). This aligns with standard supervised pretraining of large language models. However, alternative pretraining methods may also be effective. For instance, one could replace the log-probability in Eq. (3) with an $\ell_2$ loss for continuous action spaces, consider other objectives like cumulative reward (Duan et al., 2016), or explore goal-conditioned reinforcement learning (Chen et al., 2021) for in-context RL. While our work focuses on log-likelihood pretraining, theoretical investigation of alternative methods is an interesting direction for future work.

**Possibility of surpassing the expert algorithm by online training** Our work considers offline pretraining by imitating the expert algorithm (i.e., $\overline{\mathsf{Alg}}_E$), which can only learn a transformer matching the expert's performance at best. However, through online training, where the transformer interacts with the environment, the learned transformer may surpass existing experts by training to improve itself rather than imitating a specific algorithm. Investigating whether online training enables surpassing expert algorithms is an interesting direction for future work.

**Implications for practice** While the focus of our work is primarily theoretical, our results lead to several practical implications for in-context reinforcement learning. One key implication is the importance of training labels (i.e., expert actions $\bar{a}$). When the expert algorithm depends solely on past observations, we can learn $\mathsf{Alg}_E$ (see Theorem 9). In contrast, when $\mathsf{Alg}_E$ is the optimal action $a^\star$ (involving knowledge of the underlying MDP), we can learn the posterior average of this algorithm given past observations. This corresponds to the Thompson sampling algorithm, as in Decision-Pretrained Transformers (see Theorem 11).

Furthermore, as discussed previously, the distribution ratio between the offline and expert algorithms may impact the generalization of the learned algorithm. Both our theory (see Theorem 8) and simulations (see Figure 4) show that a small distribution ratio between the offline algorithm $\mathsf{Alg}_0$ and the expert algorithm $\overline{\mathsf{Alg}}_E$ is essential, otherwise the online performance of the learned algorithm may substantially degrade. This suggests that incorporating trajectories generated purely from the expert ("on-policy ICRL") into the offline dataset is advantageous, when feasible.

## B    TECHNICAL PRELIMINARIES

In this work, we will apply the following standard concentration inequality (see e.g. Lemma A.4 in Foster et al. (2021)).

**Lemma 14.** *For any sequence of random variables $(X_t)_{t \leq T}$ adapted to a filtration $\{\mathcal{F}_t\}_{t=1}^T$, we have with probability at least $1 - \delta$ that*

$$\sum_{s=1}^t X_s \leq \sum_{s=1}^t \log \mathbb{E}[\exp(X_s) \mid \mathcal{F}_{s-1}] + \log(1/\delta), \quad \textit{for all } t \in [T].$$

**Lemma 15.** *Adopt the notations in Definition 4. Then for any $\theta \in \Theta$, there exists $\theta_0 \in \Theta_0$ such that* $\|\mathsf{Alg}_{\theta_0}(\cdot|D_{t-1}, s_t) - \mathsf{Alg}_\theta(\cdot|D_{t-1}, s_t)\|_1 \leq 2\rho$.

*Proof of Lemma 15.* For any $\theta \in \Theta$, let $\theta_0 \in \Theta_0$ be such that $\|\log \mathsf{Alg}_{\theta_0}(\cdot|D_{t-1}, s_t) - \log \mathsf{Alg}_\theta(\cdot|D_{t-1}, s_t)\|_\infty \leq \rho$ for all $D_{t-1}, s_t$ and $t \in [T]$. Then

$$\|\mathsf{Alg}_{\theta_0}(\cdot|D_{t-1}, s_t) - \mathsf{Alg}_\theta(\cdot|D_{t-1}, s_t)\|_1$$

$$= \sum_{a \in \mathcal{A}_t} |\mathsf{Alg}_{\theta_0}(a|D_{t-1}, s_t) - \mathsf{Alg}_\theta(a|D_{t-1}, s_t)|$$

$$\leq \sum_{a \in \mathcal{A}_t} e^{\max\{\log \mathsf{Alg}_{\theta_0}(\cdot|D_{t-1}, s_t), \log \mathsf{Alg}_\theta(\cdot|D_{t-1}, s_t)\}}$$

$$\cdot |\log \mathsf{Alg}_{\theta_0}(\cdot|D_{t-1}, s_t) - \log \mathsf{Alg}_\theta(\cdot|D_{t-1}, s_t)|$$

$$\leq \rho \sum_{a \in \mathcal{A}_t} e^{\max\{\log \mathsf{Alg}_{\theta_0}(\cdot|D_{t-1}, s_t), \log \mathsf{Alg}_\theta(\cdot|D_{t-1}, s_t)\}}$$

$$\leq \rho \sum_{a \in \mathcal{A}_t} (\mathsf{Alg}_{\theta_0}(\cdot|D_{t-1}, s_t) + \mathsf{Alg}_\theta(\cdot|D_{t-1}, s_t)) \leq 2\rho,$$

where the second line uses a Taylor expansion of $e^x$, the fourth line uses the assumption on $\theta_0$, the last line uses $e^{\max\{x,y\}} \leq e^x + e^y$ and the fact that $\mathsf{Alg}_{\theta_0}(\cdot|D_{t-1}, s_t), \mathsf{Alg}_\theta(\cdot|D_{t-1}, s_t)$ are probability functions. □

We have the following upper bound on the covering number of the transformer class $\{\mathrm{TF}_{\boldsymbol{\theta}}^{\mathsf{R}} : \boldsymbol{\theta} \in \Theta_{D,L,M,D',B}\}$.

**Lemma 16.** *For the space of transformers* $\{\mathrm{TF}_{\boldsymbol{\theta}}^{\mathsf{R}} : \boldsymbol{\theta} \in \overline{\Theta}_{D,L,M,D',B}\}$ *with*

$$\overline{\Theta}_{D,L,M,D',B} := \left\{ \boldsymbol{\theta} = (\boldsymbol{\theta}_{\mathtt{attn}}^{[L]}, \boldsymbol{\theta}_{\mathtt{mlp}}^{[L]}) : \max_{\ell \in [L]} M^{(\ell)} \leq M, \max_{\ell \in [L]} D'^{(\ell)} \leq D', \|\boldsymbol{\theta}\| \leq B \right\},$$

*where $M^{(\ell)}, D'^{(\ell)}$ denote the number of heads and hidden neurons in the $\ell$-th layer respectively, the covering number of the set of induced algorithms $\{\mathsf{Alg}_{\boldsymbol{\theta}}, \boldsymbol{\theta} \in \overline{\Theta}_{D,L,M,D',B}\}$ (c.f. Eq. 2) satisfies*

$$\log \mathcal{N}_{\overline{\Theta}_{D,L,M,D',B}}(\rho) \leq cL^2 D(MD + D') \log \left( 2 + \frac{\max\{B, L, \mathsf{R}\}}{\rho} \right)$$

*for some universal constant $c > 0$.*

**Remark of Lemma 16.** Note that the transformer classes $\Theta_{D,L,M,D',B}, \overline{\Theta}_{D,L,M,D',B}$ have the same expressivity as one can augment any $\mathrm{TF}_{\boldsymbol{\theta}} \in \overline{\Theta}_{D,L,M,D',B}$ such that the resulting $\mathrm{TF}_{\boldsymbol{\theta},\mathrm{aug}} \in \Theta_{D,L,M,D',B}$ by adding heads or hidden neurons with fixed zero weights. Therefore, the same bound in Lemma 16 follows for $\Theta_{D,L,M,D',B}$, and throughout the paper we do not distinguish $\Theta_{D,L,M,D',B}$ and $\overline{\Theta}_{D,L,M,D',B}$ and use them interchangeably. We also use $M^{(\ell)}, D'^{(\ell)}$ to represent the number of heads and hidden neurons in the $\ell$-th layer of transformers, respectively.

*Proof of Lemma 16.* We start with introducing Proposition J.1 in Bai et al. (2023).

**Proposition 17** (Proposition J.1 in Bai et al. (2023)). *The function $\mathrm{TF}^{\mathsf{R}}$ is $(LB_H^L B_\Theta)$-Lipschitz w.r.t. $\boldsymbol{\theta} \in \Theta_{D,L,M,D',B}$ for any fixed input $\mathbf{H}$. Namely, for any $\boldsymbol{\theta}_1, \boldsymbol{\theta}_2 \in \Theta_{D,L,M,D',B}$, we have*

$$\left\| \mathrm{TF}_{\boldsymbol{\theta}_1}^{\mathsf{R}}(\mathbf{H}) - \mathrm{TF}_{\boldsymbol{\theta}_2}^{\mathsf{R}}(\mathbf{H}) \right\|_{2,\infty} \leq LB_H^L B_\Theta \|\boldsymbol{\theta}_1 - \boldsymbol{\theta}_2\|,$$

*where $\|\mathbf{A}\|_{2,\infty} := \sup_{t \in [T]} \|\mathbf{A}_{\cdot t}\|_2$ for any matrix $\mathbf{A} \in \mathbb{R}^{K \times T}$, and $B_\Theta := B\mathsf{R}(1 + B\mathsf{R}^2 + B^3\mathsf{R}^2), B_H := (1 + B^2)(1 + B^2\mathsf{R}^3)$.*

As in the Proof of Theorem 20 in Bai et al. (2023), we can verify using Example 5.8 in Wainwright (2019) that the $\delta$-covering number

$$\log N(\delta; B_{\|\cdot\|}(r), \|\cdot\|) \leq L(3MD^2 + 2DD') \log(1 + 2r/\delta), \tag{11}$$

where $B_{\|\cdot\|}(r)$ denotes any ball of radius $r$ under the norm $\|\cdot\|$. Moreover, we have the following continuity result on the log-softmax function

**Lemma 18** (Continuity of log-softmax). *For any $\mathbf{u}, \mathbf{v} \in \mathbb{R}^d$, we have*

$$\left\| \log \left( \frac{e^{\mathbf{u}}}{\|e^{\mathbf{u}}\|_1} \right) - \log \left( \frac{e^{\mathbf{v}}}{\|e^{\mathbf{v}}\|_1} \right) \right\|_\infty \leq 2 \|\mathbf{u} - \mathbf{v}\|_\infty$$

We defer the proof of Lemma 18 to the end of this section.

Note that $\mathsf{Alg}_{\boldsymbol{\theta}}(\cdot|D_{t-1}, s_t)$ corresponds to $K$ entries in one column of $\mathbf{H}^{(L)}$ applied through the softmax function. Therefore, combining Proposition 17, Lemma 18 and Eq. (11), we conclude that for any $r > 0$, there exists a subset $\Theta_0 \in \Theta_{D,L,M,D',B}$ with size $L(3MD^2 + 2DD')\log(1 + 2r/\delta)$ such that for any $\boldsymbol{\theta} \in \Theta_{D,L,M,D',B}$, there exists $\boldsymbol{\theta}_0 \in \Theta_0$ with

$$\left\| \log \mathsf{Alg}_{\boldsymbol{\theta}}(\cdot|D_{t-1}, s_t) - \log \mathsf{Alg}_{\boldsymbol{\theta}_0}(\cdot|D_{t-1}, s_t) \right\|_\infty \leq 2LB_H^L B_\Theta \delta$$

for all $D_T$. Substituting $r = B$ and letting $\delta = \rho/(2LB_H^L B_\Theta)$ yields the upper bound on $\mathcal{N}_{\Theta_{D,L,M,D',B}}(\rho)$ in Lemma 16.

*Proof of Lemma 18.* Define $\mathbf{w} := \mathbf{u} - \mathbf{v}$. Then

$$\left\| \log \left( \frac{e^{\mathbf{u}}}{\|e^{\mathbf{u}}\|_1} \right) - \log \left( \frac{e^{\mathbf{v}}}{\|e^{\mathbf{v}}\|_1} \right) \right\|_\infty$$
$$\leq \|\mathbf{u} - \mathbf{v}\|_\infty + \left| \log \|e^{\mathbf{u}}\|_1 - \log \|e^{\mathbf{v}}\|_1 \right|$$
$$= \|\mathbf{u} - \mathbf{v}\|_\infty + \int_0^1 \left\langle \frac{e^{\mathbf{v}+t\mathbf{w}}}{\|e^{\mathbf{v}+t\mathbf{w}}\|_1}, \mathbf{w} \right\rangle dt$$
$$\leq \|\mathbf{u} - \mathbf{v}\|_\infty + \int_0^1 \left\| \frac{e^{\mathbf{v}+t\mathbf{w}}}{\|e^{\mathbf{v}+t\mathbf{w}}\|_1} \right\|_1 \cdot \|\mathbf{w}\|_\infty \, dt$$
$$= 2 \|\mathbf{u} - \mathbf{v}\|_\infty \,,$$

where the third line uses the Newton-Leibniz formula. $\qquad\square$

$\square$

We present the following standard results on the convergence of GD and AGD. We refer the reader to Nesterov (2003) for the proof of these results.

**Proposition 19** (Convergence guarantee of GD and AGD). *Suppose $L(\mathbf{w})$ is a $\alpha$-strongly convex and $\beta$-smooth function on $\mathbb{R}^d$. Denote the condition number $\kappa := \beta/\alpha$ and $\mathbf{w}^* := \arg\min_{\mathbf{w}} L(\mathbf{w})$.*

*(a). The gradient descent iterates $\mathbf{w}_{\mathrm{GD}}^{t+1} := \mathbf{w}_{\mathrm{GD}}^t - \eta \nabla L(\mathbf{w}_{\mathrm{GD}}^t)$ with stepsize $\eta = 1/\beta$ and initial point $\mathbf{w}_{\mathrm{GD}}^0 = \mathbf{0}_d$ satisfies*

$$\|\mathbf{w}_{\mathrm{GD}}^t - \mathbf{w}^*\|_2^2 \leq \exp(-\frac{t}{\kappa})\|\mathbf{w}_{\mathrm{GD}}^0 - \mathbf{w}^*\|_2^2,$$

$$L(\mathbf{w}_{\mathrm{GD}}^t) - L(\mathbf{w}^*) \leq \frac{\beta}{2} \exp(-\frac{t}{\kappa})\|\mathbf{w}_{\mathrm{GD}}^0 - \mathbf{w}^*\|_2^2.$$

*(b). The accelerated gradient descent (AGD, Nesterov (2003)) iterates $\mathbf{w}_{\mathrm{AGD}}^{t+1} := \mathbf{v}_{\mathrm{GD}}^t - \frac{1}{\beta} L(\mathbf{v}_{\mathrm{AGD}}^t)$, $\mathbf{v}_{\mathrm{AGD}}^{t+1} := \mathbf{w}_{\mathrm{AGD}}^{t+1} + \frac{\sqrt{\kappa}-1}{\sqrt{\kappa}+1}(\mathbf{w}_{\mathrm{AGD}}^{t+1} - \mathbf{w}_{\mathrm{AGD}}^t)$ with $\mathbf{w}_{\mathrm{AGD}}^0 = \mathbf{v}_{\mathrm{AGD}}^0 = \mathbf{0}_d$ satisfies*

$$\|\mathbf{w}_{\mathrm{AGD}}^t - \mathbf{w}^*\|_2^2 \leq (1+\kappa) \exp(-\frac{t}{\sqrt{\kappa}})\|\mathbf{w}_{\mathrm{AGD}}^0 - \mathbf{w}^*\|_2^2,$$

$$L(\mathbf{w}_{\mathrm{AGD}}^t) - L(\mathbf{w}^*) \leq \frac{\alpha+\beta}{2} \exp(-\frac{t}{\sqrt{\kappa}})\|\mathbf{w}_{\mathrm{AGD}}^0 - \mathbf{w}^*\|_2^2.$$

## C  EXPERIMENTAL DETAILS

This section provides implementation details of our experiments and some additional simulations.

### C.1 Implementation details

**Model and embedding**   Our experiments use a GPT-2 model (Radford et al., 2019) with ReLU activation layers. The model has $L = 8$ attention layers, $M = 4$ attention heads, and embedding dimension $D = 32$. Following standard implementations in Vaswani et al. (2017), we add Layer Normalization (Ba et al., 2016) after each attention and MLP layer to facilitate optimization. We consider the embedding and extraction mappings as described in Appendix E.1, and train transformer $\mathrm{TF}_{\widehat{\theta}}(\cdot)$ via maximizing Eq. (3).

**Online algorithms**   We compare the regret of the algorithm induced by the transformer with empirical average, Thompson sampling, and LinUCB (or UCB for Bernoulli bandits).

(Emp) **Empirical average**. For time $t \le A$, the agent selects each action once. For time $t > A$, the agent computes the average of the historical rewards for each action and selects the action with the maximal averaged historical rewards.

(TS) **Thompson sampling**. For linear bandits with Gaussian noises, we consider Thompson sampling introduced in Appendix F.1 with $r = \sigma = 1.5$ and $\lambda = 1$ (note that in this case TS does not correspond to posterior sampling as we assume $\mathbf{w}^*$ follows the uniform distribution on $[0, 1]^d$). For Bernoulli bandits, we consider the standard TS sampling procedure (see, for example, Algorithm 3.2 in Russo et al. (2018)).

(LinUCB) **Linear UCB and UCB**. For linear bandits, we use LinUCB (Appendix E.2) with $\lambda = 1$ and $\alpha = 2$. For multi-armed Bernoulli bandits, LinUCB reduces to UCB, which selects $a_t = \arg\max_{a \in \mathcal{A}}\{\widehat{\mu}_{t,a} + \sqrt{1/N_t(a)}\}$, where $\mu_{t,a}$ is the average reward for action $a$ up to time $t$, and $N_t(a)$ is the number of times action $a$ was selected up to time $t$.

### C.2 Additional experiments and plots

We provide additional experiments and plots in this section. In all experiments, we choose the number of samples $n = 100\mathrm{K}$.

Additional plots of suboptimality $\langle a_t^* - a_t, \mathbf{w}^* \rangle$ over time are shown in Figure 2 for the two experiments in Section 5. In both cases, the transformer is able to imitate the expected expert policy $\overline{\mathrm{Alg}}_E$, as its suboptimality closely matches $\overline{\mathrm{Alg}}_E$ (LinUCB and TS for the left and right panel, respectively). While the empirical average (Emp) has lower suboptimality early on, its gap does not converge to zero. In contrast, both LinUCB and Thompson sampling are near-optimal up to $\tilde{\mathcal{O}}(1)$ factors in terms of their (long-term) regret.

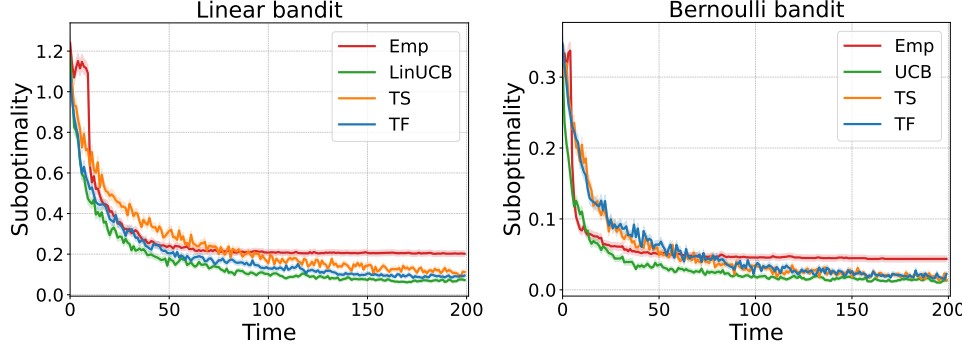

Figure 2: Suboptimalities of transformer (TF), empirical average (Emp), Thompson sampling (TS), and LinUCB (or UCB). Left: linear bandit with $d = 5$, $A = 10$, $\sigma = 1.5$, $\mathrm{Alg}_0 = \mathrm{Alg}_E = \mathrm{LinUCB}$. Right: Bernoulli bandit with $d = 5$, $\mathrm{Alg}_0 = (\mathrm{Alg}_{\mathrm{unif}} + \mathrm{Alg}_{\mathrm{TS}})/2$, and $\mathrm{Alg}_E = a_t^*$. The simulation is repeated 500 times. Shading displays the standard deviation of the sub-optimality estimates.

Additional simulations were run with $\mathrm{Alg}_0 = \mathrm{Alg}_E = \mathrm{UCB}$ for Bernoulli bandits, which has fewer actions ($A = 5$) than linear bandits ($A = 10$). Figure 3 shows the regret and suboptimality of UCB and the transformer overlap perfectly, with both algorithms exhibiting optimal behavior. This suggests the minor gaps between LinUCB and transformer in the left panel of Figure 1 and 2 are likely due to limited model capacity.

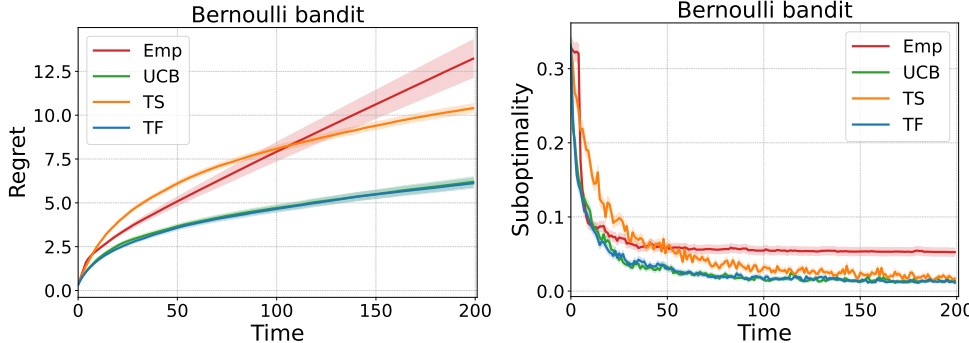

Figure 3: Regrets and suboptimalities of transformer (TF), empirical average (Emp), Thompson sampling (TS), and UCB. Settings: Bernoulli bandit with $d = 5$, and $\mathsf{Alg}_0 = \mathsf{Alg}_E = \mathrm{LinUCB}$. The simulation is repeated 500 times. Shading displays the standard deviation of the estimates.

### C.3 THE EFFECT OF DISTRIBUTION RATIO

We evaluate the effect of the distribution ratio $\mathcal{R} = \mathcal{R}_{\overline{\mathsf{Alg}_E}, \mathsf{Alg}_0}$ (Definition 5) on transformer performance. We consider the Bernoulli bandit setting from Section 5 with expert $\mathsf{Alg}_E = a^*$ giving optimal actions. The context algorithm is

$$\mathsf{Alg}_0 = \alpha \mathsf{Alg}_{\mathrm{TS}} + (1 - \alpha)\mathsf{Alg}_{\mathrm{unif}},$$

mixing uniform policy $\mathsf{Alg}_{\mathrm{unif}}$ and Thompson sampling $\mathsf{Alg}_{\mathrm{TS}}$, for $\alpha \in \{0, 0.1, 0.5, 1\}$. The case $\alpha = 0$ corresponds to the context algorithm being the i.i.d. uniform policy, and $\alpha = 1$ corresponds to the context algorithm being Thompson sampling. Note that the distribution ratio $\mathcal{R}$ may scale as $\mathcal{O}((1/\alpha) \wedge A^{\mathcal{O}(T)})$ in the worst case.

Figure 4 evaluates the learned transformers against Thompson sampling for varying context algorithms. The left plot shows cumulative regret for all algorithms. The right plot shows the regret difference between transformers and Thompson sampling. The results indicate that an increased distribution ratio impairs transformer regret, as expected. Moreover, it is observed that the transformer, even with the uniform policy (i.e., $\alpha = 0$), is capable of imitating Thompson sampling in the early stages (Time $\leq 30$), exceeding theoretical predictions. This suggests the transformer can learn Thompson sampling even when the context algorithm differs significantly from the expert algorithm.

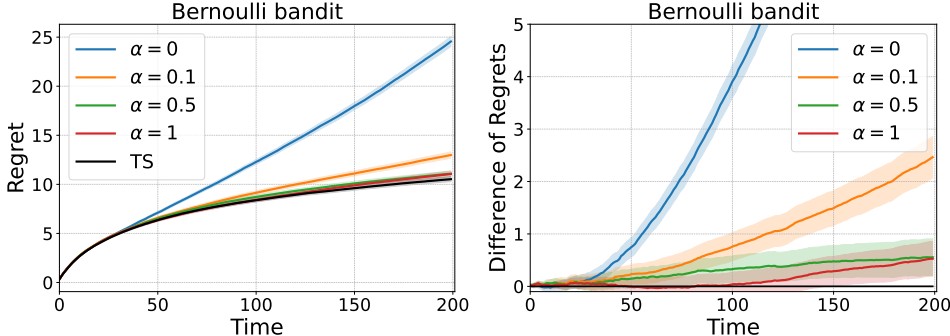

Figure 4: Regrets and difference of regrets between transformers and Thompson sampling, for different context algorithms. Settings: Bernoulli bandit with $d = 5$, $\mathsf{Alg}_E = a_t^*$ and $\mathsf{Alg}_0 = \alpha \mathsf{Alg}_{\mathrm{TS}} + (1 - \alpha)\mathsf{Alg}_{\mathrm{unif}}$ with $\alpha \in \{0, 0.1, 0.5, 1\}$. The simulation is repeated 500 times. Shading displays the standard deviation of the estimates.

## D PROOFS IN SECTION 3

In this section, $c > 0$ denotes universal constants that may differ across equations.

### D.1 PROOF OF THEOREM 6

**Proof of Eq. (5)** Note that we have

$$
\sum_{t=1}^{T} \mathbb{E}_{M\sim\Lambda, a_{1:t-1}\sim\overline{\mathsf{Alg}}_E, s_t} \sqrt{\mathrm{D}_{\mathrm{H}}^2(\overline{\mathsf{Alg}}_E(\cdot|D_{t-1}, s_t), \mathsf{Alg}_{\widehat{\theta}}(\cdot|D_{t-1}, s_t))}
$$

$$
= \sum_{t=1}^{T} \mathbb{E}_{M\sim\Lambda, a_{1:t-1}\sim\mathsf{Alg}_0, s_t} \left[ \left( \prod_{s=1}^{t-1} \frac{\overline{\mathsf{Alg}}_E(a_s|D_{s-1}, s_s)}{\mathsf{Alg}_0(a_s|D_{s-1}, s_s)} \right) \cdot \sqrt{\mathrm{D}_{\mathrm{H}}^2(\overline{\mathsf{Alg}}_E(\cdot|D_{t-1}, s_t), \mathsf{Alg}_{\widehat{\theta}}(\cdot|D_{t-1}, s_t))} \right]
$$

$$
\leq \sum_{t=1}^{T} \sqrt{ \mathbb{E}_{\Lambda,\mathsf{Alg}_0} \left( \prod_{s=1}^{t-1} \frac{\overline{\mathsf{Alg}}_E(a_s|D_{s-1}, s_s)}{\mathsf{Alg}_0(a_s|D_{s-1}, s_s)} \right)^2 \cdot \mathbb{E}_{\Lambda,\mathsf{Alg}_0} \mathrm{D}_{\mathrm{H}}^2(\overline{\mathsf{Alg}}_E(\cdot|D_{t-1}, s_t), \mathsf{Alg}_{\widehat{\theta}}(\cdot|D_{t-1}, s_t))}
$$

$$
\leq \sqrt{ \mathbb{E}_{\Lambda,\mathsf{Alg}_0} \left( \prod_{s=1}^{T} \frac{\overline{\mathsf{Alg}}_E(a_s|D_{s-1}, s_s)}{\mathsf{Alg}_0(a_s|D_{s-1}, s_s)} \right)^2 } \cdot \sum_{t=1}^{T} \sqrt{ \mathbb{E}_{\Lambda,\mathsf{Alg}_0} \mathrm{D}_{\mathrm{H}}^2(\overline{\mathsf{Alg}}_E(\cdot|D_{t-1}, s_t), \mathsf{Alg}_{\widehat{\theta}}(\cdot|D_{t-1}, s_t))},
$$

where the second line follows from a change of distribution argument, the third line follows from Cauchy-Schwartz inequality, and the fourth line uses the fact that

$$
\mathbb{E}_{x,y\sim\mathbb{P}_1\cdot\mathbb{P}_2} \left( \frac{\mathbb{Q}_1(x)\mathbb{Q}_2(y|x)}{\mathbb{P}_1(x)\mathbb{P}_2(y|x)} \right)^2 = \int \frac{\mathbb{Q}_1(x)^2\mathbb{Q}_2^2(y|x)}{\mathbb{P}_1(x)\mathbb{P}_2(y|x)} d\mu(x, y)
$$

$$
= \int \frac{\mathbb{Q}_1(x)^2}{\mathbb{P}_1(x)} \left( \int \frac{\mathbb{Q}_2^2(y|x)}{\mathbb{P}_2(y|x)} d\mu(y|x) \right) d\mu(x) \geq \int \frac{\mathbb{Q}_1(x)^2}{\mathbb{P}_1(x)} d\mu(x) = \mathbb{E}_{x\sim\mathbb{P}_1} \left( \frac{\mathbb{Q}_1(x)}{\mathbb{P}_1(x)} \right)^2,
$$

for any probability densities $\{\mathbb{Q}_i, \mathbb{P}_i\}_{i=1,2}$ with respect to some base measure $\mu$.

Continuing the calculation of the above lines of bounds, we have

$$
\sum_{t=1}^{T} \mathbb{E}_{M\sim\Lambda, a_{1:t-1}\sim\overline{\mathsf{Alg}}_E, s_t} \sqrt{\mathrm{D}_{\mathrm{H}}^2(\overline{\mathsf{Alg}}_E(\cdot|D_{t-1}, s_t), \mathsf{Alg}_{\widehat{\theta}}(\cdot|D_{t-1}, s_t))}
$$

$$
\leq \sqrt{T} \sqrt{ \mathbb{E}_{M\sim\Lambda, a_{1:T-1}\sim\mathsf{Alg}_0, s_t} \left( \prod_{s=1}^{T} \frac{\overline{\mathsf{Alg}}_E(a_s|D_{s-1}, s_s)}{\mathsf{Alg}_0(a_s|D_{s-1}, s_s)} \right)^2 }
$$

$$
\cdot \sqrt{ \sum_{t=1}^{T} \mathbb{E}_{M\sim\Lambda, a_{1:t-1}\sim\mathsf{Alg}_0, s_t} \mathrm{D}_{\mathrm{H}}^2(\overline{\mathsf{Alg}}_E(\cdot|D_{t-1}, s_t), \mathsf{Alg}_{\widehat{\theta}}(\cdot|D_{t-1}, s_t))}
$$

$$
= \sqrt{T} \sqrt{ \mathbb{E}_{M\sim\Lambda, a_{1:T-1}\sim\overline{\mathsf{Alg}}_E, s_t} \left[ \prod_{s=1}^{T} \frac{\overline{\mathsf{Alg}}_E(a_s|D_{s-1}, s_s)}{\mathsf{Alg}_0(a_s|D_{s-1}, s_s)} \right] }
$$

$$
\cdot \sqrt{ \sum_{t=1}^{T} \mathbb{E}_{M\sim\Lambda, a_{1:t-1}\sim\mathsf{Alg}_0, s_t} \mathrm{D}_{\mathrm{H}}^2(\overline{\mathsf{Alg}}_E(\cdot|D_{t-1}, s_t), \mathsf{Alg}_{\widehat{\theta}}(\cdot|D_{t-1}, s_t))}
$$

$$
\leq cT \sqrt{\mathcal{R}_{\overline{\mathsf{Alg}}_E, \mathsf{Alg}_0}} \sqrt{ \frac{\log \mathcal{N}_{\Theta}(1/(nT)^2) + \log(T/\delta)}{n} + \varepsilon_{\mathsf{real}} }
$$

$$
\leq cT \sqrt{\mathcal{R}_{\overline{\mathsf{Alg}}_E, \mathsf{Alg}_0}} \left( \sqrt{ \frac{\log[\mathcal{N}_{\Theta}(1/(nT)^2)T/\delta]}{n} } + \sqrt{\varepsilon_{\mathsf{real}}} \right),
$$

where the first inequality follows from the Cauchy-Schwartz inequality, the first equality is due to a change of distribution argument, the second inequality uses Lemma 20. This completes the proof of Eq. (5).

**Proof of Eq. (6)** For any bounded function $f$ such that $|f(D_T)| \leq F$ for some $F > 0$, we have

$$
\left| \mathbb{E}_{M\sim\Lambda, a\sim\overline{\mathsf{Alg}}_E}[f(D_T)] - \mathbb{E}_{M\sim\Lambda, a\sim\mathsf{Alg}_{\widehat{\theta}}}[f(D_T)] \right|
$$

$$= \Big| \sum_{t=1}^{T} \mathbb{E}_{M \sim \Lambda, a_{1:t} \sim \overline{\mathsf{Alg}}_E, a_{t+1:T} \sim \mathsf{Alg}_{\widehat{\theta}}}[f(D_T)] - \mathbb{E}_{M \sim \Lambda, a_{1:t-1} \sim \overline{\mathsf{Alg}}_E, a_{t:T} \sim \mathsf{Alg}_{\widehat{\theta}}}[f(D_T)] \Big|$$

$$\leq 2F \sum_{t=1}^{T} \mathbb{E}_{M \sim \Lambda, a_{1:t-1} \sim \overline{\mathsf{Alg}}_E, s_t} \mathrm{D}_{\mathrm{TV}}(\overline{\mathsf{Alg}}_E(\cdot|D_{t-1}, s_t), \mathsf{Alg}_{\widehat{\theta}}(\cdot|D_{t-1}, s_t)),$$

where the first equality uses the performance difference lemma, the last line follows from the variational representation of the total variation distance

$$\mathrm{D}_{\mathrm{TV}}(\mathbb{P}, \mathbb{Q}) = \sup_{\|f\|_{\infty}=1} \mathbb{E}_{\mathbb{P}}[f(X)]/2 - \mathbb{E}_{\mathbb{Q}}[f(X)]/2,$$

and

$$\mathrm{D}_{\mathrm{TV}}(\mathbb{P}_1(x)\mathbb{P}_2(y \mid x)\mathbb{P}_3(z \mid y), \mathbb{P}_1(x)\mathbb{P}_4(y \mid x)\mathbb{P}_3(z \mid y)) = \mathbb{E}_{x \sim \mathbb{P}_1} \mathrm{D}_{\mathrm{TV}}(\mathbb{P}_2(y \mid x), \mathbb{P}_4(y \mid x)) \tag{12}$$

for probability densities $\{\mathbb{P}_i\}_{i=1,2,3,4}$ with respect to some base measure $\mu$. Since $\mathrm{D}_{\mathrm{TV}}(\mathbb{P}, \mathbb{Q}) \leq \sqrt{\mathrm{D}_{\mathrm{H}}^2(\mathbb{P}, \mathbb{Q})}$ for any distributions $\mathbb{P}, \mathbb{Q}$, it follows from Eq. (5) that

$$\left| \mathbb{E}_{M \sim \Lambda, a \sim \overline{\mathsf{Alg}}_E}[f(D_T)] - \mathbb{E}_{M \sim \Lambda, a \sim \mathsf{Alg}_{\widehat{\theta}}}[f(D_T)] \right|$$

$$\leq cF\sqrt{\mathcal{R}_{\overline{\mathsf{Alg}}_E, \mathsf{Alg}_0}} \cdot T\Big( \sqrt{\frac{\log[\mathcal{N}_\Theta(1/(nT)^2)T/\delta]}{n}} + \sqrt{\varepsilon_{\mathsf{real}}} \Big)$$

with probability at least $1 - \delta$ for some universal constant $c > 0$. Letting $f(D_T) = \sum_{t=1}^{T} r_t$ and noting that $|f(D_T)| \leq T$ concludes the proof of Theorem 6.

## D.2 Proof of Proposition 7

By the jointly convexity of $\mathrm{KL}(\mathbb{P} \parallel \mathbb{Q})$ with respect to $(\mathbb{P}, \mathbb{Q})$ and the fact that $\mathrm{D}_{\mathrm{H}}^2(\mathbb{P}, \mathbb{Q}) \leq \mathrm{KL}(\mathbb{P} \parallel \mathbb{Q})$, we have

$$\mathbb{E}_{D_{t-1}, s_t \sim \mathbb{P}_\Lambda^{\mathsf{Alg}_0}} \mathrm{D}_{\mathrm{H}}^2(\overline{\mathsf{Alg}}_E(\cdot|D_{t-1}, s_t), \mathbb{P}_{\mathrm{TS}}(\cdot|D_{t-1}, s_t))$$

$$\leq \mathbb{E}_{D_{t-1}, s_t \sim \mathbb{P}_\Lambda^{\mathsf{Alg}_0}} \mathrm{KL}(\overline{\mathsf{Alg}}_E(\cdot|D_{t-1}, s_t) \parallel \mathbb{P}_{\mathrm{TS}}(\cdot|D_{t-1}, s_t))$$

$$\leq \mathbb{E}_{D_T \sim \mathbb{P}_\Lambda^{\mathsf{Alg}_0}} \mathrm{KL}(\widehat{a}_t^* \parallel \mathbb{P}_{\mathrm{TS},t}(\cdot|D_T)) \leq \varepsilon_{\mathsf{approx}}.$$

Therefore, applying Lemma 20 gives

$$\mathbb{E}_{M \sim \Lambda, D_T \sim \mathbb{P}_M^{\mathsf{Alg}_0}} \left[ \sum_{t=1}^{T} \mathrm{D}_{\mathrm{H}}^2\big(\mathsf{Alg}_{\widehat{\theta}}(\cdot|D_{t-1}, s_t), \mathsf{Alg}_{\mathrm{TS}}(\cdot|D_{t-1}, s_t)\big) \right]$$

$$\leq 2\mathbb{E}_{M \sim \Lambda, D_T \sim \mathbb{P}_M^{\mathsf{Alg}_0}} \left[ \sum_{t=1}^{T} \mathrm{D}_{\mathrm{H}}^2\big(\mathsf{Alg}_{\widehat{\theta}}(\cdot|D_{t-1}, s_t), \overline{\mathsf{Alg}}_E(\cdot|D_{t-1}, s_t)\big) + \mathrm{D}_{\mathrm{H}}^2(\mathsf{Alg}_E(\cdot|D_{t-1}, s_t), \mathsf{Alg}_{\mathrm{TS}}(\cdot|D_{t-1}, s_t)) \right]$$

$$\leq c\Big( \frac{T \log[\mathcal{N}_\Theta(1/(nT)^2)T/\delta]}{n} + T(\varepsilon_{\mathsf{real}} + \varepsilon_{\mathsf{approx}}) \Big)$$

with probability at least $1 - \delta$. Proposition 7 follows from similar arguments as in the proof of Theorem 6 with $\varepsilon_{\mathsf{real}}$ replaced by $\varepsilon_{\mathsf{real}} + \varepsilon_{\mathsf{approx}}$.

## D.3 An auxiliary lemma

**Lemma 20** (General guarantee for supervised pretraining). *Suppose Assumption A holds. Then the solution to Eq. (3) achieves*

$$\mathbb{E}_{D_T \sim \mathbb{P}_\Lambda^{\mathsf{Alg}_0}} \left[ \sum_{t=1}^{T} \mathrm{D}_{\mathrm{H}}^2\big(\mathsf{Alg}_{\widehat{\theta}}(\cdot|D_{t-1}, s_t), \overline{\mathsf{Alg}}_E(\cdot|D_{t-1}, s_t)\big) \right] \leq c\frac{T \log[\mathcal{N}_\Theta(1/(nT)^2)T/\delta]}{n} + T\varepsilon_{\mathsf{real}}.$$

*with probability at least $1 - \delta$ for some universal constant $c > 0$.*

*Proof of Lemma 20.*

Define

$$\mathcal{L}_{nt}(\boldsymbol{\theta}) := \sum_{i=1}^{n} \log \mathsf{Alg}_{\theta}(\overline{a}_t^i | D_{t-1}^i, s_t^i), \quad \text{and} \quad \mathcal{L}_{nt}(\text{expert}) := \sum_{i=1}^{n} \log \overline{\mathsf{Alg}}_E(\overline{a}_t^i | D_{t-1}^i, s_t^i),$$

and let $\mathcal{L}_n(\boldsymbol{\theta}) = \sum_{t=1}^{T} \mathcal{L}_{nt}(\boldsymbol{\theta})$, $\mathcal{L}_n(\text{expert}) = \sum_{t=1}^{T} \mathcal{L}_{nt}(\text{expert})$. We claim that with probability at least $1 - \delta$

$$\sum_{t=1}^{T} \mathbb{E}_{D_T} \Big[ \mathrm{D}_{\mathrm{H}}^2 (\mathsf{Alg}_{\theta}(\cdot | D_{t-1}, s_t), \overline{\mathsf{Alg}}_E(\cdot | D_{t-1}, s_t)) \Big]$$

$$\leq \frac{\mathcal{L}_n(\text{expert}) - \mathcal{L}_n(\theta)}{n} + 2\frac{T \log \mathcal{N}_{\Theta}(1/(nT)^2)}{n} + 2\frac{T \log(T/\delta)}{n} + \frac{4}{n} \qquad (13)$$

for all $\theta \in \Theta, i \in [T]$, where $D_T$ follows distribution $\mathbb{P}_{\mathsf{M}}^{\mathsf{Alg}_0}(\cdot)$, $\mathsf{M} \sim \Lambda$. For now, we assume this claim holds. Moreover, it follows from Lemma 14 and the fact $\mathcal{L}_n(\widehat{\theta}) \geq \mathcal{L}_n(\theta^*)$ that

$$\frac{\mathcal{L}_n(\text{expert}) - \mathcal{L}_n(\widehat{\theta})}{n} \leq \frac{\mathcal{L}_n(\text{expert}) - \mathcal{L}_n(\theta^*)}{n} = \sum_{t=1}^{T} \frac{\mathcal{L}_{nt}(\text{expert}) - \mathcal{L}_{nt}(\theta^*)}{n}$$

$$\leq \frac{T \log(T/\delta)}{n} + \sum_{t=1}^{T} \log \mathbb{E}_{\overline{D}_T} \left[ \frac{\overline{\mathsf{Alg}}_E(\overline{a}_t | D_{t-1}, s_t)}{\mathsf{Alg}_{\theta^*}(\overline{a}_t | D_{t-1}, s_t)} \right]$$

$$\leq \frac{T \log(T/\delta)}{n} + T\varepsilon_{\mathsf{real}} \qquad (14)$$

with probability at least $1 - \delta$.

Choosing $\theta = \widehat{\theta}$ in Eq. (13) and combining it with Eq. (14) and a union bound, we obtain

$$\sum_{t=1}^{T} \mathbb{E}_{D_T} \Big[ \mathrm{D}_{\mathrm{H}}^2 (\mathsf{Alg}_{\widehat{\theta}}(\cdot | D_{t-1}, s_t), \overline{\mathsf{Alg}}_E(\cdot | D_{t-1}, s_t)) \Big]$$

$$\leq T\varepsilon_{\mathsf{real}} + 2\Big( \frac{T \log \mathcal{N}_{\Theta}(1/(nT)^2) + 2T \log(2T/\delta) + 2}{n} \Big)$$

$$\leq T\varepsilon_{\mathsf{real}} + cT\Big( \frac{\log \mathcal{N}_{\Theta}(1/(nT)^2) + \log(T/\delta)}{n} \Big)$$

with probability at least $1 - \delta$ for some universal constant $c > 0$. This completes the proof.

**Proof of Eq. (13)** Let $\Theta_0$ be a $1/(nT)^2$-covering set of $\Theta$ with covering number $n_{\mathrm{cov}} = |\Theta_i|$. For $k \in [n_{\mathrm{cov}}], t \in [T], i \in [n]$, define

$$\ell_{kt}^i = \log \frac{\overline{\mathsf{Alg}}_E(\overline{a}_t^i | D_{t-1}^i, s_t^i)}{\mathsf{Alg}_{\theta_k}(\overline{a}_t^i | D_{t-1}^i, s_t^i)},$$

where $(D_T^i, \overline{a}^i)$ are the trajectory and expert actions collected in the $i$-th instance. Using Lemma 14 with $X_s = -\ell_{kt}^s$ and a union bound over $(k, t)$, conditioned on the trajectories $(D_T^1, \dots, D_T^n)$, we have

$$\frac{1}{2} \sum_{i=1}^{n} \ell_{kt}^i + \log(n_{\mathrm{cov}} T/\delta) \geq \sum_{i=1}^{n} -\log \mathbb{E} \Big[ \exp \Big( -\frac{\ell_{kt}^i}{2} \Big) \Big]$$

for all $k \in [n_{\mathrm{cov}}], t \in [T]$ with probability at least $1 - \delta$. Note that

$$\mathbb{E} \Big[ \exp \Big( -\frac{\ell_{kt}^i}{2} \Big) \Big| D_{t-1}^i, s_t^i \Big] = \mathbb{E}_{\mathbb{D}} \left[ \sqrt{\frac{\mathsf{Alg}_{\theta_k}(\overline{a}_t^i | D_{t-1}^i, s_t^i)}{\overline{\mathsf{Alg}}_E(\overline{a}_t^i | D_{t-1}^i, s_t^i)}} \Big| D_{t-1}^i, s_t^i \right]$$

$$= \sum_{a \in \mathcal{A}_t} \sqrt{\mathsf{Alg}_{\theta_k}(a | D_{t-1}^i, s_t^i) \overline{\mathsf{Alg}}_E(a | D_{t-1}^i, s_t^i)},$$

where the last inequality uses the assumption that the actions $\bar{a}^i$ are generated using the expert $\overline{\mathsf{Alg}}_E(\cdot|D_{t-1}^i, s_t^i)$. Therefore, for any $\theta \in \Theta$ covered by $\theta_k$, we have

$$
\begin{aligned}
&-\log \mathbb{E}\Big[\exp\Big(-\frac{\ell_{kt}^i}{2}\Big)\Big] \\
&\geq 1 - \mathbb{E}_{D^i}\Big[\sum_{a\in\mathcal{A}_t} \sqrt{\mathsf{Alg}_{\theta_k}(a|D_{t-1}^i, s_t^i)\overline{\mathsf{Alg}}_E(a|D_{t-1}^i, s_t^i)}\Big] \\
&= 1 - \mathbb{E}_{D^i}\Big[\sum_{a\in\mathcal{A}_t} \sqrt{\mathsf{Alg}_{\theta}(a|D_{t-1}^i, s_t^i)\overline{\mathsf{Alg}}_E(a|D_{t-1}^i, s_t^i)}\Big] \\
&\qquad - \mathbb{E}_{D^i}\Big[\sum_{a\in\mathcal{A}_t} \sqrt{\overline{\mathsf{Alg}}_E(a|D_{t-1}^i, s_t^i)}\Big(\sqrt{\mathsf{Alg}_{\theta_k}(a|D_{t-1}^i, s_t^i)} - \sqrt{\mathsf{Alg}_{\theta}(a|D_{t-1}^i, s_t^i)}\Big)\Big] \\
&\geq \frac{1}{2}\mathbb{E}_{D^i}\Big[\mathrm{D}_{\mathrm{H}}^2(\overline{\mathsf{Alg}}_E(\cdot|D_{t-1}^i, s_t^i), \mathsf{Alg}_{\theta}(\cdot|D_{t-1}^i, s_t^i))\Big] \\
&\qquad - \mathbb{E}_{D^i}\Big[\sum_{a\in\mathcal{A}} \Big(\sqrt{\mathsf{Alg}_{\theta}(\cdot|D_{t-1}^i, s_t^i)} - \sqrt{\mathsf{Alg}_{\theta_k}(\cdot|D_{t-1}^i, s_t^i)}\Big)^2\Big]^{1/2} \\
&\geq \frac{1}{2}\mathbb{E}_{D^i}\Big[\mathrm{D}_{\mathrm{H}}^2(\overline{\mathsf{Alg}}_E(\cdot|D_{t-1}^i, s_t^i), \mathsf{Alg}_{\theta}(\cdot|D_{t-1}^i, s_t^i))\Big] - \|\mathsf{Alg}_{\theta}(\cdot|D_{t-1}^i, s_t^i) - \mathsf{Alg}_{\theta_k}(\cdot|D_{t-1}^i, s_t^i)\|_1^{1/2} \\
&\geq \frac{1}{2}\mathbb{E}_{D^i}\Big[\mathrm{D}_{\mathrm{H}}^2(\overline{\mathsf{Alg}}_E(\cdot|D_{t-1}^i, s_t^i), \mathsf{Alg}_{\theta}(\cdot|D_{t-1}^i, s_t^i))\Big] - \frac{\sqrt{2}}{nT}
\end{aligned}
$$

for all $i \in [n], t \in [T]$, where the first inequality uses $-\log x \geq 1 - x$, the second inequality follows from Cauchy-Schwartz inequality, the third inequality uses $(\sqrt{x} - \sqrt{y})^2 \leq |x - y|$ for $x, y \geq 0$, the last inequality uses the fact that $\theta$ is covered by $\theta_k$ and Lemma 15. Since any $\theta \in \Theta$ is covered by $\theta_k$ for some $k \in [n_{\mathrm{cov}}]$, and for this $k$ summing over $t \in [T]$ gives

$$
\sum_{i=1}^{n}\sum_{t=1}^{T} \ell_{kt}^i = \mathcal{L}_n(\text{expert}) - \mathcal{L}_n(\theta_k) \leq \mathcal{L}_n(\text{expert}) - \mathcal{L}_n(\theta) + \frac{1}{nT} \leq \mathcal{L}_n(\text{expert}) - \mathcal{L}_n(\theta) + 1.
$$

Therefore, with probability at least $1 - \delta$, we have

$$
\begin{aligned}
&\frac{1}{2}\Big(\mathcal{L}_n(\text{expert}) - \mathcal{L}_n(\theta) + 1\Big) + T\log(n_{\mathrm{cov}}T/\delta) + \sqrt{2} \\
&\geq \frac{n}{2}\sum_{t=1}^{T} \mathbb{E}_{D_T}\Big[\mathrm{D}_{\mathrm{H}}^2(\mathsf{Alg}_{\theta}(\cdot|D_{t-1}, s_t), \mathsf{Alg}_{\text{expert}}(\cdot|D_{t-1}, s_t))\Big]
\end{aligned}
$$

for all $\theta \in \Theta$, where $D_T$ follows $\mathbb{P}_\Lambda^{\mathsf{Alg}_0}$. Multiplying both sides by $2/n$ and letting $n_{\mathrm{cov}} = \mathcal{N}_\Theta(1/(nT)^2)$ yields Eq. (13). $\qquad\square$

# E  SOFT LINUCB FOR LINEAR STOCHASTIC BANDIT

Throughout this section, we use $c > 0$ to denote universal constants whose values may vary from line to line. Moreover, for notational simplicity, we use $\mathrm{O}(\cdot)$ to hide universal constants, $\mathcal{O}(\cdot)$ to hide polynomial terms in the problem parameters $(\sigma, b_a^{-1}, B_a, B_w, \lambda^{\pm 1})$, and $\tilde{\mathcal{O}}(\cdot)$ to hide both poly-logarithmic terms in $(T, A, d, 1/\varepsilon, 1/\tau)$ and polynomial terms in $(\sigma, b_a^{-1}, B_a, B_w, \lambda^{\pm 1})$. We also use the bold font $\mathbf{a}_t \in \mathbb{R}^d$ to denote the selected action vector $a_t$ at time $t \in [T]$.

This section is organized as follows. Section E.1 discusses the embedding and extraction formats of transformers for the stochastic linear bandit environment. Section E.2 describes the LinUCB and the soft LinUCB algorithms. Section E.3 introduces and proves a lemma on approximating the linear ridge regression estimator, which is important for proving Theorem 8. We prove Theorem 8 in Section E.4 and prove Theorem 9 in Section E.5.

### E.1 EMBEDDING AND EXTRACTION MAPPINGS

Consider the embedding in which for each $t \in [T]$, we have two tokens $\mathbf{h}_{2t-1}, \mathbf{h}_{2t} \in \mathbb{R}^D$ such that

$$
\mathbf{h}_{2t-1} = \begin{bmatrix} \mathbf{0}_{d+1} \\ \mathbb{A}_t \\ \mathbf{0}_A \\ \mathbf{0} \\ \mathbf{pos}_{2t-1} \end{bmatrix} =: \begin{bmatrix} \mathbf{h}^a_{2t-1} \\ \mathbf{h}^b_{2t-1} \\ \mathbf{h}^c_{2t-1} \\ \mathbf{h}^d_{2t-1} \end{bmatrix}, \quad
\mathbf{h}_{2t} = \begin{bmatrix} \mathbf{a}_t \\ r_t \\ \mathbf{0}_{Ad} \\ \mathbf{0}_A \\ \mathbf{0} \\ \mathbf{pos}_{2t} \end{bmatrix} =: \begin{bmatrix} \mathbf{h}^a_{2t} \\ \mathbf{h}^b_{2t} \\ \mathbf{h}^c_{2t} \\ \mathbf{h}^d_{2t} \end{bmatrix},
$$

where $\mathbf{h}^b_{2t-1} = \mathbb{A}_t = \begin{bmatrix} \mathbf{a}^\top_{t,1} & \dots & \mathbf{a}^\top_{t,A} \end{bmatrix}^\top$ denotes the action set at time $t$, $\mathbf{h}^a_{2t} = \begin{bmatrix} \mathbf{a}^\top_t & r_t \end{bmatrix}^\top$ denotes the action and the observed reward at time $t$, $\mathbf{h}^c_{2t-1}$ is used to store the (unnormalized) policy at time $t$, $\mathbf{0}$ in $\mathbf{h}^d$ denotes an additional zero vector with dimension $\mathrm{O}(dA)$, and $\mathbf{pos}_i := (i, i^2, 1)^\top$ for $i \in [2T]$ is the positional embedding. Note that the token dimension $D = O(dA)$. In addition, we define the token matrix $\mathbf{H}_t := [\mathbf{h}_1, \dots, \mathbf{h}_{2t}] \in \mathbb{R}^{D \times 2t}$ for all $t \in [T]$.

**Offline pretraining**  During pretraining, the transformer $\mathrm{TF}_{\boldsymbol{\theta}}$ takes in $\mathbf{H}^{\mathrm{pre}}_T := \mathbf{H}_T$ as the input token matrix and generates $\mathbf{H}^{\mathrm{post}}_T := \mathrm{TF}_{\boldsymbol{\theta}}(\mathbf{H}^{\mathrm{pre}}_T)$ as the output. For each step $t \in [T]$, we define the induced policy $\mathrm{Alg}_{\boldsymbol{\theta}}(\cdot | D_{t-1}, s_t) := \frac{\exp(\mathbf{h}^{\mathrm{post},c}_{2t-1})}{\|\exp(\mathbf{h}^{\mathrm{post},c}_{2t-1})\|_1} \in \Delta^A$, whose $i$-th entry is the probability of selecting action $\mathbf{a}_{t,i}$ given $(D_{t-1}, s_t)$. We then find the transformer $\widehat{\boldsymbol{\theta}} \in \Theta$ by solving Eq. (3). Due to the decoder structure of transformer $\mathrm{TF}_{\boldsymbol{\theta}}$, the $2t - 1$-th token only has access to the first $2t - 1$ tokens. Therefore the induced policy is determined by the historical data $(D_{t-1}, s_t)$ and does not depend on future observations.

**Rollout**  At each time $t \in [T]$, given the action set $\mathbb{A}_t$ (i.e., current state $s_t$) and the previous data $D_{t-1}$, we first construct the token matrix $\mathbf{H}^{\mathrm{pre}}_{\mathrm{roll},t} = [\mathbf{H}_{t-1}, \mathbf{h}_{2t-1}] \in \mathbb{R}^{D \times (2t-1)}$. The transformer then takes $\mathbf{H}^{\mathrm{pre}}_{\mathrm{roll},t}$ as the input and generates $\mathbf{H}^{\mathrm{post}}_{\mathrm{roll},t} = [\mathbf{H}^{\mathrm{post}}_{t-1}, \mathbf{h}^{\mathrm{post}}_{2t-1}] = \mathrm{TF}_{\boldsymbol{\theta}}(\mathbf{H}^{\mathrm{pre}}_{\mathrm{roll},t})$. Next, the agent selects an action $\mathbf{a}_t \in \mathbb{A}_t$ according to the induced policy $\mathrm{Alg}_{\boldsymbol{\theta}}(\cdot | D_{t-1}, s_t) := \frac{\exp(\mathbf{h}^{\mathrm{post},c}_{2t-1})}{\|\exp(\mathbf{h}^{\mathrm{post},c}_{2t-1})\|_1} \in \Delta^A$ and observes the reward $r_t$.

**Embedding and extraction mappings**  To integrate the above construction into the framework described in Section 2, we have the embedding vectors $\mathrm{h}(s_t) := \mathbf{h}_{2t-1}, \mathrm{h}(a_t, r_t) := \mathbf{h}_{2t}$, the concatenation operator $\mathrm{cat}(\mathbf{h}_1, \dots, \mathbf{h}_N) := [\mathbf{h}_1, \dots, \mathbf{h}_N]$, the input token matrix

$$
\mathbf{H} = \mathbf{H}^{\mathrm{pre}}_{\mathrm{roll},t} := \mathrm{cat}(\mathrm{h}(s_1), \mathrm{h}(a_1, r_1), \dots, \mathrm{h}(a_{t-1}, r_{t-1}), \mathrm{h}(s_t)) \in \mathbb{R}^{D \times (2t-1)},
$$

the output token matrix $\overline{\mathbf{H}} = \mathbf{H}^{\mathrm{post}}_{\mathrm{roll},t}$, and the linear extraction map $\mathtt{A}$ satisfies $\mathtt{A} \cdot \overline{\mathbf{h}}_{-1} = \mathtt{A} \cdot \overline{\mathbf{h}}^{\mathrm{post}}_{2t-1} = \mathbf{h}^{\mathrm{post},c}_{2t-1}$.

### E.2 LINUCB AND SOFT LINUCB

Let $T$ be the total time and $\lambda, \alpha > 0$ be some prespecified values. At each time $t \in [T]$, LinUCB consists of the following steps:

1. Computes the ridge estimator $\mathbf{w}^t_{\mathrm{ridge},\lambda} = \arg\min_{\mathbf{w} \in \mathbb{R}^d} \frac{1}{2t} \sum_{j=1}^{t-1} (r_j - \langle \mathbf{a}_j, \mathbf{w} \rangle)^2 + \frac{\lambda}{2t} \|\mathbf{w}\|_2^2$.

2. For each action $k \in [A]$, computes $v^*_{tk} := \left\langle \mathbf{a}_{t,k}, \mathbf{w}^t_{\mathrm{ridge},\lambda} \right\rangle + \alpha \sqrt{\mathbf{a}^\top_{t,k} \mathbf{A}^{-1}_t \mathbf{a}_{t,k}}$, where $\mathbf{A}_t = \lambda \mathbf{I}_d + \sum_{j=1}^{t-1} \mathbf{a}_j \mathbf{a}^\top_j$.

3. Selects the action $\mathbf{a}_{t,j}$ with $j := \arg\max_{k \in [A]} v^*_{tk}$.

Unless stated otherwise, in step 2 above we choose $\alpha = \alpha(\delta)$ with $\delta = 1/(2 B_a B_w T)$ and

$$
\alpha(\delta) := \sqrt{\lambda} B_w + \sigma \sqrt{2 \log(1/\delta) + d \log((d\lambda + T B_a^2)/(d\lambda))} = \mathcal{O}(\sqrt{d \log T}) = \tilde{\mathcal{O}}(\sqrt{d}).
$$

In this work, to facilitate the analysis of supervised pretraining, we consider soft LinUCB (denoted by $\mathrm{sLinUCB}(\tau)$), which replaces step 3 in LinUCB with

3' Selects the action $\mathbf{a}_{t,j}$ with probability $\frac{\exp(v^*_{tj}/\tau)}{\|\exp(v^*_{tj}/\tau)\|_1}$ for $j \in [A]$.

Note that soft LinUCB recovers the standard LinUCB as $\tau \to 0$.

### E.3   APPROXIMATION OF THE RIDGE ESTIMATOR

In this section, we present a lemma on how transformers can approximately implement the ridge regression estimator in-context.

Throughout the proof, for $t \in [2T]$, we let $\mathbf{h}_t^{(L)}$ denote the $i$-th token in the output token matrix obtained after passing through an $L$-layer transformer. We also define $\text{read}_{\mathbf{w}_{\text{ridge}}} : \mathbb{R}^D \mapsto \mathbb{R}^d$ be the operator that gives the values of $d$ coordinates in the token vector that are used to store the estimation of the ridge estimate.

**Lemma 21** (Approximation of the ridge estimator). *For any small $\varepsilon > 0$, there exists an attention-only (i.e., no MLP layers) transformer $\text{TF}_{\boldsymbol{\theta}}(\cdot)$ with*

$$L = \left\lceil \frac{4T(B_a^2 + \lambda)}{\lambda} \log(TB_a(B_aB_w + \sigma)/(\lambda\varepsilon)) \right\rceil = \tilde{\mathcal{O}}(T), \quad \max_{\ell \in [L]} M^{(l)} \leq 3, \quad \|\boldsymbol{\theta}\| \leq \sqrt{2} + \frac{\lambda + 2}{B_a^2 + \lambda} = \mathcal{O}(1)$$

*such that $\|\text{read}_{\mathbf{w}_{\text{ridge}}}(\mathbf{h}_{2t-1}^{(L)}) - \mathbf{w}_{\text{ridge},\lambda}^t\|_2 \leq \varepsilon$ for all $t \in [T]$.*

*Moreover, there exists a transformer $\text{TF}_{\boldsymbol{\theta}}(\cdot)$ with*

$$L = \left\lceil 2\sqrt{2T} \sqrt{\frac{B_a^2 + \lambda}{\lambda}} \log \left( \frac{(2T(B_a^2 + \lambda) + \lambda)TB_a(B_aB_w + \sigma)}{\lambda^2 \varepsilon} \right) \right\rceil = \tilde{\mathcal{O}}(\sqrt{T}), \quad \max_{\ell \in [L]} M^{(l)} \leq 4,$$

$$\max_{\ell \in [L]} D'^{(\ell)} \leq 4d, \quad \|\boldsymbol{\theta}\| \leq 10 + \frac{\lambda + 2}{B_a^2 + \lambda} = \mathcal{O}(1)$$

*such that $\|\text{read}_{\mathbf{w}_{\text{ridge}}}(h_{2t-1}^{(L)}) - \mathbf{w}_{\text{ridge},\lambda}^t\|_2 \leq \varepsilon$ for all $t \in [T]$.*

Results similar to Lemma 21 have been shown in Bai et al. (2023) under a different scenario. However, we remark that the second part of Lemma 21 has a weaker requirement on the number of layers as we prove that transformers can implement accelerated gradient descent (AGD, Nesterov (2003)) in-context.

*Proof of Lemma 21.* Note that $\lambda \mathbf{I}_d \preceq \mathbf{A}_t \preceq (TB_a^2 + \lambda)\mathbf{I}_d$. Therefore the optimization problem

$$\mathbf{w}_{\text{ridge},\lambda}^t = \arg\min_{\mathbf{w} \in \mathbb{R}^d} L(\mathbf{w}) := \arg\min_{\mathbf{w} \in \mathbb{R}^d} \frac{1}{2(2t-1)} \sum_{j=1}^{t-1} (r_j - \langle \mathbf{a}_j, \mathbf{w} \rangle)^2 + \frac{\lambda}{2(2t-1)} \|\mathbf{w}\|_2^2$$

is $\lambda/(2t-1)$-strongly convex and $(B_a^2 + \lambda)$-smooth and the condition number $\kappa \leq 2T(B_a^2 + \lambda)/\lambda$. Moreover, by the definition of $\mathbf{w}_{\text{ridge},\lambda}^t$ we have

$$\|\mathbf{w}_{\text{ridge},\lambda}^t\|_2 = \|(\lambda\mathbf{I}_d + \sum_{j=1}^{t-1} \mathbf{a}_j \mathbf{a}_j^\top)^{-1}(\sum_{j=1}^{t-1} \mathbf{a}_j r_j)\|_2 \leq \|(\lambda\mathbf{I}_d + \sum_{j=1}^{t-1} \mathbf{a}_j \mathbf{a}_j^\top)^{-1}\|_2 \cdot \|\sum_{j=1}^{t-1} \mathbf{a}_j r_j\|_2$$

$$\leq \frac{TB_a(B_aB_w + \sigma)}{\lambda}$$

for all $t \in [T]$.

**Proof of part 1**   By Proposition 19, we see that $L = \lceil 4T(B_a^2 + \lambda) \log(TB_a(B_aB_w + \sigma)/(\lambda\varepsilon))/\lambda \rceil$ steps of gradient descent with stepsize $\eta = 1/(B_a^2 + \lambda)$ starting from $\mathbf{w}_{\text{GD}}^0 = \mathbf{0}_d$ finds $\mathbf{w}_{\text{GD}}^L$ such that $\|\mathbf{w}_{\text{GD}}^L - \mathbf{w}_{\text{ridge},\lambda}^t\|_2 \leq \varepsilon$.

Now we prove that one attention-only layer can implement one step of gradient descent

$$\mathbf{w}_{\text{GD}}^{\ell+1} := \mathbf{w}_{\text{GD}}^\ell - \frac{\eta}{2t-1} \sum_{j=1}^{t-1} (\langle \mathbf{a}_j, \mathbf{w}_{\text{GD}}^\ell \rangle - r_j)\mathbf{a}_j - \frac{\eta\lambda}{2t-1} \mathbf{w}_{\text{GD}}^\ell.$$

We encode the algorithm using the last token (i.e., the $2t - 1$-th token). Denote the first $d$ entries of $\mathbf{h}_{2t-1}^d$ by $\hat{\mathbf{w}}$ and define $\text{read}_{\mathbf{w}_{\text{ridge}}}(\mathbf{h}_{2t-1}) = \hat{\mathbf{w}}$. Starting from $\hat{\mathbf{w}}^0 = \mathbf{0}_d$, for each layer $\ell \in [L]$, we

let the number of heads $M^{(\ell)} = 3$ and choose $\mathbf{Q}_{1,2,3}^{(\ell)}, \mathbf{K}_{1,2,3}^{(\ell)}, \mathbf{V}_{1,2,3}^{(\ell)}$ such that for even tokens $\mathbf{h}_{2j}$ with $j \leq t-1$ and odd tokens $\mathbf{h}_{2j-1}$ with $j \leq t$

$$\mathbf{Q}_1^{(\ell)}\mathbf{h}_{2t-1}^{(\ell-1)} = \begin{bmatrix} \widehat{\mathbf{w}}^{\ell-1} \\ 1 \end{bmatrix}, \quad \mathbf{K}_1^{(\ell)}\mathbf{h}_{2j}^{(\ell-1)} = \begin{bmatrix} \mathbf{a}_j \\ -r_j \end{bmatrix}, \quad \mathbf{V}_1^{(\ell)}\mathbf{h}_{2j}^{(\ell-1)} = -\eta \begin{bmatrix} \mathbf{0} \\ \mathbf{a}_j \\ \mathbf{0} \end{bmatrix}, \quad \mathbf{K}_1^{(\ell)}\mathbf{h}_{2j-1}^{(\ell-1)} = \mathbf{0}, \quad \mathbf{V}_1^{(\ell)}\mathbf{h}_{2j-1}^{(\ell-1)} = \mathbf{0}$$

$$\mathbf{Q}_2^{(\ell)} = -\mathbf{Q}_1^{(\ell)}, \quad \mathbf{K}_2^{(\ell)} = \mathbf{K}_1^{(\ell)}, \quad \mathbf{V}_2^{(\ell)} = -\mathbf{V}_1^{(\ell)},$$

$$\mathbf{Q}_3^{(\ell)}\mathbf{h}_{2t-1}^{(\ell-1)} = \begin{bmatrix} 1 \\ -(2t-1) \\ 1 \end{bmatrix}, \quad \mathbf{K}_3^{(\ell)}\mathbf{h}_{2j}^{(\ell-1)} = \begin{bmatrix} 1 \\ 1 \\ 2j \end{bmatrix}, \quad \mathbf{K}_3^{(\ell)}\mathbf{h}_{2j-1}^{(\ell-1)} = \begin{bmatrix} 1 \\ 1 \\ 2j-1 \end{bmatrix}, \quad \mathbf{V}_3^{(\ell)}\mathbf{h}_{2t-1}^{(\ell-1)} = -\eta\lambda \begin{bmatrix} \mathbf{0} \\ \widehat{\mathbf{w}}^{\ell-1} \\ \mathbf{0} \end{bmatrix}.$$

Summing up the three heads and noting that $t = \sigma(t) - \sigma(-t)$, we see that the $\widehat{\mathbf{w}}$ part of $\mathbf{h}_{2t-1}$ (i.e., $\mathrm{read}_{\mathbf{w}_{\mathrm{ridge}}}(\mathbf{h}_{2t-1})$) has the update

$$\widehat{\mathbf{w}}^l = \widehat{\mathbf{w}}^{l-1} - \frac{\eta}{2t-1} \sum_{j=1}^{t}[\sigma(\langle \mathbf{a}_j, \widehat{\mathbf{w}}^{l-1}\rangle - r_j) - \sigma(r_j - \langle \mathbf{a}_j, \widehat{\mathbf{w}}^{l-1}\rangle)]\mathbf{a}_j$$

$$- \frac{\eta\lambda}{2t-1}\Big[\sum_{j=1}^{t-1}(\sigma(1+2j-2t)\mathbf{V}_3^{(\ell)}\mathbf{h}_{2j-1}^{(\ell-1)} + \sigma(1+2j-2t+1)\mathbf{V}_3^{(\ell)}\mathbf{h}_{2j}^{(\ell-1)}) + \mathbf{V}_3^{(\ell)}\mathbf{h}_{2t-1}^{(\ell-1)}\Big]$$

$$= \widehat{\mathbf{w}}^{l-1} - \frac{\eta}{2t-1}\sum_{j=1}^{t}[\langle \mathbf{a}_j, \widehat{\mathbf{w}}^{l-1}\rangle - r_j]\mathbf{a}_j - \frac{\eta\lambda}{2t-1}\mathbf{V}_3^{(\ell)}\mathbf{h}_{2t-1}^{(\ell-1)}$$

$$= \widehat{\mathbf{w}}^{l-1} - \frac{\eta}{2t-1}\sum_{j=1}^{t-1}[\langle \mathbf{a}_j, \widehat{\mathbf{w}}^{l-1}\rangle - r_j]\mathbf{a}_j - \frac{\eta\lambda}{2t-1}\widehat{\mathbf{w}}^{l-1},$$

which is one step of gradient descent with stepsize $\eta$. Moreover, it is easy to see that one can choose the marices such that $\max_{m\in[3]}\|\mathbf{Q}_m^{(\ell)}\|_{\mathrm{op}} = \max_{m\in[3]}\|\mathbf{K}_m^{(\ell)}\|_{\mathrm{op}} = \sqrt{2}$ and $\|\mathbf{V}_1^{(\ell)}\|_{\mathrm{op}} = \|\mathbf{V}_2^{(\ell)}\|_{\mathrm{op}} = \eta, \|\mathbf{V}_3^{(\ell)}\|_{\mathrm{op}} = \lambda\eta$. Therefore the norm of the transformer $\|\boldsymbol{\theta}\| \leq \sqrt{2} + (\lambda+2)/(B_a^2 + \lambda)$.

**Proof of part 2** Similarly, Proposition 19 shows $L = \lceil 2\sqrt{2T(B_a^2 + \lambda)/\lambda}\log((1 + \kappa)TB_a(B_aB_w + \sigma)/(\lambda\varepsilon))\rceil$ steps of accelerated gradient descent gives $\|\mathbf{w}_{\mathrm{AGD}}^L - \mathbf{w}_{\mathrm{ridge},\lambda}^t\|_2 \leq \varepsilon$.

Again, we encode the algorithm using the last token (i.e., the $2t-1$-th token). Denote the first $d, d+1 \sim 2d, 2d+1 \sim 3d$ entries of $\mathbf{h}_{2t-1}^d$ by $\widehat{\mathbf{w}}_a, \widehat{\mathbf{w}}_b, \widehat{\mathbf{v}}$ respectively. Starting from $\widehat{\mathbf{w}}_a^0 = \widehat{\mathbf{w}}_b^0 = \widehat{\mathbf{v}}^0 = \mathbf{0}_d$, AGD updates the parameters as follows:

$$\widehat{\mathbf{w}}_a^\ell = \widehat{\mathbf{w}}_a^{\ell-1} + (\widehat{\mathbf{v}}^{\ell-1} - \widehat{\mathbf{w}}_a^{\ell-1}) - \eta\nabla L(\widehat{\mathbf{v}}^{\ell-1}), \tag{15a}$$

$$\widehat{\mathbf{v}}^\ell = \widehat{\mathbf{v}}^{\ell-1} + [\widehat{\mathbf{w}}_a^\ell + \frac{\sqrt{\kappa}-1}{\sqrt{\kappa}+1}(\widehat{\mathbf{w}}_a^\ell - \widehat{\mathbf{w}}_b^{\ell-1}) - \widehat{\mathbf{v}}^{\ell-1}], \tag{15b}$$

$$\widehat{\mathbf{w}}_b^\ell = \widehat{\mathbf{w}}_b^{\ell-1} + (\widehat{\mathbf{w}}_a^\ell - \widehat{\mathbf{w}}_b^{\ell-1}). \tag{15c}$$

We show that one attention layer and one MLP layer can implement one step of AGD as above. Namely, Eq. (15a) can be obtained using the same attention layer we constructed for gradient descent with $\widehat{\mathbf{v}}$ replacing $\widehat{\mathbf{w}}$, and an extra head with

$$\mathbf{Q}_4^{(\ell)}\mathbf{h}_{2t-1}^{(\ell-1)} = \begin{bmatrix} 2t-1 \\ -(2t-1)^2 \\ 1 \end{bmatrix}, \quad \mathbf{K}_4^{(\ell)}\mathbf{h}_i^{(\ell-1)} = \begin{bmatrix} 1 \\ 1 \\ i^2 \end{bmatrix}, \quad \mathbf{V}_4^{(\ell)}\mathbf{h}_{2t-1}^{(\ell-1)} = \begin{bmatrix} \mathbf{0} \\ \widehat{\mathbf{v}}^{\ell-1} - \widehat{\mathbf{w}}_a^{l-1} \\ \mathbf{0} \end{bmatrix}$$

for $i \leq 2t-1$ that gives $\widehat{\mathbf{v}}^{\ell-1} - \widehat{\mathbf{w}}_a^{\ell-1}$. Denote the output tokens of the attention layer by $\tilde{\mathbf{h}}$. Eq. (15b), (15c) can be implemented using one layer of MLP. Concretely, we choose $\mathbf{W}_1^{(\ell)}, \mathbf{W}_2^{(\ell)}$ such that

$$\mathbf{W}_1^{(\ell)}\tilde{\mathbf{h}}_{2t-1}^{(\ell-1)} = \begin{bmatrix} \mathbf{w}_a^\ell + \frac{\sqrt{\kappa}-1}{\sqrt{\kappa}+1}(\widehat{\mathbf{w}}_a^\ell - \widehat{\mathbf{w}}_b^{\ell-1}) - \widehat{\mathbf{v}}^{l-1} \\ -\mathbf{w}_a^\ell - \frac{\sqrt{\kappa}-1}{\sqrt{\kappa}+1}(\widehat{\mathbf{w}}_a^{\ell-1} - \widehat{\mathbf{w}}_b^{\ell-1}) + \widehat{\mathbf{v}}^{l-1} \\ \mathbf{w}_a^\ell - \mathbf{w}_b^{\ell-1} \\ -\mathbf{w}_a^\ell + \mathbf{w}_b^{\ell-1} \end{bmatrix}, \quad \mathbf{W}_2^{(\ell)}\sigma(\mathbf{W}_1^{(\ell)}\tilde{\mathbf{h}}_{2t-1}^{(\ell-1)}) = \begin{bmatrix} \mathbf{0} \\ \widehat{\mathbf{w}}_b^\ell \\ \widehat{\mathbf{v}}^\ell \\ \mathbf{0} \end{bmatrix}.$$

Since $t = \sigma(t) - \sigma(-t)$ for $t \in \mathbb{R}$, it is readily verified that one can choose the linear maps such that $\|\mathbf{W}_1^{(\ell)}\|_{\mathrm{op}} \leq 4\sqrt{2}, \|\mathbf{W}_2^{(\ell)}\|_{\mathrm{op}} = \sqrt{2}$. Combining this with the attention layer for Eq. (15a) and noting that $\|\mathbf{V}_4^{(\ell)}\|_{\mathrm{op}} = \sqrt{2}$, we verify that the transformer we constructed has norm $\|\boldsymbol{\theta}\| \leq 10 + (\lambda + 2)/(B_a^2 + \lambda)$. This completes the proof of Lemma 21. $\qquad\square$

### E.4 PROOF OF THEOREM 8

We construct a transformer that implements the following steps at each time $t \in [T]$ starting with $\mathbf{h}_{2t-1}^x = \mathbf{h}_{2t-1}^{\mathrm{pre},x}$ for $x \in \{a, b, c, d\}$

$$\mathbf{h}_{2t-1} = \begin{bmatrix} \mathbf{h}_{2t-1}^{\mathrm{pre},a} \\ \mathbf{h}_{2t-1}^{\mathrm{pre},b} \\ \mathbf{h}_{2t-1}^{\mathrm{pre},c} \\ \mathbf{h}_{2t-1}^{\mathrm{pre},d} \end{bmatrix} \xrightarrow{\text{step 1}} \begin{bmatrix} \mathbf{h}_{2t-1}^{\mathrm{pre},\{a,b,c\}} \\ \widehat{\mathbf{w}}_{\mathrm{ridge}} \\ \star \\ \mathbf{0} \\ \mathbf{pos} \end{bmatrix} \xrightarrow{\text{step 2}} \begin{bmatrix} \mathbf{h}_{2t-1}^{\mathrm{pre},\{a,b,c\}} \\ \widehat{\mathbf{w}}_{\mathrm{ridge}} \\ \star \\ \widehat{\mathbf{A}_t^{-1}\mathbf{a}_{t,1}} \\ \vdots \\ \widehat{\mathbf{A}_t^{-1}\mathbf{a}_{t,A}} \\ \mathbf{0} \\ \mathbf{pos} \end{bmatrix} \xrightarrow{\text{step 3}} \begin{bmatrix} \mathbf{h}_{2t-1}^{\mathrm{pre},\{a,b,c\}} \\ \widehat{\mathbf{w}}_{\mathrm{ridge}} \\ \star \\ \widehat{\mathbf{A}_t^{-1}\mathbf{a}_{t,1}} \\ \vdots \\ \widehat{\mathbf{A}_t^{-1}\mathbf{a}_{t,A}} \\ \widehat{v}_{t1}/\tau \\ \vdots \\ \widehat{v}_{tA}/\tau \\ \mathbf{0} \\ \mathbf{pos} \end{bmatrix} =: \begin{bmatrix} \mathbf{h}_{2t-1}^{\mathrm{post},a} \\ \mathbf{h}_{2t-1}^{\mathrm{post},b} \\ \mathbf{h}_{2t-1}^{\mathrm{post},c} \\ \mathbf{h}_{2t-1}^{\mathrm{post},d} \end{bmatrix},$$

$$(16)$$

where $\mathbf{pos} := [t, i^2, 1]^\top$; $\widehat{\mathbf{w}}_{\mathrm{ridge}}$ is an approximation to the ridge estimator $\mathbf{w}_{\mathrm{ridge},\lambda}^t$; $\widehat{\mathbf{A}_t^{-1}\mathbf{a}_{t,k}}$ are approximations to $\mathbf{A}_t^{-1}\mathbf{a}_{t,k}$; $\widehat{v}_{tk}$ are approximations to $v_{tk} := \langle \widehat{\mathbf{w}}_{\mathrm{ridge}}, \mathbf{a}_{t,k} \rangle + \alpha\sqrt{\langle \mathbf{a}_{t,k}, \widehat{\mathbf{A}_t^{-1}\mathbf{a}_{t,k}} \rangle}$, which are also approximations to

$$v_{tk}^* := \langle \mathbf{w}_{\mathrm{ridge},\lambda}^t, \mathbf{a}_{t,k} \rangle + \alpha\sqrt{\langle \mathbf{a}_{t,k}, \mathbf{A}_t^{-1}\mathbf{a}_{t,k} \rangle}$$

for $k \in [A]$. After passing through the transformer, we obtain the policy

$$\mathsf{Alg}_{\boldsymbol{\theta}}(\cdot|D_{t-1}, s_t) := \frac{\exp(\mathbf{h}_{2t-1}^{\mathrm{post},c})}{\|\exp(\mathbf{h}_{2t-1}^{\mathrm{post},c})\|_1} \in \Delta^A.$$

We claim the following results which we will prove later.

**Step 1** For any $\varepsilon > 0$, there exists a transformer $\mathrm{TF}_{\boldsymbol{\theta}}(\cdot)$ with

$$L = \left\lceil 2\sqrt{2T}\sqrt{\frac{B_a^2 + \lambda}{\lambda}} \log\left(\frac{(2T(B_a^2 + \lambda) + \lambda)TB_a(B_aB_w + \sigma)}{\lambda^2\varepsilon}\right)\right\rceil = \tilde{\mathcal{O}}(\sqrt{T}),$$

$$\max_{\ell \in [L]} M^{(l)} \leq 4, \quad \max_{\ell \in [L]} D'^{(l)} \leq 4d, \quad \|\boldsymbol{\theta}\| \leq 10 + \frac{\lambda + 2}{B_a^2 + \lambda} = \mathcal{O}(1)$$

that implements step 1 in (16) with $\|\widehat{\mathbf{w}}_{\mathrm{ridge}} - \mathbf{w}_{\mathrm{ridge},\lambda}^t\|_2 \leq \varepsilon$.

**Step 2** For any $\varepsilon > 0$, there exists a transformer $\mathrm{TF}_{\boldsymbol{\theta}}(\cdot)$ with

$$L = \left\lceil 2\sqrt{2T}\sqrt{\frac{B_a^2 + \lambda}{\lambda}} \log\left(\frac{(2T(B_a^2 + \lambda) + \lambda)B_a}{\lambda^2\varepsilon}\right)\right\rceil = \tilde{\mathcal{O}}(\sqrt{T}), \quad \max_{\ell \in [L]} M^{(l)} \leq 4A,$$

$$\max_{\ell \in [L]} D'^{(l)} \leq 4dA, \quad \|\boldsymbol{\theta}\| \leq 10 + A\left(\frac{\lambda + 3}{B_a^2 + \lambda} + \sqrt{2}\right) = \mathcal{O}(A)$$

that implements step 2 in (16) with $\|\widehat{\mathbf{A}_t^{-1}\mathbf{a}_{t,k}} - \mathbf{A}_t^{-1}\mathbf{a}_{t,k}\|_2 \leq \varepsilon$ for $k \in [A]$.

Step 3  Suppose that the approximation error in Step 2 satisfies $\varepsilon_2 \le b_a^2/[2(B_a^2+\lambda)TB_a]$. For any $\varepsilon > 0$, there exists a one-layer transformer $\mathrm{TF}_{\boldsymbol{\theta}}(\cdot)$ with

$$L = 2,\ \max_{\ell \in [L]} M^{(l)} \le 4A,\ \max_{\ell \in [L]} D'^{(\ell)} \le \mathcal{O}(A\sqrt{T\alpha/(\tau\varepsilon)}),\ \|\boldsymbol{\theta}\| \le \mathcal{O}(A + T(\alpha/(\tau\varepsilon))^{1/4} + \alpha/\tau)$$

that implements step 3 in (16) with $|\widehat{v}_{tk}/\tau - v_{tk}/\tau| \le \varepsilon$ for $k \in [A]$.

Denote the errors $\varepsilon$ appear in each step by $\varepsilon_1, \varepsilon_2, \varepsilon_3$, respectively. Define for all $k \in [A]$ that

$$v_{tk}^* := \langle \mathbf{w}_{\mathrm{ridge},\lambda}^t, \mathbf{a}_{t,k} \rangle + \alpha\sqrt{\langle \mathbf{a}_{t,k}, \mathbf{A}_t^{-1}\mathbf{a}_{t,k}\rangle},$$

which are the actual values used to compare across different actions in LinUCB. Then for all $k \in [A]$, we have the approximation error

$$
\begin{aligned}
\left|\frac{v_{tk}^*}{\tau} - \frac{\widehat{v}_{tk}}{\tau}\right| &\le \left|\frac{v_{tk}^*}{\tau} - \frac{v_{tk}}{\tau}\right| + \left|\frac{v_{tk}}{\tau} - \frac{\widehat{v}_{tk}}{\tau}\right| \\
&\le \frac{1}{\tau}|\langle \mathbf{w}_{\mathrm{ridge},\lambda}^t - \widehat{\mathbf{w}}_{\mathrm{ridge}}, \mathbf{a}_{t,k}\rangle| + \frac{1}{\tau}\left|\alpha\sqrt{\langle \mathbf{a}_{t,k}, \mathbf{A}_t^{-1}\mathbf{a}_{t,k}\rangle} - \alpha\sqrt{\langle \mathbf{a}_{t,k}, \widehat{\mathbf{A}_t^{-1}\mathbf{a}_{t,k}}\rangle}\right| + \varepsilon_3 \\
&\le \frac{B_a\varepsilon_1}{\tau} + \frac{\alpha B_a\varepsilon_2}{2\tau\min\left\{\sqrt{\langle \mathbf{a}_{t,k}, \mathbf{A}_t^{-1}\mathbf{a}_{t,k}\rangle}, \sqrt{\langle \mathbf{a}_{t,k}, \widehat{\mathbf{A}_t^{-1}\mathbf{a}_{t,k}}\rangle}\right\}} + \varepsilon_3 \\
&\le \frac{B_a\varepsilon_1}{\tau} + \frac{\sqrt{T(B_a^2+\lambda)}\alpha B_a\varepsilon_2}{b_a\tau} + \varepsilon_3,
\end{aligned}
$$

where the last line uses Eq. (17). For a targeted approximation error $\varepsilon$, choosing $\varepsilon_1 = \varepsilon\tau/(12B_a), \varepsilon_2 = \min\{b_a\tau\varepsilon/(12\sqrt{T(B_a^2+\lambda)}\alpha B_a), b_a^2/[2(B_a^2+\lambda)TB_a]\}$ and $\varepsilon_3 = \varepsilon/12$, we obtain $|v_{tk}^*/\tau - \widehat{v}_{tk}/\tau| \le \varepsilon/2$ for all $k \in [A]$.

From the proof of each step, we can verify that the token dimension $D$ can be chosen to be of order $O(dA)$. Moreover, due to the convergence guarantee for each iteration of AGD in Proposition 19, it can be verified that there exists some sufficiently large value $\mathsf{R} > 0$ with $\log \mathsf{R} = \tilde{\mathcal{O}}(1)$ such that we have $\|\mathbf{h}_i^{(\ell)}\|_2 \le \mathsf{R}$ for all layer $\ell \in [L]$ and all token $i \in [2T]$ in our TF construction. Therefore, $\mathrm{TF}_{\boldsymbol{\theta}}^{\mathsf{R}}$ and $\mathrm{TF}_{\boldsymbol{\theta}}^{\infty}$ generate the same output for all the token matrices we consider, and w.l.o.g. we may assume in the proof of each step that the transformers we consider are those without truncation (i.e., $\mathrm{TF}_{\boldsymbol{\theta}} = \mathrm{TF}_{\boldsymbol{\theta}}^{\infty}$).

Finally, combining Step 1—3 with $\alpha = \tilde{\mathcal{O}}(\sqrt{d})$ and applying Lemma 18 completes the proof of Theorem 8.

**Proof of Step 1**  We use the first $d$ entries of $\mathbf{h}_{2t-1}^d$ to represent $\widehat{\mathbf{w}}_{\mathrm{ridge}}$ and the $d+1 \sim 3d$ entries (denoted by $\star$) to record intermediate results for computing $\widehat{\mathbf{w}}_{\mathrm{ridge}}$. Step 1 follows immediately from the second part of Lemma 21.

**Proof of Step 2**  Note that

$$\mathbf{A}_t^{-1}\mathbf{a}_{t,k} = \arg\min_{\mathbf{x}\in\mathbb{R}^d} \frac{1}{2(2t-1)}\mathbf{x}^\top \mathbf{A}_t\mathbf{x} - \frac{1}{(2t-1)}\langle \mathbf{x}, \mathbf{a}_{t,k}\rangle =: \arg\min_{\mathbf{x}\in\mathbb{R}^d} L_k(\mathbf{x})$$

is the global minimizer of a $\lambda/(2t-1)$-strongly convex and $(B_a^2+\lambda)$-smooth quadratic function with the condition number $\kappa \le 2T(B_a^2+\lambda)/\lambda$. Moreover, we have

$$\|\mathbf{A}_t^{-1}\mathbf{a}_{t,k}\|_2 \le \|\mathbf{A}_t^{-1}\|_{\mathrm{op}}\|\mathbf{a}_{t,k}\|_2 \le B_a/\lambda.$$

It follows from Proposition 19 that $L = \lceil 2\sqrt{2T(B_a^2+\lambda)/\lambda}\log((1+\kappa)B_a/(\lambda\varepsilon))\rceil$ steps of accelerated gradient descent finds $\widehat{\mathbf{A}_t^{-1}\mathbf{a}_{t,k}}$ with $\|\widehat{\mathbf{A}_t^{-1}\mathbf{a}_{t,k}} - \mathbf{A}_t^{-1}\mathbf{a}_{t,k}\|_2 \le \varepsilon$.

Similar to the proof of Lemma 21, we can construct a transformer such that each (self-attention+MLP) layer implements one step of the accelerated gradient descent (AGD) for all $k \in [A]$. Denote the $(k+2)d+1 \sim (k+3)d, (A+1+2k)d+1 \sim (A+2+2k)d, (A+2+2k)d+1 \sim (A+3+2k)d$ entries of $\mathbf{h}_{2t-1}^d$ by $\widehat{\mathbf{w}}_{a,tk}, \widehat{\mathbf{w}}_{b,k}, \widehat{\mathbf{v}}_k$ for $k \in [A]$. Note that in the input vector $\mathbf{h}_{2t-1}^{\mathrm{pre},d}$ we have $\widehat{\mathbf{w}}_{a,tk}^0, \widehat{\mathbf{w}}_{b,k}^0, \widehat{\mathbf{v}}_k^0 = \mathbf{0}_d$.

For each layer $\ell \in [L]$ and $k \in [A]$, we choose $\mathbf{Q}^{(\ell)}_{k1,k2,k3,k4}, \mathbf{K}^{(\ell)}_{k1,k2,k3,k4}, \mathbf{V}^{(\ell)}_{k1,k2,k3,k4}$ such that for even tokens $\mathbf{h}_{2j}$ with $j \leq t-1$ and odd tokens $\mathbf{h}_{2j-1}$ with $j \leq t$

$$\mathbf{Q}^{(\ell)}_{k1}\mathbf{h}^{(\ell-1)}_{2t-1} = \begin{bmatrix} \widehat{\mathbf{v}}^{\ell-1}_k \\ \mathbf{0} \end{bmatrix}, \quad \mathbf{K}^{(\ell)}_{k1}\mathbf{h}^{(\ell-1)}_{2j} = \begin{bmatrix} \mathbf{a}_j \\ \mathbf{0} \end{bmatrix}, \quad \mathbf{K}^{(\ell)}_{k1}\mathbf{h}^{(\ell-1)}_{2j-1} = \mathbf{0}, \quad \mathbf{V}^{(\ell)}_{k1}\mathbf{h}^{(\ell-1)}_{2j} = -\eta \begin{bmatrix} \mathbf{0} \\ \mathbf{a}_j \\ \mathbf{0} \end{bmatrix}, \quad \mathbf{V}^{(\ell)}_{k1}\mathbf{h}^{(\ell-1)}_{2j-1} = \mathbf{0}$$

$$\mathbf{Q}^{(\ell)}_{k2} = -\mathbf{Q}^{(\ell)}_{k1}, \quad \mathbf{K}^{(\ell)}_{k2} = \mathbf{K}^{(\ell)}_{k1}, \quad \mathbf{V}^{(\ell)}_{k2} = -\mathbf{V}^{(\ell)}_{k1},$$

$$\mathbf{Q}^{(\ell)}_{k3}\mathbf{h}^{(\ell-1)}_{2t-1} = \begin{bmatrix} 1 \\ 1-2t \\ 1 \\ 0 \end{bmatrix}, \quad \mathbf{K}^{(\ell)}_{k3}\mathbf{h}^{(\ell-1)}_{2j} = \begin{bmatrix} 1 \\ 1 \\ 2j \\ 0 \end{bmatrix}, \quad \mathbf{K}^{(\ell)}_{k3}\mathbf{h}^{(\ell-1)}_{2j-1} = \begin{bmatrix} 1 \\ 1 \\ 2j-1 \\ 0 \end{bmatrix}, \quad \mathbf{V}^{(\ell)}_{k3}\mathbf{h}^{(\ell-1)}_{2t-1} = \eta \begin{bmatrix} 0 \\ \mathbf{a}_{j,k} - \lambda\widehat{\mathbf{v}}^{\ell-1}_k \\ 0 \end{bmatrix},$$

$$\mathbf{Q}^{(\ell)}_{k4}\mathbf{h}^{(\ell-1)}_{2t-1} = \begin{bmatrix} 2t-1 \\ -(2t-1)^2 \\ 1 \\ 0 \end{bmatrix}, \quad \mathbf{K}^{(\ell)}_{k4}\mathbf{h}^{(\ell)}_{2j} = \begin{bmatrix} 1 \\ 1 \\ (2j)^2 \\ 0 \end{bmatrix}, \quad \mathbf{K}^{(\ell)}_{k4}\mathbf{h}^{(\ell)}_{2j-1} = \begin{bmatrix} 1 \\ 1 \\ (2j-1)^2 \\ 0 \end{bmatrix}, \quad \mathbf{V}^{(\ell)}_{k4}\mathbf{h}^{(\ell-1)}_{2t-1} = \begin{bmatrix} 0 \\ \widehat{\mathbf{v}}^{\ell-1}_k - \widehat{\mathbf{w}}^{\ell-1}_{a,k} \\ 0 \end{bmatrix},$$

where $\eta = 1/(B^2_a + \lambda)$ and the values $\mathbf{V}^{(\ell)}_{kt}\mathbf{h}^{(\ell-1)}_{2j}, \mathbf{V}^{(\ell)}_{kt}\mathbf{h}^{(\ell-1)}_{2j-1}$, $t = 1,2,3,4$ are supported on the entries corresponding to $\widehat{\mathbf{w}}_{a,k}$. Summing up the $M = 4A$ heads and noting that $t = \sigma(t) - \sigma(-t)$, we see that the $\widehat{\mathbf{w}}_{a,k}$ part of $\mathbf{h}_t$ has the update

$$\widehat{\mathbf{w}}^\ell_{a,k} = \widehat{\mathbf{w}}^{\ell-1}_{a,k} - \frac{\eta}{2t-1}\sum^{t-1}_{j=1}[\sigma(\langle \mathbf{a}_j, \widehat{\mathbf{v}}^{\ell-1}_k\rangle) - \sigma(-\langle \mathbf{a}_j, \widehat{\mathbf{v}}^{\ell-1}_k\rangle)]\mathbf{a}_j - \frac{\eta\lambda}{2t-1}\mathbf{V}^{(\ell)}_{k3}\mathbf{h}^{(\ell-1)}_{2t-1} + \mathbf{V}^{(\ell)}_{k4}\mathbf{h}^{(\ell-1)}_{2t-1}$$

$$= \widehat{\mathbf{w}}^{\ell-1}_{a,k} - \frac{\eta}{2t-1}\sum^{t-1}_{j=1}\langle \mathbf{a}_j, \widehat{\mathbf{v}}^{\ell-1}_k\rangle \mathbf{a}_j - \frac{\eta\lambda}{2t-1}\mathbf{V}^{(\ell)}_{k3}\mathbf{h}^{(\ell-1)}_{2t-1} + \mathbf{V}^{(\ell)}_{k4}\mathbf{h}^{(\ell-1)}_{2t-1}$$

$$= \widehat{\mathbf{v}}^{\ell-1}_k - \frac{\eta}{2t-1}\sum^{t-1}_{j=1}\langle \mathbf{a}_j, \widehat{\mathbf{v}}^{\ell-1}_k\rangle \mathbf{a}_j - \frac{\eta\lambda}{2t-1}\widehat{\mathbf{v}}^{\ell-1}_k + \frac{\eta}{2t-1}\mathbf{a}_{t,k}$$

$$= \widehat{\mathbf{v}}^{\ell-1}_k - \eta\nabla L(\mathbf{v}^{\ell-1}_k),$$

which is one step of gradient descent with step size $\eta$ (c.f. Eq. 15a). Moreover, it can be verified that one can choose the matrices such that $\max_{k\in[A],m\in[4]}\|\mathbf{Q}^{(\ell)}_{km}\|_{\mathrm{op}} = \max_{k\in[A],m\in[4]}\|\mathbf{K}^{(\ell)}_{km}\|_{\mathrm{op}} \leq \sqrt{2}$ and $\max_{k\in[A]}\|\mathbf{V}^{(\ell)}_{k1}\|_{\mathrm{op}} = \max_{k\in[A]}\|\mathbf{V}^{(\ell)}_{k2}\|_{\mathrm{op}} = \eta$, $\max_{k\in[A]}\|\mathbf{V}^{(\ell)}_{k3}\|_{\mathrm{op}} \leq (\lambda + 1)\eta$, $\max_{k\in[A]}\|\mathbf{V}^{(\ell)}_{k4}\|_{\mathrm{op}} \leq \sqrt{2}$. Therefore, the norm of the attention layer

$$\|\boldsymbol{\theta}\| \leq \sqrt{2}(A+1) + A(\lambda+3)/(B^2_a + \lambda).$$

Following the construction as in the proof of Lemma 21, we can choose $\mathbf{W}^{(\ell)}_1, \mathbf{W}^{(\ell)}_2$ that implement Eq. (15b), (15b) for all $k \in [A]$ simultaneously and we also have $\|\mathbf{W}^{(\ell)}_1\|_{\mathrm{op}} \leq 4\sqrt{2}, \mathbf{W}^{(\ell)}_2 = \sqrt{2}$ with $D''^{(\ell)} = 4dA$. It follows from combining the bounds for the weight matrices that

$$\|\boldsymbol{\theta}\| \leq \sqrt{2}(A+1) + A(\frac{\lambda+3}{B^2_a + \lambda} + \sqrt{2}) + \sqrt{2} + 4\sqrt{2} \leq 10 + A(\frac{\lambda+3}{B^2_a + \lambda} + \sqrt{2}) = \mathcal{O}(A).$$

**Proof of Step 3** Denote the $i$-th token of the output of step 2 (i.e., the input of step 3) by $\mathbf{h}^{(0)}_i$. We use the $(3A+3)d + 1 \sim (3A+3)d + A$ entries of $\mathbf{h}^d_{2t-1}$ to record $\widehat{v}_{t1}/\tau, \ldots, \widehat{v}_{tA}/\tau$ and use the $(3A+3)d + A + 1 \sim (3A+3)d + 2A$ entries to store additional information (denoted by $v_{a,t1}, \ldots, v_{a,tA}$) for computing $\widehat{v}_{t1}/\tau, \ldots, \widehat{v}_{tA}/\tau$. Concretely, for all $k \in [A]$, we choose $\mathbf{Q}^{(\ell)}_{k1,k2,k3,k4}, \mathbf{K}^{(\ell)}_{k1,k2,k3,k4}, \mathbf{V}^{(\ell)}_{k1,k2,k3,k4}$ such that for even tokens $\mathbf{h}_{2j}$ with $j \leq t-1$ and odd tokens $\mathbf{h}_{2j-1}$ with $j \leq t$

$$\mathbf{Q}^{(1)}_{k1}\mathbf{h}^{(0)}_{2t-1} = \begin{bmatrix} \widehat{\mathbf{w}}_{\mathrm{ridge}} \\ 2t-1 \\ 1 \\ 0 \end{bmatrix}, \quad \mathbf{K}^{(1)}_{k1}\mathbf{h}^{(0)}_{2j-1} = \begin{bmatrix} \mathbf{a}_{j,k} \\ -\mathsf{B} \\ \mathsf{B}(2j-1) \\ 0 \end{bmatrix}, \quad \mathbf{K}^{(1)}_{k1}\mathbf{h}^{(0)}_{2j} = \begin{bmatrix} \mathbf{0}_d \\ -\mathsf{B} \\ 2\mathsf{B}j \\ 0 \end{bmatrix},$$

$$\mathbf{V}_{k1}^{(1)}\mathbf{h}_{2j-1}^{(0)} = \begin{bmatrix} \mathbf{0} \\ 2j-1 \\ \mathbf{0} \end{bmatrix}, \quad \mathbf{V}_{k1}^{(1)}\mathbf{h}_{2j}^{(0)} = \begin{bmatrix} \mathbf{0} \\ 2j \\ \mathbf{0} \end{bmatrix},$$

$$\mathbf{Q}_{k2}^{(1)}\mathbf{h}_{2t-1}^{(0)} = \begin{bmatrix} -\widehat{\mathbf{w}}_{\mathrm{ridge}} \\ 2t-1 \\ 1 \\ \mathbf{0} \end{bmatrix}, \quad \mathbf{K}_{k2}^{(1)} = \mathbf{K}_{k1}^{(1)}, \quad \mathbf{V}_{k2}^{(1)} = -\mathbf{V}_{k1}^{(1)},$$

$$\mathbf{Q}_{k3}^{(1)}\mathbf{h}_{2t-1}^{(0)} = \begin{bmatrix} \widehat{\mathbf{A}_t^{-1}\mathbf{a}_{t,k}} \\ 2t-1 \\ 1 \\ \mathbf{0} \end{bmatrix}, \quad \mathbf{K}_{k3}^{(1)}\mathbf{h}_{2j-1}^{(0)} = \begin{bmatrix} \mathbf{a}_{j,k} \\ -\mathsf{B} \\ \mathsf{B}(2j-1) \\ \mathbf{0} \end{bmatrix}, \quad \mathbf{K}_{k3}^{(1)}\mathbf{h}_{2j}^{(0)} = \begin{bmatrix} \mathbf{0}_d \\ -\mathsf{B} \\ 2\mathsf{B}j \\ \mathbf{0} \end{bmatrix},$$

$$\mathbf{V}_{k3}^{(1)}\mathbf{h}_{2j-1}^{(0)} = \begin{bmatrix} \mathbf{0} \\ 2j-1 \\ \mathbf{0} \end{bmatrix}, \quad \mathbf{V}_{k3}^{(1)}\mathbf{h}_{2j}^{(0)} = \begin{bmatrix} \mathbf{0} \\ 2j \\ \mathbf{0} \end{bmatrix},$$

$$\mathbf{Q}_{k4}^{(1)}\mathbf{h}_{2t-1}^{(0)} = \begin{bmatrix} -\widehat{\mathbf{A}_t^{-1}\mathbf{a}_{t,k}} \\ 2t-1 \\ 1 \\ \mathbf{0} \end{bmatrix}, \quad \mathbf{K}_{k4}^{(1)} = \mathbf{K}_{k3}^{(1)}, \quad \mathbf{V}_{k4}^{(1)} = -\mathbf{V}_{k3}^{(1)},$$

where $\mathsf{B} := TB_a^2(B_aB_w + \sigma)/\lambda + 2B_a^2/\lambda$; $\mathbf{V}_{k1}^{(1)}\mathbf{h}_c^{(0)}, \mathbf{V}_{k2}^{(1)}\mathbf{h}_c^{(0)}(c = 2j-1, 2j)$ are supported on the $[(3A+3)d+k]$-th entry of $\mathbf{h}_c^d$; $\mathbf{V}_{k3}^{(1)}\mathbf{h}_c^{(0)}, \mathbf{V}_{k4}^{(1)}\mathbf{h}_c^{(0)}(c = 2j-1, 2j)$ are supported on the $[(3A+3)d+A+k]$-th entry of $\mathbf{h}_c^d$.

Since $\langle\widehat{\mathbf{w}}_{\mathrm{ridge}}, \mathbf{a}_{j,k}\rangle \le \|\widehat{\mathbf{w}}_{\mathrm{ridge}}\|_2\|\mathbf{a}_{j,k}\|_2 \le \mathsf{B}$, it follows that

$$\left\langle \mathbf{Q}_{k1}^{(1)}\mathbf{h}_{2t-1}^{(0)}, \mathbf{K}_{k1}^{(1)}\mathbf{h}_{2j-1}^{(0)} \right\rangle = \langle\widehat{\mathbf{w}}_{\mathrm{ridge}}, \mathbf{a}_{j,k}\rangle + (2j-1-(2t-1))\mathsf{B} \le 0$$

for $j < i$. Likewise $\left\langle \mathbf{Q}_{k1}^{(1)}\mathbf{h}_{2t-1}^{(0)}, \mathbf{K}_{k1}^{(1)}\mathbf{h}_{2j}^{(0)} \right\rangle \le 0$ for $j < i$. Since we assume the error $\varepsilon_2 \le b_a^2/[2(B_a^2T + \lambda)B_a]$ in Step 2, $b_a \le \|\mathbf{a}_{t,k}\|_2 \le B_a$ and $\lambda\mathbf{I}_d \preceq \mathbf{A}_t \preceq (B_a^2T + \lambda)\mathbf{I}_d$, it follows that

$$\left\langle \mathbf{a}_{t,k}, \widehat{\mathbf{A}_t^{-1}\mathbf{a}_{t,k}} \right\rangle \ge \langle\mathbf{a}_{t,k}, \mathbf{A}_t^{-1}\mathbf{a}_{t,k}\rangle - \|\mathbf{a}_{t,k}\|_2\|\mathbf{A}_t^{-1}\mathbf{a}_{t,k} - \widehat{\mathbf{A}_t^{-1}\mathbf{a}_{t,k}}\|_2$$

$$\ge \frac{b_a^2}{2(B_a^2T + \lambda)} \ge \frac{b_a^2}{2T(B_a^2 + \lambda)} =: \frac{1}{T} \cdot l_1, \tag{17}$$

$$\left\langle \mathbf{a}_{t,k}, \widehat{\mathbf{A}_t^{-1}\mathbf{a}_{t,k}} \right\rangle \le \langle\mathbf{a}_{t,k}, \mathbf{A}_t^{-1}\mathbf{a}_{t,k}\rangle + \|\mathbf{a}_{t,k}\|_2\|\mathbf{A}_t^{-1}\mathbf{a}_{t,k} - \widehat{\mathbf{A}_t^{-1}\mathbf{a}_{t,k}}\|_2 \le \frac{2B_a^2}{\lambda} =: l_2. \tag{18}$$

Therefore, $\left\langle \mathbf{Q}_{k3}^{(1)}\mathbf{h}_{2t-1}^{(0)}, \mathbf{K}_{k3}^{(1)}\mathbf{h}_{2j-1}^{(0)} \right\rangle = \left\langle \widehat{\mathbf{A}_t^{-1}\mathbf{a}_{t,k}}, \mathbf{a}_{j,k} \right\rangle + (2j-1-(2t-1))\mathsf{B} \ge 0$ iff $j = i$. Likewise $\left\langle \mathbf{Q}_{k3}^{(1)}\mathbf{h}_{2t-1}^{(0)}, \mathbf{K}_{k3}^{(1)}\mathbf{h}_{2j}^{(0)} \right\rangle \le 0$ for $j < i$. Similar results hold for the $k2, k4$-th heads. By some basic algebra and noting that $t = \sigma(t) - \sigma(-t)$ for $t \in \mathbb{R}$, we see that the attention layer updates the position for $\widehat{v}_{tk}/\tau, \widehat{v}_{a,tk}$ with the values $\langle\widehat{\mathbf{w}}_{\mathrm{ridge}}, \mathbf{a}_{t,k}\rangle, \left\langle \mathbf{a}_{t,k}, \widehat{\mathbf{A}_t^{-1}\mathbf{a}_{t,k}} \right\rangle$ for all $k \in [A]$, respectively. Moreover, it can be verified that one can choose the matrices such that

$$\max_{k\in[A],m\in[4]} \|\mathbf{Q}_{km}^{(1)}\|_{\mathrm{op}} = \max_{k\in[A],m\in[4]} \|\mathbf{V}_{km}^{(1)}\|_{\mathrm{op}} = 1, \quad \max_{k\in[A],m\in[4]} \|\mathbf{K}_{km}^{(1)}\|_{\mathrm{op}} \le \mathsf{B}.$$

Now, to compute the value of $\widehat{v}_{tk}/\tau$ in step 3 in (16), what remains is to approximately compute $\alpha\sqrt{\widehat{v}_{a,tk}}$, add the result to the position for $\widehat{v}_{tk}/\tau$, and multiplied it by $1/\tau$.

Since $\widehat{v}_{a,tk} = \left\langle \mathbf{a}_{t,k}, \widehat{\mathbf{A}_t^{-1}\mathbf{a}_{t,k}} \right\rangle \in [l_1/T, l_2]$, to approximately compute $\sqrt{\left\langle \mathbf{a}_{t,k}, \widehat{\mathbf{A}_t^{-1}\mathbf{a}_{t,k}} \right\rangle}$, it suffices to approximate $f(x) = \sqrt{x}, x \in [l_1/T, l_2]$. For any level of approximation error $\varepsilon_{\mathrm{approx}} > 0$, let $(x_1, x_2, \ldots, x_N) \in [l_1/T, l_2]$ satisfy

$$x_1 = l_1/T, \; x_N = l_2, \quad 0 \le \sqrt{x_{j+1}} - \sqrt{x_j} \le \varepsilon_{\mathrm{approx}}, \quad \text{for } j \in [N-1].$$

Define the function

$$\tilde{f}(x) := \sqrt{x_1} + \sum_{j=1}^{N-1} \sigma(x - x_j) \frac{1}{\sqrt{x_{j+1}} + \sqrt{x_j}}.$$

Note that $\tilde{f}(x)$ is a piecewise linear function on $[l_1/T, l_2]$ with $\tilde{f}(x_i) = \sqrt{x_i}$ for $i \in [N]$. By some basic algebra, it can the shown that for $\varepsilon_{\text{approx}} < l_1/T$, the difference between $f(x)$ and $\tilde{f}(x)$

$$\max_{c \in [x_j, x_{j+1}]} |\tilde{f}(c) - f(c)| = \max_{t \in [0,1]} \left| \sqrt{x_j} + \frac{t}{\sqrt{x_j + 1} + \sqrt{x_j}} - \sqrt{x_j + t(x_{j+1} - x_j)} \right| \le \varepsilon_{\text{approx}}$$

when $\sqrt{x_{j+1}} - \sqrt{x_j} < c\sqrt{\varepsilon_{\text{approx}} l_1/T}$ for some universal constant $c > 0$ and all $j \in [N - 1]$.

Therefore, there exists a function $\tilde{f}(x)$ with $N = \mathcal{O}(\sqrt{T/\varepsilon_{\text{approx}}})$ that satisfies

$$\max_{[l_1/T, l_2]} |\tilde{f}(x) - f(x)| \le \varepsilon_{\text{approx}}.$$

As a consequence, we verify that one can implement $\tilde{f}(\hat{v}_{a,tk})$ for all $k \in [A]$ simultaneously by constructing a two-layer MLP with

$$\|\mathbf{W}_1^{(1)}\|_{\text{op}} \le \mathrm{O}(\sqrt{N}), \quad \|\mathbf{W}_2^{(1)}\|_{\text{op}} \le \mathcal{O}(\sqrt{TN}), \quad D' \le AN.$$

Choose $\varepsilon_{\text{approx}} = \tau\varepsilon/\alpha$. Substituting the expressions for $N, \varepsilon_{\text{approx}}$ into the upper bounds on the norms, we obtain

$$\|\mathbf{W}_1^{(1)}\|_{\text{op}} \le \mathcal{O}((\alpha T/(\tau\varepsilon))^{1/4}), \quad \|\mathbf{W}_2^{(1)}\|_{\text{op}} \le \mathcal{O}(T^{3/4}(\alpha/(\tau\varepsilon))^{1/4}), \quad D' \le \mathcal{O}(A(\alpha T/(\tau\varepsilon))^{1/2}).$$

Lastly, we can construct another two-layer MLP with weights $\mathbf{W}_1^{(2)}, \mathbf{W}_2^{(2)}$ such that it implements the summation and multiplication updates

$$\hat{\mathbf{v}} \leftarrow \hat{\mathbf{v}} + \mathbf{W}_2^{(2)}\sigma(\mathbf{W}_1^{(2)}\mathbf{h}_{2t-1}^{(1)}) \approx \hat{\mathbf{v}} + \left(\frac{1}{\tau} - 1\right)\hat{\mathbf{v}} + \frac{\alpha}{\tau}\left[\sqrt{\hat{v}_{a,t1}}, \ldots, \sqrt{\hat{v}_{a,tA}}\right]^\top = \frac{\hat{v}_{tk}}{\tau}$$

with $\|\hat{v}_{tk}/\tau - v_{tk}/\tau\| \le \varepsilon$ for all $k \in [A]$. We verify that the weight matrices can be chosen with

$$\|\mathbf{W}_1^{(2)}\|_{\text{op}} \le \mathrm{O}(1), \quad \|\mathbf{W}_2^{(2)}\|_{\text{op}} \le \mathrm{O}(\alpha/\tau)$$

and $D' \le \mathrm{O}(A)$.

Therefore the norm of the transformer that implements step 3 satisfies

$$\|\boldsymbol{\theta}\| \le \mathrm{O}(\mathsf{B} + 1 + 4A + T^{3/4}(\alpha/(\tau\varepsilon))^{1/4} + \alpha/\tau) = \mathcal{O}(A + T(\alpha/(\tau\varepsilon))^{1/4} + \alpha/\tau).$$

This conclude the proof of Step 3.

### E.5 Proof of Theorem 9

By Theorem 6 and Theorem 8 with $\varepsilon_{\text{real}} = \varepsilon = 1/T^3$, it suffices to show soft LinUCB with parameter $\tau$ has the regret guarantee

$$\mathbb{E}_{\mathsf{M} \sim \Lambda}\left[\sum_{t=1}^{T} \max_k \langle \mathbf{a}_{t,k}, \mathbf{w}^* \rangle - \mathfrak{R}_{\mathsf{M}, \mathsf{Alg}_{\mathrm{sLinUCB}(\tau)}}(T)\right] \le \mathcal{O}(d\sqrt{T}\log(T)).$$

This follows directly from a regret analysis similar to that for LinUCB (see e.g. Chu et al. (2011) or Theorem 19.2 in Lattimore & Szepesvári (2020)). Concretely, note that $v_{tk}^* = \left\langle \mathbf{w}_{\text{ridge},\lambda}^t, \mathbf{a}_{t,k} \right\rangle + \alpha\sqrt{\left\langle \mathbf{a}_{t,k}, \mathbf{A}_t^{-1}\mathbf{a}_{t,k} \right\rangle}$ is the solution to the optimization problem

$$\begin{aligned} \text{maximize} \quad & \langle \mathbf{w}, \mathbf{a}_{t,k} \rangle \\ \text{subject to} \quad & \mathbf{w} \in \mathbb{C}_t := \{\mathbf{w} | (\mathbf{w} - \mathbf{w}_{\text{ridge},\lambda}^t)^\top \mathbf{A}_t(\mathbf{w} - \mathbf{w}_{\text{ridge},\lambda}^t) \le \alpha^2\}, \end{aligned}$$

where we recall $\alpha = \alpha(\delta_0)$ with $\delta_0 = 1/(2B_aB_wT)$ and

$$\alpha = \alpha(\delta_0) := \sqrt{\lambda}B_w + \sigma\sqrt{2\log(1/\delta_0) + d\log((d\lambda + TB_a^2)/(d\lambda))}. \tag{19}$$

Moreover, standard analysis as in the proof of Theorem 19.2 in Lattimore & Szepesvári (2020) shows with probability over $1 - 1/(2B_aB_wT)$ we have $\mathbf{w}^* \in \mathbb{C}_t$ for all $t \in [T]$. Denote this event by $\mathcal{E}_0$. Moreover, let $p_{t,k}$ denote the probability of soft LinUCB selecting the action $\mathbf{a}_{t,k}$ at time $t$ for all $k \in [A]$. For any $\varepsilon > 0$, let $\mathbb{S}_t(\varepsilon) := \{k \in [A] : v_{tk}^* - \max_{j \in [A]} v_{tj}^* \leq \varepsilon\}$.

Therefore, on the event $\mathcal{E}_0$ at time $t$ we have

$$\max_j v_{tj}^* - \sum_{k=1}^A p_{t,k} v_{tk}^* = \sum_{k \in \mathbb{S}_t(\varepsilon_0)} p_{t,k}(\max_j v_{tj}^* - v_{tk}^*) + \sum_{k \notin \mathbb{S}_t(\varepsilon_0)} p_{t,k}(\max_j v_{tj}^* - v_{tk}^*)$$

$$\leq \varepsilon_0 + \sum_{k \notin \mathbb{S}_t(\varepsilon_0)} \exp\Big(-\frac{\varepsilon_0}{\tau}\Big)(\max_j v_{tj}^* - v_{tk}^*)$$

$$\leq \varepsilon_0 + 2A\exp\Big(-\frac{\varepsilon_0}{\tau}\Big)B_a(B_w + 2\alpha/\sqrt{\lambda}),$$

where the second line uses

$$p_{t,k} \leq \exp\Big(-\frac{\varepsilon_0}{\tau}\Big) \cdot \max p_{t,k} \leq \exp\Big(-\frac{\varepsilon_0}{\tau}\Big),$$

and the last line follows from that $|v_{tj}^*| \leq B_a(B_w + 2\alpha/\sqrt{\lambda})$ on the event $\mathcal{E}_0$. Choosing $\varepsilon_0 = \varepsilon_1/2 := 1/\sqrt{4T}$ and noting that $\tau = \varepsilon_0/\log(4TAB_a(B_w + 2\alpha/\sqrt{\lambda})) = \tilde{\mathcal{O}}(1/\sqrt{T})$, we obtain

$$\max_j v_{tj}^* - \sum_{k=1}^A p_{t,k} v_{tk}^* \leq \varepsilon_1.$$

Now, on the event $\mathcal{E}_0$, we have

$$\max_{j \in [A]} \langle \mathbf{w}^*, \mathbf{a}_{t,j} \rangle \leq \max_{j \in [A]} v_{tj}^* \leq \sum_{k=1}^A p_{t,k} v_{tk}^* + \varepsilon_1 = \sum_{k=1}^A p_{t,k} \langle \tilde{\mathbf{w}}_k, \mathbf{a}_{t,k} \rangle + \varepsilon_1$$

for some $\tilde{\mathbf{w}}_k \in \mathbb{C}_t, k \in [A]$. Therefore, on $\mathcal{E}_0$ for each $t \in [T]$

$$\max_{j \in [A]} \langle \mathbf{w}^*, \mathbf{a}_{t,j} \rangle - \sum_{k=1}^A p_{t,k} \langle \mathbf{w}^*, \mathbf{a}_{t,k} \rangle \leq \varepsilon_1 + \sum_{k=1}^A p_{t,k} \langle \tilde{\mathbf{w}}_k - \mathbf{w}^*, \mathbf{a}_{t,k} \rangle$$

$$\leq \varepsilon_1 + \sum_{k=1}^A p_{t,k} \|\tilde{\mathbf{w}}_k - \mathbf{w}^*\|_{\mathbf{A}_t} \cdot \|\mathbf{a}_{t,k}\|_{\mathbf{A}_t^{-1}} \leq \varepsilon_1 + 2\alpha\mathbb{E}_{k\sim\mathbf{p}_t}\|\mathbf{a}_{t,k}\|_{\mathbf{A}_t^{-1}}.$$

Moreover, note that $\max_{j \in [A]} \langle \mathbf{w}^*, \mathbf{a}_{t,j} \rangle - \langle \mathbf{w}^*, \mathbf{a}_{t,k} \rangle \leq 2B_wB_a$ and $\|\mathbf{a}_{t,k}\|_{\mathbf{A}_t^{-1}} \leq B_a/\sqrt{\lambda}$. Summing over $t \in [T]$ and using the tower property of martingales, we obtain

$$\mathbb{E}_{M\sim\Lambda}\Big[\sum_{t=1}^T \max_k \langle \mathbf{a}_{t,k}, \mathbf{w}^* \rangle - \mathfrak{R}_{M,\mathsf{Alg}_{\mathsf{sLinUCB}(\tau)}}(T)\Big]$$

$$= \mathbb{E}\Big[\max_{j \in [A]} \langle \mathbf{w}^*, \mathbf{a}_{t,j} \rangle - \sum_{k=1}^A p_{t,k} \langle \mathbf{w}^*, \mathbf{a}_{t,k} \rangle\Big]$$

$$\leq \mathbb{E}[2\sum_{t=1}^T \alpha\mathbb{E}_{k\sim\mathbf{p}_t}\|\mathbf{a}_{t,k}\|_{\mathbf{A}_t^{-1}} + \varepsilon_1 T + 2B_wB_aT \cdot \mathbf{1}_{\{\mathcal{E}_0^c\}}]$$

$$\leq \mathbb{E}[2\sum_{t=1}^T \alpha\Big(\frac{B_a}{\sqrt{\lambda}} \wedge \|\mathbf{a}_{t,k}\|_{\mathbf{A}_t^{-1}}\Big) + \varepsilon_1 T + 2B_wB_aT \cdot \mathbf{1}_{\{\mathcal{E}_0^c\}}]$$

$$\leq 2\mathbb{E}\Big[\alpha\sqrt{T}(\frac{B_a}{\sqrt{\lambda}} + 1)\sqrt{\sum_{t=1}^T (1 \wedge \|\mathbf{a}_{t,k}\|_{\mathbf{A}_t^{-1}}^2)} + \varepsilon_1 T\Big] + 2B_wB_aT\mathbb{P}(\mathcal{E}_0^c)$$

$$\leq \sqrt{8d(B_a/\sqrt{\lambda}+1)^2 T\alpha^2 \log((d\lambda + TB_a^2)/(d\lambda))} + \varepsilon_1 T + 1,$$

where the fourth line uses the fact that

$$\frac{B_a}{\sqrt{\lambda}} \wedge \|\mathbf{a}_{t,k}\|_{\mathbf{A}_t^{-1}} \leq (\frac{B_a}{\sqrt{\lambda}}+1) \cdot (1 \wedge \|\mathbf{a}_{t,k}\|_{\mathbf{A}_t^{-1}}^2)$$

and Cauchy-Schwartz inequality, the last line follows from Lemma 19.4 of Lattimore & Szepesvári (2020). Plugging in $\varepsilon_1 = 1/\sqrt{T}$ and Eq. (19) gives the upper bound on expected regret

$$\mathbb{E}_{\mathsf{M}\sim\Lambda}\Big[\sum_{t=1}^{T} \max_k \langle \mathbf{a}_{t,k}, \mathbf{w}^*\rangle - \mathfrak{R}_{\mathsf{M},\mathsf{Alg}_{\mathrm{sLinUCB}(\tau)}}(T)\Big] \leq \mathcal{O}(d\sqrt{T}\log(T))$$

for soft LinUCB with parameter $\tau$.

Moreover, the second part of Theorem 9 (i.e., the upper bound on $\log \mathcal{N}_\Theta$) follows directly from Lemma 16 and Eq. (8).

## F  THOMPSON SAMPLING FOR STOCHASTIC LINEAR BANDIT

Throughout this section, we use $c > 0$ to denote universal constants whose values may vary from line to line. Moreover, for notational simplicity, we use $\mathrm{O}(\cdot)$ to hide universal constants, $\mathcal{O}(\cdot)$ to hide polynomial terms in the problem parameters $(\lambda^{\pm 1}, r^{\pm 1}, b_a^{-1}, B_a)$, and $\tilde{\mathcal{O}}(\cdot)$ to hide both poly-logarithmic terms in $(\mathsf{M}_0, \mathsf{C}_0, T, A, d, 1/\varepsilon, 1/\delta_0)$ and polynomial terms in $(\lambda^{\pm 1}, r^{\pm 1}, b_a^{-1}, B_a)$. We also use the bold font letter $\mathbf{a}_t \in \mathbb{R}^d$ to denote the selected action $a_t$ at time $t \in [T]$.

This section is organized as follows. Section F.1 describes the Thompson sampling algorithm for stochastic linear bandits. Section F.2 introduces some additional definitions, assumptions, and the formal version of Theorem 10 as in Theorem 23. We prove Theorem 23 in Section F.3 and prove Theorem 11 in Section F.4. Lastly, the proof of Lemma 24 used in the proof of Theorem 23 is provided in Section F.5.

### F.1  THOMPSON SAMPLING ALGORITHM

Consider the stochastic linear bandit setup as in Section 4.1, but instead we assume a Gaussian prior distribution $\mathbf{w}^\star \sim \mathcal{N}(0, \lambda \mathbf{I}_d)$ and Gaussian noises $\{\varepsilon_t\}_{t\geq 0} \sim_{iid} \mathcal{N}(0, r)$. Furthermore, we assume there exist $(b_a, B_a)$ such that $b_a \leq \|\mathbf{a}_{t,k}\|_2 \leq B_a$. At each time $t \in [T]$, Thompson sampling consists of the following steps:

1. Computes

$$\mu_t := \Big(\frac{r}{\lambda}\mathbf{I}_d + \sum_{j=1}^{t-1}\mathbf{a}_j\mathbf{a}_j^\top\Big)^{-1}\sum_{j=1}^{t-1}\mathbf{a}_j y_j, \quad \boldsymbol{\Sigma}_t := \frac{r}{\lambda}\mathbf{I}_d + \sum_{j=1}^{t-1}\mathbf{a}_j\mathbf{a}_j^\top.$$

2. Selects the action $\mathbf{a}_t = \mathbf{a}_{t,k}$ with probability

$$\mathbb{P}_{\tilde{\mathbf{w}}_t \sim \mathcal{N}(\mu_t, r\boldsymbol{\Sigma}_t^{-1})}\Big(k = \arg\max_{j\in[A]} \langle \mathbf{a}_{t,j}, \tilde{\mathbf{w}}_t\rangle\Big).$$

Note that Thompson sampling is equivalent to the posterior sampling procedure in our stochastic linear bandit setup, i.e., we select an action with probability that equals the posterior probability of the action being optimal. We allow $\lambda$ to be either some constant independent of $T, d$, or has the form $\lambda = \lambda_0/d$ for some constant $\lambda_0 > 0$. The latter case is considered so that the bandit parameter vector $\mathbf{w}^*$ has $\ell_2$ norm of order $\tilde{\mathcal{O}}(1)$ with high probability. In this case, we use $\mathcal{O}(\cdot)$ to hide polynomial terms in the problem parameters $(\lambda_0^{\pm 1}, r^{\pm 1}, b_a^{-1}, B_a)$, and $\tilde{\mathcal{O}}(\cdot)$ to hide both poly-logarithmic terms in $(\mathsf{M}_0, \mathsf{C}_0, T, A, d, 1/\varepsilon, 1/\delta_0)$ and polynomial terms in $(\lambda_0^{\pm 1}, r^{\pm 1}, b_a^{-1}, B_a)$.

### F.2  DEFINITIONS AND ASSUMPTIONS

For any actions $\mathbf{a}_{t,1}, \ldots, \mathbf{a}_{t,A} \in \mathbb{R}^d$, we define

$$f_k(\mathbf{a}_{t,1}, \ldots, \mathbf{a}_{t,A}; \mu_t, r\boldsymbol{\Sigma}_t^{-1}) := \log \mathbb{P}_{\tilde{\mathbf{w}}_t \sim \mathcal{N}(\mu_t, r\boldsymbol{\Sigma}_t^{-1})}\Big(k = \arg\max_{j\in[A]} \langle \mathbf{a}_{t,j}, \tilde{\mathbf{w}}_t\rangle\Big).$$

For any $k \in [A]$, $\mathbf{x}_1, \ldots, \mathbf{x}_A \in \mathbb{R}^d$, $y_1, \ldots, y_A \in \mathbb{R}$, we introduce

$$g_k(\mathbf{x}_1, \ldots, \mathbf{x}_A, y_1, \ldots, y_A) := \log \mathbb{P}_{\mathbf{z} \sim \mathcal{N}(0, \mathbf{I}_d)}\Big( \langle \mathbf{x}_k - \mathbf{x}_j, \mathbf{z} \rangle + y_k - y_j \geq 0, \text{ for all } j \in [A]\Big).$$

It can be verified that

$$f_k(\mathbf{a}_1, \ldots, \mathbf{a}_A; \mu_t, r\boldsymbol{\Sigma}_t^{-1}) = g_k(\sqrt{r}\boldsymbol{\Sigma}_t^{-1/2}\mathbf{a}_{t,1}, \ldots, \sqrt{r}\boldsymbol{\Sigma}_t^{-1/2}\mathbf{a}_{t,A}, \langle \mu_t, \mathbf{a}_{t,1} \rangle, \ldots, \langle \mu_t, \mathbf{a}_{t,A} \rangle).$$

For any $\eta_1 \in [0, 1]$, we also define the truncated log-probabilities

$$f_{k,\eta_1}(\mathbf{a}_{t1}, \ldots, \mathbf{a}_{t,A}; \mu_t, r\boldsymbol{\Sigma}_t^{-1}) := \log \Big[ \mathbb{P}\Big( k = \arg\max_{j \in [A]} \langle \mathbf{a}_{t,j}, \tilde{\mathbf{w}}_t \rangle \Big) \vee \eta_1 \Big],$$

$$g_{k,\eta_1}(\mathbf{x}_1, \ldots, \mathbf{x}_A, y_1, \ldots, y_A) := \log \Big[ \mathbb{P}\Big( \langle \mathbf{x}_k - \mathbf{x}_j, \mathbf{z} \rangle + y_k - y_j \geq 0, \text{ for all } j \in [A]\Big) \vee \eta_1 \Big].$$

Define in addition the region $D_\eta := \{\mathbf{x}_1, \ldots, \mathbf{x}_A, y_1, \ldots, y_k : \|\mathbf{x}_i - \mathbf{x}_j\|_2 \geq \eta, \text{ for all } i \neq j\}$. We verify that on the set $D_\eta$, the function $g_{k,\eta_1}$ is Lipschitz continuous in any of its arguments (see Lemma 24 for more).

We adopt the following definition in Bai et al. (2023).

**Definition 22** (Approximability by sum of relus). *A function $g : \mathbb{R}^d \to \mathbb{R}$ is $(\varepsilon, R, \mathsf{M}_0, \mathsf{C}_0)$-approximable by sum of relus, if there exists a "$(\mathsf{M}_0, \mathsf{C}_0)$-sum of relus" function*

$$f_{\mathsf{M}_0, \mathsf{C}_0}(\mathbf{z}) = \sum_{m=1}^{\mathsf{M}_0} c_m \sigma\left( \mathbf{w}_m^\top [\mathbf{z}; 1] \right) \quad with \quad \sum_{m=1}^{\mathsf{M}_0} |c_m| \leq \mathsf{C}_0, \max_{m \in [\mathsf{M}_0]} \|\mathbf{w}_m\|_1 \leq 1, \mathbf{w}_m \in \mathbb{R}^{d+1}, c_m \in \mathbb{R},$$

*such that $\sup_{\mathbf{z} \in [-R, R]^d} |g(\mathbf{z}) - f_{\mathsf{M}_0, \mathsf{C}_0}(\mathbf{z})| \leq \varepsilon$.*

**Assumption B** (Approximation of log-posterior probability). *There exist $\mathsf{M}_0, \mathsf{C}_0 > 0$ depending on $(1/\varepsilon, 1/\eta_1,$ $1/\eta_2, R_\delta, A)$ such that for any $\varepsilon > 0, \eta_1 \in (0, 1), \eta_2 > 0, \delta \in (0, 1/2)$ and $k \in [A]$, $g_{k,\eta_1}(\mathbf{x}_1, \ldots, \mathbf{x}_A, y_1, \ldots, y_A)$ is $(\varepsilon, R_\delta, \mathsf{M}_0, \mathsf{C}_0)$-approximable by sum of relus on $D_{\eta_2}$ with $R_\delta := 2B_a\sqrt{\lambda}(1 + 2\sqrt{\log(2/\delta)} + \sqrt{d}) = \tilde{\mathcal{O}}(\sqrt{\lambda d})$.*

Assumption B states that the (truncated) log-policy of Thompson sampling can be approximated via a two-layer MLP on a compact set with $\tilde{\mathcal{O}}(\sqrt{d})$-radius when $\lambda = \tilde{\mathcal{O}}(1)$ (or with $\tilde{\mathcal{O}}(1)$–radius when $\lambda = \lambda_0/d = \tilde{\mathcal{O}}(1/d)$).

**Assumption C** (Difference between the actions). *There exists some $\eta > 0$ such that for all instances $\mathsf{M}$ and any time $t \in [T]$, we have $\|\mathbf{a}_{t,j} - \mathbf{a}_{t,k}\|_2 \geq \eta$ for all $1 \leq j < k \leq A$.*

With the definitions and assumptions at hand, we now present the formal statement of Theorem 10 as in Theorem 23.

**Theorem 23** (Approximating the Thompson sampling, Formal statement of Theorem 10). *For any $0 < \delta_0 < 1/2$, consider the same embedding mapping $\mathtt{h}$ and extraction mapping $\mathtt{A}$ as for soft LinUCB in E.1, and consider the standard concatenation operator $\mathtt{cat}$. Under Assumption B, C, for $\varepsilon < (\eta \wedge 1)/4$, there exists a transformer $\mathrm{TF}_{\boldsymbol{\theta}}^{\mathsf{R}}(\cdot)$ with $\log \mathsf{R} = \tilde{\mathcal{O}}(1)$,*

$$D = \tilde{\mathcal{O}}(T^{1/4}Ad), \ L = \tilde{\mathcal{O}}(\sqrt{T}), \ M = \tilde{\mathcal{O}}(AT^{1/4}), \ D' = \tilde{\mathcal{O}}(A(T^{1/4}d + \mathsf{M}_0)),$$

$$\|\boldsymbol{\theta}\| \leq \tilde{\mathcal{O}}(T + AT^{1/4} + \sqrt{\mathsf{M}_0 A} + \mathsf{C}_0), \tag{20}$$

*such that with probability at least $1 - \delta_0$ over $(\mathsf{M}, D_T) \sim \mathbb{P}_\Lambda^{\mathsf{Alg}}$ for any $\mathsf{Alg}$, we have*

$$\log \mathsf{Alg}_{\mathrm{TS}}(\mathbf{a}_{t,k}|D_{t-1}, s_t) - \log \mathsf{Alg}_{\boldsymbol{\theta}}(\mathbf{a}_{t,k}|D_{t-1}, s_t) \leq \varepsilon, \quad \text{for all } t \in [T], k \in [A].$$

*Here $\mathsf{M}_0, \mathsf{C}_0$ are the values defined in Assumption B with $\eta_1 = \varepsilon/(4A), \eta_2 = \eta, \delta = \delta_0$, and $\tilde{\mathcal{O}}(\cdot)$ hides polynomial terms in $(\lambda^{\pm 1}, r^{\pm 1}, b_a^{-1}, B_a)$ and poly-logarithmic terms in $(\mathsf{M}_0, \mathsf{C}_0, T, A, d, 1/\delta_0, 1/\varepsilon)$.*

### F.3 Proof of Theorem 23 (and hence Theorem 10)

We construct a transformer implementing the following steps at each time $t \in [T]$ starting with $\mathbf{h}_{2t-1}^{\star} = \mathbf{h}_{2t-1}^{\text{pre},\star}$ for $\star \in \{a, b, c, d\}$

$$\mathbf{h}_{2t-1} = \begin{bmatrix} \mathbf{h}_{2t-1}^{\text{pre},a} \\ \mathbf{h}_{2t-1}^{\text{pre},b} \\ \mathbf{h}_{2t-1}^{\text{pre},c} \\ \mathbf{h}_{2t-1}^{\text{pre},d} \end{bmatrix} \xrightarrow{\text{step 1}} \begin{bmatrix} \mathbf{h}_{2t-1}^{\text{pre},\{a,b,c\}} \\ \widehat{\mu}_t \\ \star \\ \mathbf{0} \\ \mathbf{pos} \end{bmatrix} \xrightarrow{\text{step 2}} \begin{bmatrix} \mathbf{h}_{2t-1}^{\text{pre},\{a,b,c\}} \\ \widehat{\mu}_t \\ \widehat{\boldsymbol{\Sigma}_t^{1/2} \mathbf{a}_{t,1}} \\ \vdots \\ \widehat{\boldsymbol{\Sigma}_t^{1/2} \mathbf{a}_{t,A}} \\ \star \\ \mathbf{0} \\ \mathbf{pos} \end{bmatrix} \xrightarrow{\text{step 3}} \begin{bmatrix} \mathbf{h}_{2t-1}^{\text{pre},\{a,b,c\}} \\ \widehat{\mu}_t \\ \sqrt{r}\widehat{\boldsymbol{\Sigma}_t^{-1/2} \mathbf{a}_{t,1}} \\ \vdots \\ \sqrt{r}\widehat{\boldsymbol{\Sigma}_t^{-1/2} \mathbf{a}_{t,A}} \\ \langle \widehat{\mu}_t, \mathbf{a}_{t,1} \rangle \\ \vdots \\ \langle \widehat{\mu}_t, \mathbf{a}_{t,A} \rangle \\ \star \\ \mathbf{0} \\ \mathbf{pos} \end{bmatrix} \tag{21}$$

$$\xrightarrow{\text{step 4}} \begin{bmatrix} \mathbf{h}_{2t-1}^{\text{pre},\{a,b\}} \\ \widehat{v}_{t1} \\ \vdots \\ \widehat{v}_{tA} \\ \mathbf{h}_{2t-1}^d \end{bmatrix} =: \begin{bmatrix} \mathbf{h}_{2t-1}^{\text{post},a} \\ \mathbf{h}_{2t-1}^{\text{post},b} \\ \mathbf{h}_{2t-1}^{\text{post},c} \\ \mathbf{h}_{2t-1}^{\text{post},d} \end{bmatrix},$$

where $\mathbf{pos} := [t, t^2, 1]^\top$; $\mu_t, \boldsymbol{\Sigma}_t$ are the mean and covariance of the distribution we sample $\tilde{\mathbf{w}}$ from; $\widehat{v}_{tk}$ are approximations to $v_{tk}^* := \log \mathbb{P}(j = \arg\max_{k \in [A]} \langle \mathbf{a}_{t,k}, \tilde{\mathbf{w}}_t \rangle)$. In addition, we use $\mathbf{h}^\star, \star \in \{a, b, c, d\}$ to denote the corresponding parts of a token vector $\mathbf{h}$. After passing through the transformer, we obtain the policy

$$\mathsf{Alg}_{\boldsymbol{\theta}}(\cdot | D_{t-1}, s_t) := \frac{\exp(\mathbf{h}_{2t-1}^{\text{post},c})}{\| \exp(\mathbf{h}_{2t-1}^{\text{post},c}) \|_1} = \frac{\exp(\widehat{\mathbf{v}}_t)}{\| \exp(\widehat{\mathbf{v}}_t) \|_1} \in \Delta^A.$$

In step 1—3 of (21), we use transformer to approximately generate the arguments

$$(\sqrt{r}\boldsymbol{\Sigma}_t^{-1/2} \mathbf{a}_{t,1}, \dots, \sqrt{r}\boldsymbol{\Sigma}_t^{-1/2} \mathbf{a}_{t,A}, \langle \mu_t, \mathbf{a}_{t,1} \rangle, \dots, \langle \mu_t, \mathbf{a}_{t,A} \rangle)$$

of the function $g_k$ (or $g_{k,\eta_2}$), and in step 4 of (21), we use transformer to approximate the truncated log-probability $g_{k,\eta_1}$ for some $\eta_1 \in (0, 1)$ by exploiting Assumption B, C.

For any $0 < \delta_0 < 1/2$, define $B_w := \sqrt{\lambda}(\sqrt{d} + 2\sqrt{\log(2/\delta_0)})$ and the event

$$\mathcal{E}_{\delta_0} := \{\max_{t \in [T]} |\varepsilon_t| \leq \sqrt{2r \log(2T/\delta_0)}\} \cup \{\|\mathbf{w}^*\|_2 \leq B_w\}.$$

Then by a standard tail bound for gaussian variables $\{\varepsilon_t\}_{t=1}^T$, a union bound over $t \in [T]$, and Eq. (4.3) in Laurent & Massart (2000), we have

$$\mathbb{P}(\mathcal{E}_{\delta_0}) \geq 1 - \delta_0.$$

We claim the following results which we will prove later.

**Step 1** Under the high probability event $\mathcal{E}_{\delta_0}$, for any $\varepsilon > 0$, there exists a transformer $\text{TF}_{\boldsymbol{\theta}}(\cdot)$ with

$$L = \left\lceil 2\sqrt{2T}\sqrt{\frac{B_a^2 + \widetilde{\lambda}}{\widetilde{\lambda}}} \log\left(\frac{(2T(B_a^2 + \widetilde{\lambda}) + \widetilde{\lambda})TB_a(B_aB_w + \sqrt{2r\log(2T/\delta_0)})}{\widetilde{\lambda}^2\varepsilon}\right)\right\rceil = \tilde{\mathcal{O}}(\sqrt{T}),$$

$$\max_{\ell \in [L]} M^{(l)} \leq 4, \quad \max_{\ell \in [L]} D'^{(\ell)} \leq 4d, \quad \|\boldsymbol{\theta}\| \leq 10 + (\tilde{\lambda} + 2)/(B_a^2 + \tilde{\lambda}),$$

where $\widetilde{\lambda} := r/\lambda$ that implements step 1 in (21) with $\|\widehat{\mu}_t - \mu_t\|_2 \leq \varepsilon$.

**Step 2** For any $\varepsilon > 0$, there exists s transformer $\mathrm{TF}_{\boldsymbol{\theta}}(\cdot)$ with

$$L = \tilde{\mathcal{O}}(\sqrt{T}), \quad \max_{\ell \in [L]} M^{(\ell)} = \tilde{\mathcal{O}}(AT^{1/4}), \quad \max_{\ell \in [L]} D'^{(\ell)} = \tilde{\mathcal{O}}(T^{1/4}Ad), \quad \|\boldsymbol{\theta}\| \le \tilde{\mathcal{O}}(T+AT^{1/4})$$

that implements step 2 in (21) such that $\|\widehat{\boldsymbol{\Sigma}_t^{1/2}\mathbf{a}_{t,k}} - \boldsymbol{\Sigma}_t^{1/2}\mathbf{a}_{t,k}\|_2 \le \varepsilon$ for all $k \in [A]$.

**Step 3** Under the high probability event $\mathcal{E}_{\delta_0}$, for any $\varepsilon > 0$, assume Step 1, 2 above are implemented with the approximation error less than $\varepsilon/B_a, \varepsilon\tilde{\lambda}/\sqrt{4r}$ respectively, then there exists a transformer $\mathrm{TF}_{\boldsymbol{\theta}}(\cdot)$ with

$$L = \lceil 2 + 2\sqrt{2T(B_a^2 + \tilde{\lambda})/\tilde{\lambda}} \log((1+\tilde{\kappa})4\sqrt{r}\sqrt{T(B_a^2+\tilde{\lambda})}B_a/\varepsilon) \rceil = \tilde{\mathcal{O}}(\sqrt{T}), \quad \max_{\ell \in [L]} M^{(l)} = 4A,$$

$$\max_{\ell \in [L]} D'^{(\ell)} \le 4Ad, \quad \|\boldsymbol{\theta}\| \le \tilde{\mathcal{O}}(T+A)$$

that implements step 3 in (21) with $\|\sqrt{r}\widehat{\boldsymbol{\Sigma}_t^{-1/2}\mathbf{a}_{t,k}} - \sqrt{r}\boldsymbol{\Sigma}_t^{-1/2}\mathbf{a}_{t,k}\|_2 \le \varepsilon, |\langle \widehat{\mu}_t, \mathbf{a}_{t,k} \rangle - \langle \mu_t, \mathbf{a}_{t,k} \rangle| \le \varepsilon$ for all $k \in [A]$.

**Step 4** Under Assumption B, C and the high probability event $\mathcal{E}_{\delta_0}$, suppose the approximation error $\varepsilon_3$ in Step 3 satisfies $\varepsilon_3 \le R_{\delta_0}/2 = \tilde{\mathcal{O}}(\sqrt{\lambda d})$, and suppose the vector

$$\mathbf{v} := (\sqrt{r}\widehat{\boldsymbol{\Sigma}_t^{-1/2}\mathbf{a}}_{t,1}, \ldots \sqrt{r}\widehat{\boldsymbol{\Sigma}_t^{-1/2}\mathbf{a}}_{t,A}, \langle \widehat{\mu}_t, \mathbf{a}_{t,1} \rangle \ldots \langle \widehat{\mu}_t, \mathbf{a}_{t,A} \rangle)$$

lies in $D_{\eta/2}$, for any $\varepsilon > 0$ there exists an MLP-only transformer $\mathrm{TF}_{\boldsymbol{\theta}}(\cdot)$ with

$$L = 1, \quad D' = \mathsf{M}_0 A, \quad \|\mathbf{W}_1\|_{\mathrm{op}} \le \sqrt{\mathsf{M}_0 A}, \quad \|\mathbf{W}_2\|_{\mathrm{op}} \le \mathsf{C}_0$$

that implements step 4 in (21) such that $|\widehat{v}_{tk} - g_{k,\eta_1}(\mathbf{v})| \le \varepsilon$ for all $k \in [A]$ amd $\eta_1 = c\varepsilon/A$ for some universal constant $c > 0$.[1]

To complete the proof, we in addition present the following lemma.

**Lemma 24.** *For any $\eta_1 \in (0,1), \eta > 0$, $g_{k,\eta_1}(\mathbf{x}_1, \ldots, \mathbf{x}_A, y_1, \ldots, y_A)$ is $1/2$-Holder continuous in its arguments on $D_\eta$, namely,*

$$|g_{k,\eta_1}(\mathbf{x}_1, \ldots, \mathbf{x}_j, \ldots, \mathbf{x}_A, y_1, \ldots, y_A) - g_{k,\eta_1}(\mathbf{x}_1, \ldots, \mathbf{x}_j', \ldots \mathbf{x}_A, y_1, \ldots, y_A)|$$

$$\le \frac{2A}{\eta_1}\left(\sqrt{\frac{2\|\mathbf{x}_j - \mathbf{x}_j'\|_2}{\eta}} + \frac{2\|\mathbf{x}_j - \mathbf{x}_j'\|_2}{\eta}\right),$$

$$|g_{k,\eta_1}(\mathbf{x}_1, \ldots, \mathbf{x}_A, y_1, \ldots, y_j, \ldots, y_A) - g_{k,\eta_1}(\mathbf{x}_1, \ldots \mathbf{x}_A, y_1, \ldots, y_j', \ldots, y_A)|$$

$$\le \frac{2A|y_j - y_j'|}{\eta\eta_1}$$

*for any*

$$(\mathbf{x}_1, \ldots, \mathbf{x}_j, \ldots, \mathbf{x}_A, y_1, \ldots, y_A), \ (\mathbf{x}_1, \ldots, \mathbf{x}_j', \ldots, \mathbf{x}_A, y_1, \ldots, y_A) \in D_\eta,$$

$$(\mathbf{x}_1, \ldots, \mathbf{x}_A, y_1, \ldots, y_j, \ldots, y_A), \ (\mathbf{x}_1, \ldots, \mathbf{x}_A, y_1, \ldots, y_j', \ldots, y_A) \in D_\eta$$

*for all $k, j \in [A]$.*

See the proof in Section F.5.

Now, we complete the proof by combining Step 1— 4 and using Lemma 24.

Let $\varepsilon_1, \varepsilon_2, \varepsilon_3, \varepsilon_4$ denote the approximation errors $\varepsilon$ appearing in Step 1, 2, 3, 4, respectively. W.l.o.g., we assume $\varepsilon_1, \varepsilon_2, \varepsilon_3, \varepsilon_4 < 1/4 \wedge \eta/4$. Define the vector

$$\mathbf{v}^* := (\sqrt{r}\boldsymbol{\Sigma}_t^{-1/2}\mathbf{a}_{t,1}, \ldots \sqrt{r}\boldsymbol{\Sigma}_t^{-1/2}\mathbf{a}_{t,A}, \langle \mu_t, \mathbf{a}_{t,1} \rangle \ldots \langle \mu_t, \mathbf{a}_{t,A} \rangle).$$

---

[1]Note $\mathsf{M}_0, \mathsf{C}_0$ in the formula implicitly depend on $1/\varepsilon$.

By Assumption C and a triangular inequality, we have $\mathbf{v}, \mathbf{v}^* \in D_{\eta/2}$. By the Lipschitz continuity of $f(x) = \exp(x)$ on $(-\infty, 1.5]$, we have

$$
\begin{aligned}
|\exp(\widehat{v}_{tk}) - \mathsf{Alg}_{\mathrm{TS}}(\mathbf{a}_{t,k}|D_{t-1}, s_t)| &\leq |\exp(\widehat{v}_{t,k}) - \mathsf{Alg}_{\mathrm{TS}}(\mathbf{a}_{t,k}|D_{t-1}, s_t) \vee \eta_1| + \eta_1 \\
&\leq e^{3/2}(|\widehat{v}_{tk} - g_{k,\eta_1}(\mathbf{v})| + |g_{k,\eta_1}(\mathbf{v}) - g_{k,\eta_1}(\mathbf{v}^*)|) + \eta_1, \\
&\leq e^{3/2}\Big(\varepsilon_4 + \frac{2A^2}{\eta_1}\Big(\sqrt{\frac{2\varepsilon_3}{\eta}} + \frac{2\varepsilon_3}{\eta}\Big) + \frac{2A^2\varepsilon_3}{\eta\eta_1}\Big) + \eta_1 =: \varepsilon_5,
\end{aligned}
$$

where the second inequality uses

$$
g_{k,\eta_1}(\mathbf{v}^*) = \log[\mathsf{Alg}_{\mathrm{TS}}(\mathbf{a}_{t,k}|D_{t-1}, s_t) \vee \eta_1],
$$

and the third inequality uses Lemma 24 and Step 4. Therefore,

$$
|\sum_{k=1}^{A} \exp(\widehat{v}_{t,k}) - 1| \leq A\varepsilon_5.
$$

and the constructed transformer $\mathrm{TF}_{\boldsymbol{\theta}}$ satisfies (assume $A\varepsilon_5 < 1$)

$$
\log \mathsf{Alg}_{\mathrm{TS}}(\mathbf{a}_{t,k}|D_{t-1}, s_t) - \log \mathsf{Alg}_{\boldsymbol{\theta}}(\mathbf{a}_{t,k}|D_{t-1}, s_t)
$$

$$
\leq (\log[\mathsf{Alg}_{\mathrm{TS}}(\mathbf{a}_{t,k}|D_{t-1}, s_t) \vee \eta_1] - \widehat{v}_{t,k}) + \log(\sum_{k=1}^{A} \exp(\widehat{v}_{t,k}))
$$

$$
\begin{aligned}
&\leq |\widehat{v}_{tk} - g_{k,\eta_1}(\mathbf{v})| + |g_{k,\eta_1}(\mathbf{v}) - g_{k,\eta_1}(\mathbf{v}^*)| + A\varepsilon_5 \\
&\leq (A+1)\varepsilon_5,
\end{aligned}
$$

where third line uses $\log(1+x) < x$. Finally, for the prespecified $\varepsilon > 0$, choosing $\varepsilon_1, \varepsilon_2, \varepsilon_3, \varepsilon_4, \eta_1$ such that $\varepsilon_5 \leq \varepsilon/(2A)$ gives

$$
\log \mathsf{Alg}_{\mathrm{TS}}(\mathbf{a}_{t,k}|D_{t-1}, s_t) - \log \mathsf{Alg}_{\boldsymbol{\theta}}(\mathbf{a}_{t,k}|D_{t-1}, s_t) \leq \varepsilon.
$$

This can be done via choosing $\varepsilon_1 = c_1\varepsilon^4, \varepsilon_2 = c_2\varepsilon^4, \varepsilon_3 = c_3\varepsilon^4, \varepsilon_4 = c_4\varepsilon, \eta_1 = \varepsilon/(4A)$, where $c_i$ ($i = 1, 2, 3, 4$) hide values that could depend polynomially on $(A, 1/\eta)$, such that $\varepsilon_5 \leq \varepsilon/(4A)$.

Combining the construction in Step 1— 4 yields Theorem 23.

Similar to the proof of Theorem 8, from the proof of each step, we can verify that the token dimension $D$ can be chosen to be of order $\tilde{\mathcal{O}}(T^{1/4}Ad)$ (see the proof of Step 2b for details). Moreover, due to the convergence guarantee for each iteration of AGD in Proposition 19, we can be verified that there exists some sufficiently large value $\mathsf{R} > 0$ with $\log \mathsf{R} = \tilde{\mathcal{O}}(1)$ such that we have $\|\mathbf{h}_i^{(\ell)}\|_2 \leq \mathsf{R}$ for all layer $\ell \in [L]$ and all token $i \in [2T]$ in our TF construction. Therefore, $\mathrm{TF}_{\boldsymbol{\theta}}^{\mathsf{R}}$ and $\mathrm{TF}_{\boldsymbol{\theta}}^{\infty}$ yield identical outputs for all token matrices considered, and hence we do not distinguish them in the proof of each step.

**Proof of Step 1** Note that $\mu_t$ is a ridge estimator of $\mathbf{w}_*$ with parameter $\widetilde{\lambda} = r/\lambda$ and the noise $\sup_t |\varepsilon_t| \leq \sqrt{2r\log(T/\delta_0)}$. Step 1 follows immediately from the second part of Lemma 21.

**Proof of Step 2** By the boundedness assumption of the actions, we have

$$
\tilde{\lambda} \leq \sigma_{\min}(\boldsymbol{\Sigma}_t) \leq \sigma_{\max}(\boldsymbol{\Sigma}_t) \leq T(B_a^2 + \tilde{\lambda}).
$$

Define the condition number $\tilde{\kappa} = T(B_a^2 + \tilde{\lambda})/\tilde{\lambda}$ and $\mu := \sqrt{T\tilde{\lambda}(B_a^2 + \tilde{\lambda})}$. Using the Pade decomposition for the square root function in Theorem 3.1 and the discussion afterward in Lu (1998), we have

$$
\begin{aligned}
\boldsymbol{\Sigma}_t^{1/2} &= (\tilde{\lambda}\mathbf{I}_d + \sum_{j=1}^{t-1} \mathbf{a}_j\mathbf{a}_j^{\top})^{1/2} = \sqrt{\mu}\Big(\mathbf{I}_d + \frac{(\boldsymbol{\Sigma}_t - \mu\mathbf{I}_d)}{\mu}\Big)^{1/2} \\
&= \sqrt{\mu}\Big[\mathbf{I}_d + \sum_{j=1}^{m}\Big(\mathbf{I}_d + \frac{b_j^{(m)}(\boldsymbol{\Sigma}_t - \mu\mathbf{I}_d)}{\mu}\Big)^{-1} \frac{a_j^{(m)}(\boldsymbol{\Sigma}_t - \mu\mathbf{I}_d)}{\mu}\Big] + \mathbf{E}_m
\end{aligned}
$$

for any $m \geq 0$, where

$$a_j^{(m)} = \frac{2}{2m+1} \sin^2 \frac{j\pi}{2m+1}, \quad b_j^{(m)} = \cos^2 \frac{j\pi}{2m+1},$$

and the error term $\mathbf{E}_m$ satisfies

$$\|\mathbf{E}_m\|_{\mathrm{op}}$$
$$\leq \max \left\{ 2\sqrt{T(B_a^2 + \tilde{\lambda})} \left[ 1 + \left( \frac{\sqrt{T(B_a^2 + \tilde{\lambda})} + \sqrt{\mu}}{\sqrt{T(B_a^2 + \tilde{\lambda})} - \sqrt{\mu}} \right)^{2m+1} \right]^{-1}, \quad 2\sqrt{\tilde{\lambda}} \left[ \left( \frac{\sqrt{\mu} + \sqrt{\tilde{\lambda}}}{\sqrt{\mu} - \sqrt{\tilde{\lambda}}} \right)^{2m+1} - 1 \right]^{-1} \right\}$$

$$= \max \left\{ 2\sqrt{T(B_a^2 + \tilde{\lambda})} [1 + (\frac{\tilde{\kappa}^{1/4} + 1}{\tilde{\kappa}^{1/4} - 1})^{2m+1}]^{-1}, 2\sqrt{\tilde{\lambda}} [(\frac{\tilde{\kappa}^{1/4} + 1}{\tilde{\kappa}^{1/4} - 1})^{2m+1} - 1]^{-1} \right\}$$

$$\leq 2 \max \left\{ \sqrt{T(B_a^2 + \tilde{\lambda})} (1 + \frac{2}{\tilde{\kappa}^{1/4}})^{-2m-1}, \sqrt{\tilde{\lambda}} \left[ (1 + \frac{2}{\tilde{\kappa}^{1/4}})^{2m+1} - 1 \right]^{-1} \right\}.$$

Since $(1 + 2/\tilde{\kappa}^{1/4})^{\tilde{\kappa}^{1/4}/2 + 1} > e$, it follows that choosing

$$m = \left( \frac{\tilde{\kappa}^{1/4}}{4} + 1 \right) \max \left\{ \left\lceil \log \left( \frac{2\sqrt{T(B_a^2 + \tilde{\lambda})}}{\varepsilon} \right) \right\rceil, \left\lceil \log \left( \frac{2\sqrt{\tilde{\lambda}}}{\varepsilon} + 1 \right) \right\rceil \right\} = \tilde{\mathcal{O}}(T^{1/4}).$$

gives $\|\mathbf{E}_m\|_{\mathrm{op}} \leq \varepsilon$ for any $0 < \varepsilon < 1$.

Thus, using Pade decomposition, we can write

$$\mathbf{\Sigma}_t^{1/2} \mathbf{a}_{t,k} = \sqrt{\mu} \left[ \mathbf{a}_{t,k} + \sum_{j=1}^m \left( \mu \mathbf{I}_d + b_j^{(m)} (\mathbf{\Sigma}_t - \mu \mathbf{I}_d) \right)^{-1} a_j^{(m)} (\mathbf{\Sigma}_t - \mu \mathbf{I}_d) \mathbf{a}_{t,k} \right] + \mathbf{E}_m^k$$

with $\left\| \mathbf{E}_m^k \right\|_2 \leq \varepsilon$ for all $k \in [A]$ and some $m = \tilde{\mathcal{O}}(T^{1/4})$. Next, we show that there exists a transformer that can implement the following intermediate steps that give Step 2.

$$\begin{bmatrix} \mathbf{h}_{2t-1}^{\mathrm{pre},\{a,b,c\}} \\ \widehat{\mu}_t \\ \star \\ \mathbf{0} \\ \mathbf{pos} \end{bmatrix} \xrightarrow{\text{step 2a}} \begin{bmatrix} \mathbf{h}_{2t-1}^{\mathrm{pre},\{a,b,c\}} \\ \widehat{\mu}_t \\ \mathbf{0}_{dA} \\ (\mathbf{\Sigma}_t - \mu \mathbf{I}_d) \mathbf{a}_{t,1} \\ \vdots \\ (\mathbf{\Sigma}_t - \mu \mathbf{I}_d) \mathbf{a}_{t,A} \\ \star \\ \mathbf{0} \\ \mathbf{pos} \end{bmatrix}$$

$$\xrightarrow{\text{step 2b}} \begin{bmatrix} \mathbf{h}_{2t-1}^{\mathrm{pre},\{a,b,c\}} \\ \widehat{\mu}_t \\ \mathbf{0}_{dA} \\ (\mathbf{\Sigma}_t - \mu \mathbf{I}_d) \mathbf{a}_{t,1} \\ \vdots \\ (\mathbf{\Sigma}_t - \mu \mathbf{I}_d) \mathbf{a}_{t,A} \\ \left( \mu \mathbf{I}_d + b_1^{(m)} (\mathbf{\Sigma}_t - \mu \mathbf{I}_d) \right)^{-1} a_1^{(m)} (\mathbf{\Sigma}_t - \mu \mathbf{I}_d) \mathbf{a}_{t,1} \\ \vdots \\ \left( \mu \mathbf{I}_d + b_m^{(m)} (\mathbf{\Sigma}_t - \mu \mathbf{I}_d) \right)^{-1} a_m^{(m)} (\mathbf{\Sigma}_t - \mu \mathbf{I}_d) \mathbf{a}_{t,A} \\ \star \\ \mathbf{0} \\ \mathbf{pos} \end{bmatrix} \xrightarrow{\text{step 2c}} \begin{bmatrix} \mathbf{h}_{2t-1}^{\mathrm{pre},\{a,b,c\}} \\ \widehat{\mu}_t \\ \widehat{\mathbf{\Sigma}_t^{1/2} \mathbf{a}_{t,1}} \\ \vdots \\ \widehat{\mathbf{\Sigma}_t^{1/2} \mathbf{a}_{t,A}} \\ \vdots \\ \star \\ \mathbf{0} \\ \mathbf{pos} \end{bmatrix}, \quad (22)$$

where $\star$ denotes additional terms in $\mathbf{h}^d$ that are not of concern to our analysis.

**Step 2a** There exists an attention-only transformer $\mathrm{TF}_{\boldsymbol{\theta}}(\cdot)$ with

$$L = 2, \quad \max M^{(\ell)} = 3A, \quad \|\boldsymbol{\theta}\| \le T + 2 + \mu \le \mathcal{O}(T)$$

that implements step 2a in (22).

**Step 2b** Denote $(\boldsymbol{\Sigma}_t - \mu \mathbf{I}_d)\mathbf{a}_{t,k}$ by $\mathbf{q}_k$ and $\left(\mu \mathbf{I}_d + b_m^{(m)}(\boldsymbol{\Sigma}_t - \mu \mathbf{I}_d)\right)$ by $\mathbf{M}_m$. For any $\varepsilon > 0$, there exists a transformer $\mathrm{TF}_{\boldsymbol{\theta}}(\cdot)$ with

$$L = 2\sqrt{2T(B_a^2 + \tilde{\lambda})/\tilde{\lambda}} \log((1 + \tilde{\kappa})B_a T(B_a^2 + \tilde{\lambda})/(\tilde{\lambda}\varepsilon))\rceil = \tilde{\mathcal{O}}(\sqrt{T}),$$

$$\max_{\ell \in [L]} M^{(l)} = 4Am = \tilde{\mathcal{O}}(T^{1/4}A), \ \max_{\ell \in [L]} D'^{(\ell)} \le \mathrm{O}(Adm) = \tilde{\mathcal{O}}(T^{1/4}Ad), \ \|\boldsymbol{\theta}\| \le \mathcal{O}(Am) \le \tilde{\mathcal{O}}(T^{1/4}A)$$

approximately implements step 2b in (22) such that the output component $\widehat{\mathbf{M}_m^{-1}\mathbf{q}_k}$ satisfies
$\|a_j^{(m)}\widehat{\mathbf{M}_m^{-1}\mathbf{q}_k} - a_j^{(m)}\mathbf{M}_j^{-1}\mathbf{q}_k\|_2 \le \varepsilon$ for all $j \in [m]$ and $k \in [A]$.

**Step 2c** There exists an MLP-only transformer $\mathrm{TF}_{\boldsymbol{\theta}}(\cdot)$ with

$$L = 1, D' = 2Ad(m+1) = \tilde{\mathcal{O}}(T^{1/4}Ad), \quad \|\mathbf{W}_1\|_{\mathrm{op}} = \sqrt{2}, \quad \|\mathbf{W}_2\|_{\mathrm{op}} \le \sqrt{\mu}(1+m) = \tilde{\mathcal{O}}(T^{3/4})$$

that implements step 2c in (22).

Combining the intermediate steps with the approximation error in Step 2b chosen as $\varepsilon/m$ gives Step 2 as desired.

**Proof of Step 2a** For all $k \in [A]$, we choose $\mathbf{Q}_{k1,k2,k3}^{(1)}, \mathbf{K}_{k1,k2,k3}^{(1)}, \mathbf{V}_{k1,k2,k3}^{(1)}$ such that for even token indices $2j$ with $j \le t - 1$ and odd token indices with $j \le t$

$$\mathbf{Q}_{k1}^{(1)}\mathbf{h}_{2t-1}^{(0)} = \begin{bmatrix} \mathbf{a}_{t,k} \\ \mathbf{0} \end{bmatrix}, \quad \mathbf{K}_{k1}^{(1)}\mathbf{h}_{2j}^{(0)} = \begin{bmatrix} \mathbf{a}_j \\ \mathbf{0} \end{bmatrix}, \quad \mathbf{V}_{k1}^{(1)}\mathbf{h}_{2j}^{(0)} = \begin{bmatrix} \mathbf{0} \\ \mathbf{a}_j \\ \mathbf{0} \end{bmatrix}, \quad \mathbf{K}_{k1}^{(1)}\mathbf{h}_{2j-1}^{(0)} = [\mathbf{0}], \quad \mathbf{V}_{k1}^{(1)}\mathbf{h}_{2j-1}^{(0)} = [\mathbf{0}]$$

$$\mathbf{Q}_{k2}^{(1)} = -\mathbf{Q}_{k1}^{(1)}, \quad \mathbf{K}_{k2}^{(1)} = \mathbf{K}_{k1}^{(1)}, \quad \mathbf{V}_{k2}^{(1)} = -\mathbf{V}_{k1}^{(1)},$$

$$\mathbf{Q}_{k3}^{(1)}\mathbf{h}_{2t-1}^{(0)} = \begin{bmatrix} 1 \\ -(2t-1) \\ 1 \\ \mathbf{0} \end{bmatrix}, \quad \mathbf{K}_{k3}^{(1)}\mathbf{h}_{2j-1}^{(0)} = \begin{bmatrix} 1 \\ 1 \\ 2j-1 \\ \mathbf{0} \end{bmatrix}, \quad \mathbf{K}_{k3}^{(1)}\mathbf{h}_{2j}^{(0)} = \begin{bmatrix} 1 \\ 1 \\ 2j \\ \mathbf{0} \end{bmatrix}, \quad \mathbf{V}_{k3}^{(1)}\mathbf{h}_{2t-1}^{(0)} = \begin{bmatrix} \mathbf{0} \\ (\tilde{\lambda} - \mu)\mathbf{a}_{t,k} \\ \mathbf{0} \end{bmatrix},$$

where for each $k \in [A]$, $\mathbf{V}_{k1}^{(1)}\mathbf{h}_{2j}^{(0)}, \mathbf{V}_{k3}^{(1)}\mathbf{h}_{2t-1}^{(0)}$ are supported on the same $d$ entries of $\mathbf{h}^d$. It is readily verified that summing over the attention heads and $k \in [A]$ gives the updates

$$\mathbf{0}_d \mapsto \frac{(\boldsymbol{\Sigma}_t - \mu \mathbf{I}_d)\mathbf{a}_{t,k}}{2t - 1}$$

for all $k \in [A]$. We assume the updated vectors are supported on some $Ad$ coordinates of $\mathbf{h}_{2t-1}^d$. Moreover, one can choose the matrices such that $\|\mathbf{Q}_{k1,k2,k3}^{(1)}\|_{\mathrm{op}} \le 1, \|\mathbf{K}_{k1,k2,k3}^{(1)}\|_{\mathrm{op}} \le 1, \|\mathbf{V}_{k1,k2,k3}^{(1)}\|_{\mathrm{op}} \le \max\{1, |\tilde{\lambda} - \mu|\} \le 1 + \mu$. Therefore the norm of the first layer of the attention-only transformer $\|\boldsymbol{\theta}^{(1)}\| \le 2 + \mu$.

The second layer is used to multiply the updated vectors by a factor of $2t - 1$, namely, to perform the map

$$\frac{(\boldsymbol{\Sigma}_t - \mu \mathbf{I}_d)\mathbf{a}_{t,k}}{2t - 1} \mapsto (\boldsymbol{\Sigma}_t - \mu \mathbf{I}_d)\mathbf{a}_{t,k}$$

for all $k \in [A]$, where the output vectors are supported on coordinates different from the input vectors (therefore we need $2Ad$ coordinates for embedding in step 2a). This can be achieved by choosing $\|\mathbf{Q}^{(2)}\|_{\mathrm{op}} \le T, \|\mathbf{K}^{(2)}\|_{\mathrm{op}} \le T, \|\mathbf{V}^{(2)}\|_{\mathrm{op}} \le 1$ such that

$$\mathbf{Q}_1^{(2)}\mathbf{h}_{2t-1}^{(1)} = \begin{bmatrix} (2t-1)^2 \\ -T(2t-1)^2 \\ 1 \\ \mathbf{0} \end{bmatrix}, \quad \mathbf{K}_1^{(2)}\mathbf{h}_{2j-1}^{(1)} = \begin{bmatrix} 1 \\ 1 \\ T(2j-1)^2 \\ \mathbf{0} \end{bmatrix}, \quad \mathbf{K}_{k3}^{(1)}\mathbf{h}_{2j}^{(1)} = \begin{bmatrix} 1 \\ 1 \\ T(2j)^2 \\ \mathbf{0} \end{bmatrix},$$

$$
\mathbf{V}_1^{(2)}\mathbf{h}_{2t-1}^{(1)} = \begin{bmatrix} \mathbf{0} \\ \frac{(\boldsymbol{\Sigma}_t - \mu\mathbf{I}_d)\mathbf{a}_{t,1}}{(2t-1)} \\ \vdots \\ \frac{(\boldsymbol{\Sigma}_t - \mu\mathbf{I}_d)\mathbf{a}_{t,A}}{(2t-1)} \\ \mathbf{0} \end{bmatrix}.
$$

Therefore $\|\!|\boldsymbol{\theta}^{(2)}|\!\| \leq T + 1$ and hence the two layer transformer we constructed has norm $\|\!|\boldsymbol{\theta}|\!\| \leq T + 2 + \mu$.

**Proof of step 2b** The construction is similar to the construction in Step 2 of the proof of Theorem 8. Hence we only provide a sketch of proof here. Note that

$$
a_j^{(m)}\mathbf{M}_j^{-1}\mathbf{q}_k = \arg\min_{\mathbf{x}\in\mathbb{R}^d} \frac{1}{2(2t-1)}\mathbf{x}^\top\mathbf{M}_j\mathbf{x} - \frac{1}{2t-1}\left\langle \mathbf{x}, a_j^{(m)}\mathbf{q}_k \right\rangle =: \arg\min_{\mathbf{x}\in\mathbb{R}^d} L_{k,j}(\mathbf{x})
$$

is the global minimizer of a $\tilde{\lambda}/(2t-1)$-convex and $(B_a^2 + \tilde{\lambda})$-smooth function with the conditional number $\tilde{\kappa} \leq 2T(B_a^2 + \tilde{\lambda})/\tilde{\lambda}$. Since

$$
\|a_j^{(m)}\mathbf{M}_j^{-1}\mathbf{q}_k\|_2 \leq |a_j^{(m)}|\|\mathbf{M}_j^{-1}\|_{\mathrm{op}}\|\mathbf{q}_k\|_2 \leq B_a T(B_a^2 + \tilde{\lambda})/\tilde{\lambda}.
$$

Therefore by Proposition 19 we have $L = \lceil 2\sqrt{2T(B_a^2 + \tilde{\lambda})/\tilde{\lambda}} \log((1 + \tilde{\kappa})B_a T(B_a^2 + \tilde{\lambda})/(\tilde{\lambda}\varepsilon))\rceil$ steps of accelerated gradient descent with step size $\eta = 1/(B_a^2 + \tilde{\lambda})$ gives $\|a_j^{(m)}\widehat{\mathbf{M}_j^{-1}}\mathbf{q}_k - a_j^{(m)}\mathbf{M}_j^{-1}\mathbf{q}_k\|_2 \leq \varepsilon$. Now, it remains to construct a transformer that can implement the accelerated gradient descent steps. Here we only provide the construction of the gradient $\nabla L_{k,j}(\mathbf{x})$ at the $l$-th iteration $\mathbf{x} = \mathbf{x}_{k,j}^{\ell-1} \in \mathbb{R}^d$ which belongs to the output after $\ell - 1$ transformer layers. The full construction of AGD steps follows from similar techniques as in Step 2 of the proof of Theorem 8. Concretely, for each layer $\ell \in [L]$ and $k \in [A], j \in [m]$, we choose $\mathbf{Q}_{kj1,kj2,kj3}^{(\ell)}, \mathbf{K}_{kj1,kj2,kj3}^{(\ell)}, \mathbf{V}_{kj1,kj2,kj3}^{(\ell)}$ such that for even token indices $2j$ with $s \leq t - 1$ and odd token indices with $s \leq t$

$$
\mathbf{Q}_{kj1}^{(\ell)}\mathbf{h}_{2t-1}^{(\ell-1)} = \begin{bmatrix} \mathbf{x}_{k,j}^{\ell-1} \\ \mathbf{0} \end{bmatrix}, \quad \mathbf{K}_{kj1}^{(\ell)}\mathbf{h}_{2s}^{(\ell-1)} = \begin{bmatrix} \mathbf{a}_s \\ \mathbf{0} \end{bmatrix}, \quad \mathbf{K}_{kj1}^{(\ell)}\mathbf{h}_{2s-1}^{(\ell-1)} = \mathbf{0},
$$

$$
\mathbf{V}_{kj1}^{(\ell)}\mathbf{h}_{2s}^{(\ell-1)} = -\eta\begin{bmatrix} \mathbf{0} \\ b_j^{(m)}\mathbf{a}_s \\ \mathbf{0} \end{bmatrix}, \quad \mathbf{V}_{kj1}^{(\ell)}\mathbf{h}_{2s-1}^{(\ell-1)} = \mathbf{0},
$$

$$
\mathbf{Q}_{kj2}^{(\ell)} = -\mathbf{Q}_{kj1}^{(\ell)}, \quad \mathbf{K}_{kj2}^{(\ell)} = \mathbf{K}_{kj1}^{(\ell)}, \quad \mathbf{V}_{kj2}^{(\ell)} = -\mathbf{V}_{kj1}^{(\ell)},
$$

$$
\mathbf{Q}_{kj3}^{(\ell)}\mathbf{h}_{2t-1}^{(\ell-1)} = \begin{bmatrix} 1 \\ -(2t-1) \\ 1 \\ \mathbf{0} \end{bmatrix}, \mathbf{K}_{kj3}^{(\ell)}\mathbf{h}_{2s-1}^{(\ell)} = \begin{bmatrix} 1 \\ 1 \\ (2s-1) \\ \mathbf{0} \end{bmatrix},
$$

$$
\mathbf{V}_{kj3}^{(\ell)}\mathbf{h}_{2t-1}^{(\ell-1)} = \eta\begin{bmatrix} \mathbf{0} \\ a_j^{(m)}\mathbf{q}_k - [(1 - b_j^{(m)})\mu + b_j^{(m)}\tilde{\lambda}]\mathbf{x}_{k,j}^{\ell-1} \\ \mathbf{0} \end{bmatrix}.
$$

Similarly, it can be verified that the constructed attention layer generates

$$
-\eta \cdot \nabla L_{k,j}(\mathbf{x}_{k,j}^{\ell-1}) = -\frac{\eta}{2t-1}\Big[[(1 - b_j^{(m)})\mu\mathbf{I}_d + b_j^{(m)}\tilde{\lambda}] + b_j^{(m)}\sum_{s=1}^{t-1}\mathbf{a}_s\mathbf{a}_s^\top\Big]\mathbf{x}_{k,j}^{\ell-1} + \frac{\eta a_j^{(m)}\mathbf{q}_k}{2t-1}.
$$

Therefore, a construction similar to Step 2 of the proof of Theorem 8 yields Step 2b. Moreover, note that for the construction to exist we need the embedding dimension $D = \mathrm{O}(Adm) = \tilde{\mathcal{O}}(T^{1/4}Ad)$ and the number of hidden neurons $D' = \mathrm{O}(Adm) = \tilde{\mathcal{O}}(T^{1/4}Ad)$.

**Proof of Step 2c** Note that step 2c is a linear transformation from $\mathbf{a}_{t,k}, a_j^{(m)}\widehat{\mathbf{M}_j^{-1}}\mathbf{q}_k, k \in [A], j \in [m]$ to $\sqrt{\mu}[\mathbf{a}_{t,k} + \sum_{j=1}^m (\mu\mathbf{I}_d + b_j^{(m)}(\boldsymbol{\Sigma}_t - \mu\mathbf{I}_d))^{-1}a_j^{(m)}(\boldsymbol{\Sigma}_t - \mu\mathbf{I}_d)\mathbf{a}_{t,k}]$ and we have the fact $x = \sigma(x) - \sigma(-x)$. One can thus choose $\mathbf{W}_1 = \begin{bmatrix} \mathbf{I}_{A(m+1)d} & -\mathbf{I}_{A(m+1)d} & \mathbf{0} \end{bmatrix}$ with $D' = 2A(m+1)d$ and $\mathbf{W}_2$ with $\|\mathbf{W}_2\|_{\mathrm{op}} \leq \sqrt{\mu}(1 + m)$ that implements the linear map.

**Proof of Step 3** Similar to the proof of Step 2b, given $\widehat{\mathbf{\Sigma}_t^{1/2}\mathbf{a}_{t,k}}$ we can apprxoimate $\mathbf{\Sigma}_t^{-1}\widehat{\mathbf{\Sigma}_t^{1/2}\mathbf{a}_{t,k}} \approx \widehat{\mathbf{\Sigma}_t^{-1/2}\mathbf{a}_{t,k}}$ using accelerated gradient descent. Concretely, note that

$$\sqrt{r}\mathbf{\Sigma}_t^{-1}\widehat{\mathbf{\Sigma}_t^{1/2}\mathbf{a}_{t,k}} = \underset{\mathbf{x}\in\mathbb{R}^d}{\arg\min}\,\frac{1}{2(2t-1)}\mathbf{x}^\top\mathbf{\Sigma}_t\mathbf{x} - \frac{1}{2t-1}\left\langle\mathbf{x}, \sqrt{r}\widehat{\mathbf{\Sigma}_t^{1/2}\mathbf{a}_{t,k}}\right\rangle =: \underset{\mathbf{x}\in\mathbb{R}^d}{\arg\min}\,L_k(\mathbf{x})$$

is the global minimizer of a $\tilde{\lambda}/(2t-1)$-convex and $(B_a^2 + \tilde{\lambda})$-smooth function with the conditional number $\tilde{\kappa} \leq 2T(B_a^2 + \tilde{\lambda})/\tilde{\lambda}$. Since

$$\|\sqrt{r}\mathbf{\Sigma}_t^{-1}\widehat{\mathbf{\Sigma}_t^{1/2}\mathbf{a}_{t,k}}\|_2 \leq \sqrt{r}\|\mathbf{M}_j^{-1}\|_{\mathrm{op}}\|\widehat{\mathbf{\Sigma}_t^{1/2}\mathbf{a}_{t,k}}\|_2 \leq 2\sqrt{r}\sqrt{T(B_a^2 + \tilde{\lambda})}B_a,$$

where the last inequality uses the assumption in Step 3. Therefore for any $\varepsilon_0 > 0$, it follows from Proposition 19 that $L = \lceil 2\sqrt{2T(B_a^2 + \tilde{\lambda})/\tilde{\lambda}}\log((1 + \tilde{\kappa})2\sqrt{r}\sqrt{T(B_a^2 + \tilde{\lambda})}B_a/\varepsilon_0)\rceil$ steps of accelerated gradient descent with step size $\eta = 1/(B_a^2 + \tilde{\lambda})$ gives

$$\|\sqrt{r}\widehat{\mathbf{\Sigma}_t^{-1/2}\mathbf{a}_{t,k}} - \sqrt{r}\mathbf{\Sigma}_t^{-1}\widehat{\mathbf{\Sigma}_t^{1/2}\mathbf{a}_{t,k}}\|_2 \leq \varepsilon_0.$$

Following the construction in Step 2 of the proof of Theorem 8 it can be verified that there exists a transformer $\mathrm{TF}_{\boldsymbol{\theta}}(\cdot)$ with

$$L = \lceil 2\sqrt{2T(B_a^2 + \tilde{\lambda})/\tilde{\lambda}}\log((1 + \tilde{\kappa})2\sqrt{r}\sqrt{T(B_a^2 + \tilde{\lambda})}B_a/\varepsilon_0)\rceil = \tilde{\mathcal{O}}(\sqrt{T}), \quad \max_{\ell\in[L]}M^{(l)} = 4A,$$

$$\max_{\ell\in[L]}D'^{(l)} = 4Ad, \quad \|\boldsymbol{\theta}\| \leq \mathcal{O}(A)$$

that implements the AGD steps. Therefore, the approximation error

$$\|\sqrt{r}\widehat{\mathbf{\Sigma}_t^{-1/2}\mathbf{a}_{t,k}} - \sqrt{r}\mathbf{\Sigma}_t^{-1/2}\mathbf{a}_{t,k}\|_2$$

$$\leq \|\sqrt{r}\widehat{\mathbf{\Sigma}_t^{-1/2}\mathbf{a}_{t,k}} - \sqrt{r}\mathbf{\Sigma}_t^{-1}\widehat{\mathbf{\Sigma}_t^{1/2}\mathbf{a}_{t,k}}\|_2 + \|\sqrt{r}\mathbf{\Sigma}_t^{-1}\widehat{\mathbf{\Sigma}_t^{1/2}\mathbf{a}_{t,k}} - \sqrt{r}\mathbf{\Sigma}_t^{-1}\mathbf{\Sigma}_t^{1/2}\mathbf{a}_{t,k}\|_2$$

$$\leq \varepsilon_0 + \sqrt{r}\|\mathbf{\Sigma}_t^{-1}\|_{\mathrm{op}}\|\widehat{\mathbf{\Sigma}_t^{1/2}\mathbf{a}_{t,k}} - \mathbf{\Sigma}_t^{1/2}\mathbf{a}_{t,k}\|_2$$

$$\leq \varepsilon_0 + \frac{\sqrt{r}}{\tilde{\lambda}}\|\widehat{\mathbf{\Sigma}_t^{1/2}\mathbf{a}_{t,k}} - \mathbf{\Sigma}_t^{1/2}\mathbf{a}_{t,k}\|_2 \leq \varepsilon_0 + \varepsilon/2,$$

where the last inequality uses the assumption on the approximation error in Step 2. Letting $\varepsilon_0 = \varepsilon/2$ yields $\|\sqrt{r}\widehat{\mathbf{\Sigma}_t^{-1/2}\mathbf{a}_{t,k}} - \sqrt{r}\mathbf{\Sigma}_t^{-1/2}\mathbf{a}_{t,k}\|_2 \leq \varepsilon$.

In addition to the calculation of $\sqrt{r}\widehat{\mathbf{\Sigma}_t^{-1/2}\mathbf{a}_{t,k}}$, we construct a two-layer attention-only transformer that computes $\langle\widehat{\mu}_t, \mathbf{a}_{t,k}\rangle$. Namely, we choose $\mathbf{Q}_{k1,k2}^{(1)}, \mathbf{K}_{k1,k2}^{(1)}, \mathbf{V}_{k1,k2}^{(1)}$ such that

$$\mathbf{Q}_{k1}^{(1)}\mathbf{h}_{2t-1}^{(0)} = \begin{bmatrix}\widehat{\mu}_t \\ -(2t-1) \\ \mathsf{B} \\ \mathbf{0}\end{bmatrix}, \quad \mathbf{K}_{k1}^{(1)}\mathbf{h}_{2j}^{(0)} = \begin{bmatrix}\mathbf{0} \\ \mathsf{B} \\ 2j \\ \mathbf{0}\end{bmatrix}, \quad \mathbf{K}_{k1}^{(1)}\mathbf{h}_{2j-1}^{(0)} = \begin{bmatrix}\mathbf{a}_{j,k} \\ \mathsf{B} \\ 2j-1 \\ \mathbf{0}\end{bmatrix}, \quad \mathbf{V}_{k1}^{(1)}\mathbf{h}_{2t-1}^{(0)} = \begin{bmatrix}\mathbf{0} \\ \mathbf{e}_k \\ \mathbf{0}\end{bmatrix}$$

$$\mathbf{Q}_{k2}^{(1)}\mathbf{h}_{2t-1}^{(0)} = \begin{bmatrix}-\widehat{\mu}_t \\ -(2t-1) \\ \mathsf{B} \\ \mathbf{0}\end{bmatrix}, \quad \mathbf{K}_{k2}^{(1)}\mathbf{h}_{2j}^{(0)} = \begin{bmatrix}\mathbf{0} \\ \mathsf{B} \\ 2j \\ \mathbf{0}\end{bmatrix}, \quad \mathbf{K}_{k2}^{(1)}\mathbf{h}_{2j-1}^{(0)} = \begin{bmatrix}\mathbf{a}_{j,k} \\ \mathsf{B} \\ 2j-1 \\ \mathbf{0}\end{bmatrix}, \quad \mathbf{V}_{k2}^{(1)}\mathbf{h}_{2t-1}^{(0)} = -\begin{bmatrix}\mathbf{0} \\ \mathbf{e}_k \\ \mathbf{0}\end{bmatrix},$$

where $\mathsf{B} = 2TB_a^2(B_aB_w + \sqrt{2r\log(T/\delta_0)})/\tilde{\lambda} = \tilde{\mathcal{O}}(T)$ is an upper bound of $\langle\widehat{\mu}_t, \mathbf{a}_{t,k}\rangle$ for all $k \in [A]$ under the event $\mathcal{E}_{\delta_0}$, and $\mathbf{e}_k = (0, 0, \ldots, 1, 0, \ldots, 0) \in \mathbb{R}^A$ is the one-hot vector supported on the $k$-th entry. Summing up the attention heads gives the update

$$\mathbf{0} \mapsto \frac{\langle\widehat{\mu}_t, \mathbf{a}_{t,k}\rangle}{2t-1}.$$

Note that one can choose the matrices such that

$$\|\mathbf{Q}^{(1)}_{k1,k2}\|_{\mathrm{op}} \leq \mathsf{B}, \quad \|\mathbf{K}^{(1)}_{k1,k2}\|_{\mathrm{op}} \leq \mathsf{B}, \quad \|\mathbf{V}^{(1)}_{k1,k2}\|_{\mathrm{op}} \leq 1.$$

Thus the norm of the attention layer $\left\|\left\|\boldsymbol{\theta}^{(1)}\right\|\right\| \leq \mathsf{B} + 2$.

Finally, as in the proof of Step 2a we can construct a single-layer single-head attention-only transformer with $\left\|\left\|\boldsymbol{\theta}^{(2)}\right\|\right\| \leq T + 1$ that performs the multiplication

$$\frac{\langle \widehat{\mu}_t, \mathbf{a}_{t,k}\rangle}{2t-1} \mapsto \langle \widehat{\mu}_t, \mathbf{a}_{t,k}\rangle.$$

To estimate the approximation error, note that $\|\mathbf{a}_{t,k}\|_2 \leq B_a$ and $\|\widehat{\mu}_t - \mu_t\|_2 \leq \varepsilon/B_a$ by our assumption in Step 3, it follows immediately that $|\langle \widehat{\mu}_t, \mathbf{a}_{t,k}\rangle - \langle \mu_t, \mathbf{a}_{t,k}\rangle| \leq \varepsilon$ for all $k \in [A]$. Combining the construction of the transformer layers above gives Step 3.

**Proof of Step 4** By Assumption B, $g_{k,\eta_1}(\mathbf{v})$ are $(\varepsilon, R_{\delta_0}, \mathsf{M}_0, \mathsf{C}_0)$-approximable by sum of relus for some $\mathsf{M}_0, \mathsf{C}_0$ depend polynomially on $(1/\varepsilon, 1/\eta_1, 1/\eta, 1/\delta_0, A)$. Since

$$\|\sqrt{r}\boldsymbol{\Sigma}_t^{-1/2}\mathbf{a}_{t,k}\|_2 \leq \sqrt{r}\|\boldsymbol{\Sigma}_t^{-1/2}\|_2\|\mathbf{a}_{t,k}\|_2 \leq \sqrt{\lambda}B_a, \quad |\langle \mu_t, \mathbf{a}_{t,k}\rangle| \leq \|\mu_t\|_2\|\mathbf{a}_{t,k}\|_2 = B_w B_a$$

and $R_{\delta_0} = 2(B_w B_a + \sqrt{\lambda}B_a)$, it follows from the assumption $\varepsilon_3 \leq R_{\delta_0}/2$ and a triangular inequality that $\|\mathbf{v}\|_\infty \leq R_\delta$. Therefore, using Assumption B and stacking up the approximation functions for each coordinate $k \in [A]$ we construct a two-layer MLP with $\|\mathbf{W}_1\|_{\mathrm{op}} \leq \sqrt{A\mathsf{M}_0}, \|\mathbf{W}_1\|_{\mathrm{op}} \leq \mathsf{C}_0$, $D' = \mathsf{M}_0 A$ such that

$$\mathbf{W}_2\sigma(\mathbf{W}_1\mathbf{h}^{(1)}_{2t-1}) = \begin{bmatrix} \mathbf{0} \\ \widehat{\mathbf{v}}_{t1} \\ \vdots \\ \widehat{\mathbf{v}}_{tA} \\ \mathbf{0}, \end{bmatrix}$$

where $(\widehat{\mathbf{v}}_{t1}, \ldots, \widehat{\mathbf{v}}_{tA})$ is supported on $\mathbf{h}^c_{2t-1}$ and $|\widehat{v}_{tk} - g_{k,\eta_1}(\mathbf{v})| \leq \varepsilon$ for all $k \in [A]$.

### F.4 PROOF OF THEOREM 11

Denote the transformer constructed in Theorem 23 by $\mathrm{TF}_{\boldsymbol{\theta}}$. From the proof of Theorem 23, we have

$$\log\frac{\mathsf{Alg}_{\mathrm{TS}}(\bar{a}_t|D_{t-1}, s_t)}{\mathsf{Alg}_{\boldsymbol{\theta}}(\bar{a}_t|D_{t-1}, s_t)} \leq \varepsilon$$

under the event

$$\mathcal{E}_{\delta_0} := \{\max_{t\in[T]} |\varepsilon_t| \leq \sqrt{2r\log(2T/\delta_0)}\} \cup \{\|\mathbf{w}^*\|_2 \leq B_w\} \quad \text{for all } t \in [T]$$

with probability at least $1 - \delta_0$, where $B_w := \sqrt{\lambda}(\sqrt{d} + 2\sqrt{\log(2/\delta_0)})$. Note that due to the unboundedness of the noise $\varepsilon_t$ and parameter vector $\mathbf{w}^*$, Assumption A may not be satisfied. However, setting $\delta_0 = \delta/(2n)$ and applying a union bound gives

$$\log\frac{\mathsf{Alg}_{\mathrm{TS}}(\bar{a}^i_t|D^i_{t-1}, s^i_t)}{\mathsf{Alg}_{\boldsymbol{\theta}}(\bar{a}^i_t|D^i_{t-1}, s^i_t)} \leq \varepsilon, \quad \text{for } t \in [T], i \in [n]. \tag{23}$$

with probability at least $1 - \delta/2$. From the proof of Theorem 6 we see that Assumption A is only used in Eq. (14) in the proof of Lemma 20. Moreover, it can be verified that the same result holds with Assumption A replaced by the condition in Eq. (23). Therefore, we have

$$\left|\mathfrak{R}_{\Lambda,\mathsf{Alg}_{\widehat{\theta}}}(T) - \mathfrak{R}_{\Lambda,\mathsf{Alg}_{\mathrm{TS}}}(T)\right| \leq cT^2\sqrt{\mathcal{R}}\left(\sqrt{\frac{\log[\mathcal{N}_\Theta \cdot T/\delta]}{n}} + \sqrt{\varepsilon_{\mathrm{real}}}\right)$$

$$\leq cT^2\sqrt{\mathcal{R}}\left(\sqrt{\frac{\log[\mathcal{N}_\Theta \cdot T/\delta]}{n}}\right) + \sqrt{T}$$

with probability at least $1 - \delta$ as in Theorem 6, where the second inequality follows as in our setting $\varepsilon_{\text{real}} = \varepsilon = 1/(T^3 \mathcal{R})$. Now, it suffices to show Thompson sampling has the expected regret with

$$\mathbb{E}_{\mathsf{M} \sim \Lambda} \Big[ \sum_{t=1}^{T} \max_{k} \langle \mathbf{a}_{t,k}, \mathbf{w}^* \rangle - \mathfrak{R}_{\mathsf{M}, \mathsf{Alg}_{\mathrm{TS}}}(T) \Big] = \mathcal{O}(d\sqrt{T} \log(Td)).$$

The proof follows similar arguments as in Theorem 36.4 in Lattimore & Szepesvári (2020). Define

$$\tilde{\lambda} := r/\lambda, \quad \beta := \sqrt{r} \Big( \sqrt{2rd \log(4d/\delta_{\mathrm{TS}})} + \sqrt{2 \log(2/\delta_{\mathrm{TS}}) + d \log(1 + TB_a^2/\tilde{\lambda}d)} \Big),$$

where $\delta_{\mathrm{TS}}$ will be specified later, and recall $\mathbf{\Sigma}_t = \tilde{\lambda} \mathbf{I}_d + \sum_{j=1}^{t-1} \mathbf{a}_j \mathbf{a}_j^\top$. Since $\|\mathbf{w}^\star\|_2 \leq \sqrt{2\lambda d \log(4d/\delta_{\mathrm{TS}})}$ with probability at least $1 - \delta_{\mathrm{TS}}/2$ by a union bound, it follows from Theorem 20.5 in Lattimore & Szepesvári (2020) that

$$\mathbb{P}(\|\mathbf{w}^\star\|_2 \leq \sqrt{2\lambda d \log(4d/\delta_{\mathrm{TS}})}, \quad \text{and} \quad \|\mu_t - \mathbf{w}^*\|_{\mathbf{\Sigma}_t} \geq \beta, \text{ for some } i \in [T]) \leq \delta_{\mathrm{TS}},$$

where the probability is taken over the both randomness of the noise and of the bandit instance $\mathsf{M}$.

Let $\mathcal{E}$ be the event where $\|\mu_t - \mathbf{w}^*\|_{\mathbf{\Sigma}_t} \leq \beta$ for all $i \in [T]$, and let $\mathcal{E}_0$ be the event where $\{\|\mathbf{w}^\star\|_2 \leq \sqrt{2\lambda d \log(4d/\delta_{\mathrm{TS}})}\}$. Then $\mathbb{P}(\mathcal{E} \cap \mathcal{E}_0) \geq 1 - \delta_{\mathrm{TS}}$ and the expected regret

$$\mathbb{E}_{\mathsf{M} \sim \Lambda} \Big[ \sum_{t=1}^{T} \max_{k} \langle \mathbf{a}_{t,k}, \mathbf{w}^* \rangle - \mathfrak{R}_{\mathsf{M}, \mathsf{Alg}_{\mathrm{TS}}}(T) \Big]$$

$$= \mathbb{E}[\sum_{t=1}^{T} \max_{j \in [A]} \langle \mathbf{w}^\star, \mathbf{a}_{t,k} \rangle - \langle \mathbf{w}^\star, \mathbf{a}_t \rangle]$$

$$= \mathbb{E}\Big[ \sum_{t=1}^{T} (\max_{j \in [A]} \langle \mathbf{w}^\star, \mathbf{a}_{t,k} \rangle - \langle \mathbf{w}^\star, \mathbf{a}_t \rangle) \mathbf{1}_{\mathcal{E} \cap \mathcal{E}_0} \Big] + \mathbb{E}\Big[ \sum_{t=1}^{T} \max_{j \in [A]} (\langle \mathbf{w}^\star, \mathbf{a}_{t,k} \rangle - \langle \mathbf{w}^\star, \mathbf{a}_t \rangle) \mathbf{1}_{(\mathcal{E} \cap \mathcal{E}_0)^c} \Big]$$

$$\leq \mathbb{E}\Big[ \sum_{t=1}^{T} (\max_{j \in [A]} \langle \mathbf{w}^\star, \mathbf{a}_{t,k} \rangle - \langle \mathbf{w}^\star, \mathbf{a}_t \rangle) \mathbf{1}_{\mathcal{E} \cap \mathcal{E}_0} \Big] + \mathbb{E}[2B_a T \|\mathbf{w}^*\|_2 \mathbf{1}_{(\mathcal{E} \cap \mathcal{E}_0)^c}]$$

$$\leq \mathbb{E}\Big[ \sum_{t=1}^{T} (\max_{j \in [A]} \langle \mathbf{w}^\star, \mathbf{a}_{t,k} \rangle - \langle \mathbf{w}^\star, \mathbf{a}_t \rangle) \mathbf{1}_{\mathcal{E} \cap \mathcal{E}_0} \Big] + 2B_a T \mathbb{E}[\|\mathbf{w}^*\|_2 \mathbf{1}_{(\mathcal{E} \cap \mathcal{E}_0)^c}].$$

Since

$$\mathbb{E}[\|\mathbf{w}^*\|_2 \mathbf{1}_{(\mathcal{E} \cap \mathcal{E}_0)^c}] \leq \mathbb{P}((\mathcal{E} \cap \mathcal{E}_0)^c) \sqrt{2\lambda d \log(4d/\delta_{\mathrm{TS}})} + \int_{\sqrt{2\lambda d \log(4d/\delta_{\mathrm{TS}})}}^{\infty} \mathbb{P}(\|\mathbf{w}^*\|_2 \geq t) dt$$

$$\leq \sqrt{2\lambda d \log(4d/\delta_{\mathrm{TS}})} \delta_{\mathrm{TS}} + d^{3/2} \int_{\sqrt{2\lambda \log(4d/\delta_{\mathrm{TS}})}}^{\infty} \mathbb{P}(|w_1^\star| \geq t) dt$$

$$\leq \sqrt{2\lambda d \log(4d/\delta_{\mathrm{TS}})} \delta_{\mathrm{TS}} + 2\sqrt{2} d^{3/2} \lambda^{1/2} \int_{\sqrt{\log(4d/\delta_{\mathrm{TS}})}}^{\infty} \exp(-t^2) dt$$

$$\leq \sqrt{2\lambda d \log(4d/\delta_{\mathrm{TS}})} \delta_{\mathrm{TS}} + \sqrt{2} d^{3/2} \lambda^{1/2} \int_{\log(4d/\delta_{\mathrm{TS}})}^{\infty} \frac{1}{t^{1/2}} \exp(-t) dt$$

$$\leq 2\sqrt{2\lambda d \log(4d/\delta_{\mathrm{TS}})} \delta_{\mathrm{TS}},$$

where the second line follows from a union bound over $[d]$, and the third line uses properties of subgaussian variables. Therefore, choosing $\delta_{\mathrm{TS}} = 1/[T\sqrt{d}]$ gives

$$\mathbb{E}_{\mathsf{M} \sim \Lambda} \Big[ \sum_{t=1}^{T} \max_{k} \langle \mathbf{a}_{t,k}, \mathbf{w}^* \rangle - \mathfrak{R}_{\mathsf{M}, \mathsf{Alg}_{\mathrm{TS}}}(T) \Big]$$

$$\leq \mathbb{E}\Big[ \sum_{t=1}^{T} (\max_{j \in [A]} \langle \mathbf{w}^\star, \mathbf{a}_{t,k} \rangle - \langle \mathbf{w}^\star, \mathbf{a}_t \rangle) \mathbf{1}_{\mathcal{E} \cap \mathcal{E}_0} \Big] + 6B_a \sqrt{\lambda \log(4d^2 T)}. \quad (24)$$

Now define the event $\mathcal{E}_t := \{\|\mu_t - \mathbf{w}^*\|_{\mathbf{\Sigma}_t} \leq \beta\}$, then we have $\mathcal{E}_t \in \mathcal{F}_{t-1}$ and $\cap_{t=1}^T \mathcal{E}_t = \mathcal{E}$. Also, we define the upper confidence bound $U_t(\mathbf{a}) := \langle \mu_t, \mathbf{a} \rangle + \beta \|\mathbf{a}\|_{\mathbf{\Sigma}_t^{-1}}$, which does not depend on the true parameter $\mathbf{w}^*$. Let $(\mathcal{F}_t)_{t \geq 0}$ denote the filtration generated by the data collected up to time $t$ and the random parameter vector $\mathbf{w}^*$.

Let $\mathbf{a}_t^*$ denote the optimal action at time $t$. Due to the construction of Thompson sampling, we have the distribution of $\mathbf{a}_t^*$ and $\mathbf{a}_t$ are the same conditioned on $\mathcal{F}_{t-1}$. Therefore, $\mathbb{E}[U_t(\mathbf{a}_t^*)|\mathcal{F}_{t-1}] = \mathbb{E}[U_t(\mathbf{a}_t)|\mathcal{F}_{t-1}]$ and

$$
\begin{aligned}
\mathbb{E}\Big[(\langle \mathbf{w}^\star, \mathbf{a}_t^* \rangle - \langle \mathbf{w}^\star, \mathbf{a}_t \rangle)\mathbf{1}_{\mathcal{E} \cap \mathcal{E}_0} \mid \mathcal{F}_{t-1}\Big] &\leq \mathbb{E}\Big[\big(\langle \mathbf{w}^\star, \mathbf{a}_t^* \rangle - U_t(\mathbf{a}_t^*) + U_t(\mathbf{a}_t) - \langle \mathbf{w}^\star, \mathbf{a}_t \rangle\big)\mathbf{1}_{\mathcal{E}_t} \mid \mathcal{F}_{t-1}\Big] \\
&\leq \mathbb{E}\Big[\big(U_t(\mathbf{a}_t) - \langle \mathbf{w}^\star, \mathbf{a}_t \rangle\big)\mathbf{1}_{\mathcal{E}_t} \mid \mathcal{F}_{t-1}\Big] \\
&\leq \mathbb{E}\Big[\big(\|\mathbf{a}_t\|_{\mathbf{\Sigma}_t^{-1}}\|\mu_t - \mathbf{w}^*\|_{\mathbf{\Sigma}_t} + \beta\|\mathbf{a}_t\|_{\mathbf{\Sigma}_t^{-1}}\big)\mathbf{1}_{\mathcal{E}_t} \mid \mathcal{F}_{t-1}\Big] \\
&\leq 2\beta\mathbb{E}[\|\mathbf{a}_t\|_{\mathbf{\Sigma}_t^{-1}}|\mathcal{F}_{t-1}].
\end{aligned}
$$

Moreover, we have

$$
\|\mathbf{a}_t\|_{\mathbf{\Sigma}_t^{-1}} \leq B_a/\sqrt{\tilde{\lambda}}.
$$

Combining the last two displays, we obtain

$$
\mathbb{E}\Big[\sum_{t=1}^T (\max_{j \in [A]} \langle \mathbf{w}^\star, \mathbf{a}_{t,k} \rangle - \langle \mathbf{w}^\star, \mathbf{a}_t \rangle)\mathbf{1}_{\mathcal{E} \cap \mathcal{E}_0}\Big]
$$

$$
\begin{aligned}
&\leq \sum_{t=1}^T \mathbb{E}\Big[2\beta\|\mathbf{a}_t\|_{\mathbf{\Sigma}_t^{-1}} \wedge (B_a/\sqrt{\tilde{\lambda}})\Big] \\
&\leq 2\big(\beta \vee (B_a/\sqrt{\tilde{\lambda}})\big)\mathbb{E}\Big[\sum_{t=1}^T (\|\mathbf{a}_t\|_{\mathbf{\Sigma}_t^{-1}} \wedge 1)\Big] \\
&\leq 2\big(\beta \vee (B_a/\sqrt{\tilde{\lambda}})\big)\sqrt{T}\sqrt{\mathbb{E}\Big[\sum_{t=1}^T (\|\mathbf{a}_t\|_{\mathbf{\Sigma}_t^{-1}}^2 \wedge 1)\Big]} \\
&\leq 2\big(\beta \vee (B_a/\sqrt{\tilde{\lambda}})\big)\sqrt{T}\sqrt{2d\log(1 + TB_a^2/(\tilde{\lambda}d))} \\
&= \mathcal{O}(d\sqrt{T}\log(Td)),
\end{aligned}
$$

where the fourth line uses Cauchy-Schwartz inequality and the fifth line follows from Lemma 19.4 in Lattimore & Szepesvári (2020). Combining the last display with Eq. (24) completes the proof of first part of Theorem 11. Moreover, the second part of Theorem 11 (i.e., the upper bound on $\log \mathcal{N}_\Theta$) follows directly from Lemma 16 and Eq. (9).

### F.5 Proof of Lemma 24

For any $j \neq k$, by definition of $g_{k,\eta_1}$

$$
\begin{aligned}
&|g_{k,\eta_1}(\mathbf{x}_1, \ldots, \mathbf{x}_j, \ldots, \mathbf{x}_A, y_1, \ldots, y_A) - g_{k,\eta_1}(\mathbf{x}_1, \ldots, \mathbf{x}_j', \ldots \mathbf{x}_A, y_1, \ldots, y_A)| \\
&\leq \frac{1}{\eta_1}\Big|\mathbb{P}(\langle \mathbf{x}_k - \mathbf{x}_i, \mathbf{z} \rangle + y_k - y_i \geq 0, \text{ for all } i \in [A]) \\
&\qquad - \mathbb{P}(\langle \mathbf{x}_k - \mathbf{x}_i, \mathbf{z} \rangle + y_k - y_i \geq 0, \text{ for all } i \neq j, \langle \mathbf{x}_k - \mathbf{x}_j', \mathbf{z} \rangle + y_k - y_j \geq 0)\Big| \\
&\leq \frac{1}{\eta_1}\Big(\mathbb{P}(\langle \mathbf{x}_k - \mathbf{x}_j', \mathbf{z} \rangle + y_k - y_j \geq 0 \geq \langle \mathbf{x}_k - \mathbf{x}_j, \mathbf{z} \rangle + y_k - y_j) \\
&\qquad + \mathbb{P}(\langle \mathbf{x}_k - \mathbf{x}_j', \mathbf{z} \rangle + y_k - y_j \leq 0 \leq \langle \mathbf{x}_k - \mathbf{x}_j, \mathbf{z} \rangle + y_k - y_j)\Big) \\
&\leq \frac{1}{\eta_1}\Big(\mathbb{P}(\langle \mathbf{x}_j - \mathbf{x}_j', \mathbf{z} \rangle \geq \langle \mathbf{x}_k - \mathbf{x}_j', \mathbf{z} \rangle + y_k - y_j \geq 0) \\
&\qquad + \mathbb{P}(\langle \mathbf{x}_k - \mathbf{x}_j', \mathbf{z} \rangle + y_k - y_j \leq 0 \leq \langle \mathbf{x}_k - \mathbf{x}_j, \mathbf{z} \rangle + y_k - y_j)\Big)
\end{aligned}
$$

$$\leq \frac{1}{\eta_1}\Big(\mathbb{P}(\langle \mathbf{x}_j - \mathbf{x}'_j, \mathbf{z}\rangle \geq \langle \mathbf{x}_k - \mathbf{x}'_j, \mathbf{z}\rangle + y_k - y_j \geq 0) + \mathbb{P}(\langle \mathbf{x}_j - \mathbf{x}'_j, \mathbf{z}\rangle \leq \langle \mathbf{x}_k - \mathbf{x}'_j, \mathbf{z}\rangle + y_k - y_j \leq 0)\Big).$$

Note that conditioned on $\mathbf{x}_j, \mathbf{x}'_j$ we have

$$\mathbb{P}(|\langle \mathbf{x}_j - \mathbf{x}'_j, \mathbf{z}\rangle| \leq \|\mathbf{x}_j - \mathbf{x}'_j\|_2 \sqrt{2\log(2/\delta_1)}) \geq 1 - \delta_1$$

for any $\delta_1 > 0$. Therefore we further have

$$|g_{k,\eta_1}(\mathbf{x}_1, \ldots, \mathbf{x}_j, \ldots, \mathbf{x}_A, y_1, \ldots, y_A) - g_{k,\eta_1}(\mathbf{x}_1, \ldots, \mathbf{x}'_j, \ldots \mathbf{x}_A, y_1, \ldots, y_A)|$$

$$\leq \frac{1}{\eta_1}\Big[\mathbb{P}\Big(\langle \mathbf{x}_k - \mathbf{x}'_j, \mathbf{z}\rangle + y_k - y_j \in [-\|\mathbf{x}_j - \mathbf{x}'_j\|_2 \sqrt{2\log(2/\delta_1)}, \|\mathbf{x}_j - \mathbf{x}'_j\|_2 \sqrt{2\log(2/\delta_1)}]\Big) + \delta_1\Big]$$

$$\leq \frac{1}{\eta_1}\Big[\sup_{A \in \mathcal{F}, \mu(A) = 2\|\mathbf{x}_j - \mathbf{x}'_j\|_2 \sqrt{2\log(2/\delta_1)}} \mathbb{P}(\langle \mathbf{x}_k - \mathbf{x}'_j, \mathbf{z}\rangle \in A) + \delta_1\Big]$$

$$\leq \frac{1}{\eta_1}\Big(\frac{2\|\mathbf{x}_j - \mathbf{x}'_j\|_2 \sqrt{2\log(2/\delta_1)}}{\sqrt{2\pi}\eta} + \delta_1\Big) \leq \frac{1}{\eta_1}\Big(\frac{2\|\mathbf{x}_j - \mathbf{x}'_j\|_2}{\eta\delta_1} + \delta_1\Big)$$

for any $\delta_1 > 0$, where the last inequality follows from the fact that standard Gaussian has probability density less than $1/\sqrt{2\pi}$ everywhere, $\|\mathbf{x}_k - \mathbf{x}'_j\|_2 \leq \eta$ and $\log(2/\delta_1) \leq 4/\delta_1^2$. Choosing $\delta_1 = 1 \wedge \sqrt{\frac{2\|\mathbf{x}_j - \mathbf{x}'_j\|_2}{\eta}}$ gives

$$|g_{k,\eta_1}(\mathbf{x}_1, \ldots, \mathbf{x}_j, \ldots, \mathbf{x}_A, y_1, \ldots, y_A) - g_{k,\eta_1}(\mathbf{x}_1, \ldots, \mathbf{x}'_j, \ldots \mathbf{x}_A, y_1, \ldots, y_A)|$$

$$\leq \frac{2}{\eta_1}\Big(\sqrt{\frac{2\|\mathbf{x}_j - \mathbf{x}'_j\|_2}{\eta}} + \frac{2\|\mathbf{x}_j - \mathbf{x}'_j\|_2}{\eta}\Big)$$

Similarly, for $\mathbf{x}_k \neq \mathbf{x}'_k$, we have

$$|g_{k,\eta_1}(\mathbf{x}_1, \ldots, \mathbf{x}_k, \ldots, \mathbf{x}_A, y_1, \ldots, y_A) - g_{k,\eta_1}(\mathbf{x}_1, \ldots, \mathbf{x}'_k, \ldots \mathbf{x}_A, y_1, \ldots, y_A)|$$

$$\leq \frac{1}{\eta_1}\Big(\mathbb{P}(\langle \mathbf{x}_k - \mathbf{x}_i, \mathbf{z}\rangle + y_k - y_i \geq 0 \geq \langle \mathbf{x}'_k - \mathbf{x}_i, \mathbf{z}\rangle + y_k - y_i, \text{ for some } i \in [A])$$

$$+ \mathbb{P}(\langle \mathbf{x}_k - \mathbf{x}_i, \mathbf{z}\rangle + y_k - y_i \leq 0 \leq \langle \mathbf{x}'_k - \mathbf{x}_i, \mathbf{z}\rangle + y_k - y_i, \text{ for some } i \in [A])\Big)$$

$$\leq \sum_{i \neq k} \frac{1}{\eta_1}\Big(\mathbb{P}(\langle \mathbf{x}_k - \mathbf{x}_i, \mathbf{z}\rangle + y_k - y_i \geq 0 \geq \langle \mathbf{x}'_k - \mathbf{x}_i, \mathbf{z}\rangle + y_k - y_i)$$

$$+ \mathbb{P}(\langle \mathbf{x}_k - \mathbf{x}_i, \mathbf{z}\rangle + y_k - y_i \leq 0 \leq \langle \mathbf{x}'_k - \mathbf{x}_i, \mathbf{z}\rangle + y_k - y_i)\Big)$$

$$\leq \frac{A}{\eta_1} \max_{i \neq k}\Big(\mathbb{P}(\langle \mathbf{x}_k - \mathbf{x}'_k, \mathbf{z}\rangle \geq \langle \mathbf{x}_i - \mathbf{x}'_k, \mathbf{z}\rangle + y_i - y_k \geq 0) + \mathbb{P}(\langle \mathbf{x}_k - \mathbf{x}'_k, \mathbf{z}\rangle \leq \langle \mathbf{x}_i - \mathbf{x}'_k, \mathbf{z}\rangle + y_i - y_k \leq 0)\Big).$$

Following the same argument, we have

$$|g_{k,\eta_1}(\mathbf{x}_1, \ldots, \mathbf{x}_k, \ldots, \mathbf{x}_A, y_1, \ldots, y_A) - g_{k,\eta_1}(\mathbf{x}_1, \ldots, \mathbf{x}'_k, \ldots \mathbf{x}_A, y_1, \ldots, y_A)|$$

$$\leq \frac{2A}{\eta_1}\Big(\sqrt{\frac{2\|\mathbf{x}_k - \mathbf{x}'_k\|_2}{\eta}} + \frac{2\|\mathbf{x}_k - \mathbf{x}'_k\|_2}{\eta}\Big).$$

Likewise, for any $j \neq k$ we have

$$|g_{k,\eta_1}(\mathbf{x}_1, \ldots, \mathbf{x}_A, y_1, \ldots, y_j, \ldots, y_A) - g_{k,\eta_1}(\mathbf{x}_1, \ldots, \mathbf{x}_A, y_1, \ldots, y'_j, \ldots, y_A)|$$

$$\leq \frac{1}{\eta}\Big(\mathbb{P}(\langle \mathbf{x}_k - \mathbf{x}_j, \mathbf{z}\rangle \in [\min\{y_k - y_j, y_k - y'_j\}, \max\{y_k - y_j, y_k - y'_j\}])\Big)$$

$$\leq \frac{1}{\eta_1} \sup_{A \in \mathcal{F}, \mu(A) = 2|y_j - y'_j|} \mathbb{P}(\langle \mathbf{x}_k - \mathbf{x}_j, \mathbf{z}\rangle \in A)$$

$$\leq \frac{1}{\eta_1} \frac{2|y_j - y'_j|}{\sqrt{2\pi}\eta} \leq \frac{2|y_j - y'_j|}{\eta\eta_1}$$

and

$$|g_{k,\eta_1}(\mathbf{x}_1,\ldots,\mathbf{x}_A,y_1,\ldots,y_k,\ldots,y_A) - g_{k,\eta_1}(\mathbf{x}_1,\ldots,\mathbf{x}_A,y_1,\ldots,y_k',\ldots,y_A)|$$

$$\leq \sum_{j\neq k} \frac{1}{\eta}\Big(\mathbb{P}(\langle\mathbf{x}_k - \mathbf{x}_j,\mathbf{z}\rangle \in [\min\{y_k' - y_j, y_k - y_j\},\max\{y_k' - y_j, y_k - y_j\}])\Big)$$

$$\leq \frac{A}{\eta_1} \sup_{A\in\mathcal{F},\mu(A)=2|y_j-y_j'|} \mathbb{P}(\langle\mathbf{x}_k - \mathbf{x}_j,\mathbf{z}\rangle \in A)$$

$$\leq \frac{A}{\eta_1}\frac{2|y_j - y_j'|}{\sqrt{2\pi\eta}} \leq \frac{2A|y_j - y_j'|}{\eta\eta_1}.$$

# G LEARNING IN-CONTEXT RL IN MARKOV DECISION PROCESSES

Throughout this section, we use $c > 0$ to denote universal constants whose values may vary from line to line. Moreover, for notational simplicity, we use $O(\cdot)$ to hide universal constants, $\tilde{\mathcal{O}}(\cdot)$ to hide poly-logarithmic terms in $(H, K, S, A, 1/\tau)$.

This section is organized as follows. Section G.1 discusses the embedding and extraction formats of transformers for Markov decision processes. Section G.2 describes the UCB-VI and the soft UCB-VI algorithms. We prove Theorem 12 in Section G.3 and prove Theorem 13 in Section G.4.

## G.1 EMBEDDING AND EXTRACTION MAPPINGS

To embed MDP problems into transformers, we consider an embedding similar to that for linear bandits. For each episode $k \in [K]$, we construct $2H + 1$ tokens. Concretely, for each $t \in [T]$ in the $k$-th episode, we write $t = H(k-1) + h$ and construct two tokens

$$\mathbf{h}_{2(t-1)+k} = \begin{bmatrix} -----\mathbf{0}_{A+1}----- \\ -----s_{k,h}----- \\ -----\mathbf{0}_A----- \\ \mathbf{0} \\ \mathbf{pos}_{2(t-1)+k} \end{bmatrix} =: \begin{bmatrix} \mathbf{h}_{2(t-1)+k}^a \\ \mathbf{h}_{2(t-1)+k}^b \\ \mathbf{h}_{2(t-1)+k}^c \\ \mathbf{h}_{2(t-1)+k}^d \end{bmatrix}, \quad \mathbf{h}_{2t-1+k} = \begin{bmatrix} a_{k,h} \\ -----r_{k,h}----- \\ -----\mathbf{0}_S----- \\ -----\mathbf{0}_A----- \\ \mathbf{0} \\ \mathbf{pos}_{2t-1+k} \end{bmatrix} =: \begin{bmatrix} \mathbf{h}_{2t-1+k}^a \\ \mathbf{h}_{2t-1+k}^b \\ \mathbf{h}_{2t-1+k}^c \\ \mathbf{h}_{2t-1+k}^d \end{bmatrix},$$

where $s_{k,h}, a_{k,h}$ are represented using one-hot embedding (we let $s_{k,H+1} = \mathbf{0}_S$), $\mathbf{h}_{2(t-1)+k}^c$ is used to store the (unnormalized) policy at time $t$ given current state $s_{k,h}$, $\mathbf{0}$ in $\mathbf{h}^d$ denotes an additional zero vector. At the end of each episode $k$, we add an empty token

$$\mathbf{h}_{(2H+1)k} = \mathbf{h}_k^{\mathsf{emp}} := \begin{bmatrix} \mathbf{0} & \mathbf{pos}_{(2H+1)k} \end{bmatrix}^\top$$

to store intermediate calculations. We also include in the tokens the positional embedding $\mathbf{pos}_i := (k, h, v_i, i, i^2, 1)^\top$ for $i \in [2T + K]$, where $v_i := \mathbf{1}_{\{\mathbf{h}_i^a = \mathbf{0}\}}$ denote the tokens that do not embed actions and rewards. In addition, we define the token matrix $\mathbf{H}_t := [\mathbf{h}_1,\ldots,\mathbf{h}_{2t-1+k}] \in \mathbb{R}^{D\times(2t-1+k)}$ for all $t \in [T]$.

**Offline pretraining** Similar to the bandit setting, during pretraining the transformer $\mathrm{TF}_{\boldsymbol{\theta}}$ takes in $\mathbf{H}_T^{\mathrm{pre}} := \mathbf{H}_T$ as the input token matrix, and generates $\mathbf{H}_T^{\mathrm{post}} := \mathrm{TF}_{\boldsymbol{\theta}}(\mathbf{H}_T^{\mathrm{pre}})$ as the output. For each time $t \in [T]$, we define the induced policy $\mathsf{Alg}_{\boldsymbol{\theta}}(\cdot|D_{t-1}, s_t) := \frac{\exp(\mathbf{h}_{2(t-1)+k}^{\mathrm{post},c})}{\|\exp(\mathbf{h}_{2(t-1)+k}^{\mathrm{post},c})\|_1} \in \Delta^A$, whose $i$-th entry is the probability of selecting the $i$-th action (denoted by the one-hot vector $\mathbf{e}_i$) given $(D_{t-1}, s_t)$. We then find the transformer $\widehat{\boldsymbol{\theta}} \in \Theta$ by solving Eq. (3).

**Rollout** At each time $t \in [T]$, given the current state $s_t$ and previous data $D_{t-1}$, we first construct the token matrix $\mathbf{H}_{\mathrm{roll},t}^{\mathrm{pre}} \in \mathbb{R}^{D\times 2(t-1)+k}$ that consists of tokens up to the first token for time $t$. The transformer then takes $\mathbf{H}_{\mathrm{roll},t}^{\mathrm{pre}}$ as the input and generates $\mathbf{H}_{\mathrm{roll},t}^{\mathrm{post}} = \mathrm{TF}_{\boldsymbol{\theta}}(\mathbf{H}_{\mathrm{roll},t}^{\mathrm{pre}})$. Next, the agent selects an action $a_t \in \mathcal{A}$ following the induced policy $\mathsf{Alg}_{\boldsymbol{\theta}}(\cdot|D_{t-1}, s_t) := \frac{\exp(\mathbf{h}_{2(t-1)+k}^{\mathrm{post},c})}{\|\exp(\mathbf{h}_{2(t-1)+k}^{\mathrm{post},c})\|_1} \in \Delta^A$ and observes the reward $r_t$ and next state $s_{t+1}$ ($s_{t+1} \sim \mu_1$ if $t$ is the last time step in an episode).

**Embedding and extraction mappings** To integrate the above construction into our general framework in Section 2, for $t = (k-1)H + h$, we have the embedding vectors

$$\mathtt{h}(s_t) := \mathbf{h}_{2(t-1)+k}, \quad \mathtt{h}(a_t, r_t) := \mathbf{h}_{2t-1+k}.$$

For $N \geq 1$, write

$$\lceil (N+1)/2 \rceil = (k_N - 1)H + h_N$$

for some $h_N \in [H]$, and define the concatenation operator

$$\mathtt{cat}(\mathbf{h}_1, \ldots, \mathbf{h}_N) := [\mathbf{h}_1, \ldots, \mathbf{h}_{2H}, \mathbf{h}_1^{\mathsf{emp}}, \mathbf{h}_{2H+1}, \ldots, \mathbf{h}_{4H}, \mathbf{h}_2^{\mathsf{emp}}, \mathbf{h}_{4H+1}, \ldots, \mathbf{h}_N] \in \mathbb{R}^{N+k_N-1},$$

where we insert an empty token $\mathbf{h}_k^{\mathsf{emp}}$ (i.e., a token with $\mathbf{h}^{\{a,b,c\}} = \mathbf{0}$) at the end of each episode $k$.

In this case, we have the input token matrix

$$\mathbf{H} = \mathbf{H}_{\mathrm{roll},t}^{\mathrm{pre}} := \mathtt{cat}(\mathtt{h}(s_1), \mathtt{h}(a_1, r_1), \ldots, \mathtt{h}(a_{t-1}, r_{t-1}), \mathtt{h}(s_t)) \in \mathbb{R}^{D \times [2(t-1)+k]},$$

the output token matrix $\overline{\mathbf{H}} = \mathbf{H}_{\mathrm{roll},t}^{\mathrm{post}}$, and the linear extraction map $\mathtt{A}$ satisfies

$$\mathtt{A} \cdot \overline{\mathbf{h}}_{-1} = \mathtt{A} \cdot \overline{\mathbf{h}}_{2(t-1)+k}^{\mathrm{post}} = \mathbf{h}_{2(t-1)+k}^{\mathrm{post},c}.$$

## G.2 UCB-VI AND SOFT UCB-VI

We show that transformers with the embedding in Section G.1 can approximately implement the UCB-VI algorithm in Azar et al. (2017). Namely, UCB-VI implements the following steps:

for each episode $k \in [K]$ and each step $h = H, \ldots, 1$

1. Compute the estimated transition matrix $\widehat{P}_h(s'|s,a) := \frac{N_h(s,a,s')}{N_h(s,a) \vee 1}$, where $N_h(s,a,s')$ denotes the number of times the state-action-next-state tuple $(s,a,s')$ has been visited in the first $k-1$ episodes, and $N_h(s,a) = \sum_{s'} N_h(s,a,s')$ (we assume $N_H(s,a,s') = 0$ and let $N_H(s,a)$ be the number of times $(s,a)$ is visited at timestep $H$).

2. Calculate the estimated Q-function

$$\widehat{Q}_h(s,a) = \min\{H, r_h(s,a) + b_h(s,a) + \sum_{s' \in \mathcal{S}} \widehat{P}_h(s' \mid s, a)\widehat{V}_{h+1}(s')\},$$

   where the bonus $b_h(s,a) = 2H\sqrt{\frac{\log(SAT/\delta)}{N_h(s,a) \vee 1}}$, $\widehat{V}_{H+1}(s) := 0$ for all $s \in \mathcal{S}$ and $\widehat{V}_h(s) := \max_{a \in \mathcal{A}} \widehat{Q}_h(s,a)$.

Throughout this section, we choose the small probability $\delta = 1/(KH)$.

During policy execution, at each step $h \in [H]$, UCB-VI takes the greedy action $a_h := \arg\max_a \widehat{Q}_h(s_h, a)$ and observes the reward and next state $(r_h, s_{h+1})$. To facilitate pretraining, in this work we consider a soft version of UCB-VI, which takes action $a_h$ following the softmax policy

$$\pi_h(a|s_h) = \frac{\exp(\widehat{Q}_h(s_h, a)/\tau)}{\left\| \exp(\widehat{Q}_h(s_h, a)/\tau) \right\|_1}$$

using the estimated $Q$-function for some sufficiently small $\tau > 0$. Note that soft UCB-VI recovers UCB-VI as $\tau \to 0$.

## G.3 PROOF OF THEOREM 12

Throughout the proof, we abuse the notations $\mathbf{h}_i^\star$ for $\star \in \{a, b, c, d\}$ to denote the corresponding positions in the token vector $\mathbf{h}_i$. For any $t' \in [T]$, we let $k(t'), h(t')$ be the non-negative integers such that $t' = H(k(t') - 1) + h(t')$ and $h(t') \in [H]$. For the current time $t$, we use the shorthands $k = k(t), h = h(t)$. For a token index $i \in [(2H+1)K]$, let $\overline{k}(i), \overline{h}(i)$ be the episode and time step the $i$-th token corresponds to (for the empty tokens we set $h = H+1$). Given the input token

matrix $\mathbf{H}^{\mathrm{pre}}_{\mathrm{roll},t}$, we construct a transformer that implements the following steps on the last token. $\mathbf{h}^{\star}_{2(t-1)+k} = \mathbf{h}^{\mathrm{pre},\star}_{2(t-1)+k}$ for $\star \in \{a,b,c,d\}$

$$
\begin{bmatrix} \mathbf{h}^{\mathrm{pre},a}_{2(t-1)+k} \\ \mathbf{h}^{\mathrm{pre},b}_{2(t-1)+k} \\ \mathbf{h}^{\mathrm{pre},c}_{2(t-1)+k} \\ \mathbf{h}^{\mathrm{pre},d}_{2(t-1)+k} \end{bmatrix} \xrightarrow{\text{step 1}} \begin{bmatrix} \mathbf{h}^{\mathrm{pre},\{a,b,c\}}_{2(t-1)+k} \\ N_1(s,a,s') \\ \vdots \\ N_H(s,a,s') \\ N_1(s,a) \\ \vdots \\ N_H(s,a,s') \\ N_1(s,a)r_1(s,a) \\ \vdots \\ N_H(s,a,s')r_H(s,a) \\ \star \\ \mathbf{0} \\ \mathbf{pos}_{2(t-1)+k} \end{bmatrix} \xrightarrow{\text{step 2}} \begin{bmatrix} \mathbf{h}^{\mathrm{pre},\{a,b,c\}}_{2(t-1)+k} \\ \widehat{P}_1(s,a,s') \\ \vdots \\ \widehat{P}_H(s,a,s') \\ \star \\ \mathbf{0} \\ \mathbf{pos}_{2(t-1)+k} \end{bmatrix} \xrightarrow{\text{step 3}} \begin{bmatrix} \mathbf{h}^{\mathrm{pre},\{a,b,c\}}_{2(t-1)+k} \\ \widehat{Q}_1(s,a,s') \\ \vdots \\ \widehat{Q}_H(s,a,s') \\ \widehat{V}_1(s) \\ \vdots \\ \widehat{V}_H(s) \\ \star \\ \mathbf{0} \\ \mathbf{pos}_{2(t-1)+k} \end{bmatrix}
$$

$$
\xrightarrow{\text{step 4}} \begin{bmatrix} \mathbf{h}^{\mathrm{pre},\{a,b\}}_{2(t-1)+k} \\ \dfrac{\widehat{Q}_h(s_t,a_1)}{\tau} \\ \vdots \\ \dfrac{\widehat{Q}_h(s_t,a_1)}{\tau} \\ \mathbf{h}^{d}_{2(t-1)+k} \end{bmatrix} =: \begin{bmatrix} \mathbf{h}^{\mathrm{post},a}_{2(t-1)+k} \\ \mathbf{h}^{\mathrm{post},b}_{2(t-1)+k} \\ \mathbf{h}^{\mathrm{post},c}_{2(t-1)+k} \\ \mathbf{h}^{\mathrm{post},d}_{2(t-1)+k} \end{bmatrix}, \tag{25}
$$

where $N_{\mathsf{h}}(s,a,s'), \widehat{P}_{\mathsf{h}}(s,a,s'), \widehat{Q}_{\mathsf{h}}(s,a,s') \in \mathbb{R}^{S^2 \times A}$, $N_{\mathsf{h}}(s,a) \in \mathbb{R}^{S \times A}$, $\widehat{V}_{\mathsf{h}}(s) \in \mathbb{R}^S$ for all $\mathsf{h} \in [H]$, and $\star$ denote additional quantities in $\mathbf{h}^{d}_{2(t-1)+k}$. Given the current state $s_t$, the transformer $\mathrm{TF}_{\boldsymbol{\theta}}(\cdot)$ generates the policy

$$
\mathsf{Alg}_{\boldsymbol{\theta}}(\cdot \mid D_{t-1}, s_t) := \frac{\exp(\mathbf{h}^{\mathrm{post},c}_{2(t-1)+k})}{\|\exp(\mathbf{h}^{\mathrm{post},c}_{2(t-1)+k})\|_1} \in \Delta^A.
$$

We claim the following results which we will prove later.

**Step 1** There exists an attention-only transformer $\mathrm{TF}_{\boldsymbol{\theta}}(\cdot)$ with
$$
L = 4, \quad \max_{\ell \in [L]} M^{(l)} \le \mathrm{O}(HS^2A), \quad \|\boldsymbol{\theta}\| \le \mathrm{O}(HK + HS^2A)
$$
that implements step 1 in (21).

**Step 2** There exists a one-layer transformer $\mathrm{TF}_{\boldsymbol{\theta}}(\cdot)$ with
$$
L = 1, \quad M \le \mathrm{O}(HS^2A), \quad D' \le \mathrm{O}(K^2HS^2A), \quad \|\boldsymbol{\theta}\| \le \tilde{\mathcal{O}}(HS^2A + K^3 + KH)
$$
that implements step 2 in (21).

**Step 3** There exists a transformer $\mathrm{TF}_{\boldsymbol{\theta}}(\cdot)$ with
$$
L = 2H, \quad \max_{\ell \in [L]} M^{(l)} \le 2SA, \quad \max_{\ell \in [L]} D'^{(l)} \le 3SA, \quad \|\boldsymbol{\theta}\| \le \mathrm{O}(H + SA)
$$
that implements step 3 (i.e., value iteration) in (21).

**Step 4** There exists an attention-only transformer $\mathrm{TF}_{\boldsymbol{\theta}}(\cdot)$ with
$$
L = 3, \quad \max_{\ell \in [L]} M^{(\ell)} = \mathrm{O}(HA), \quad \|\boldsymbol{\theta}\| \le \mathrm{O}(H(K+A) + 1/\tau)
$$
that implements step 4 in (21).

From the construction of Step 1—4, we verify that one can choose the constructed transformer to have the embedding dimension $D = \mathrm{O}(HS^2A)$. Moreover, due to the boundedness of the reward function, $Q$-function and the fact that the bonus $b(s,a) \le \tilde{\mathcal{O}}(H)$, we verify that there exists some $\mathsf{R} > 0$ with $\log \mathsf{R} = \tilde{\mathcal{O}}(1)$ such that $\|\mathbf{h}^{(\ell)}_i\|_2 \le \mathsf{R}$ for all layer $\ell \in [L]$ and all token $i \in [K(2H+1)]$. Therefore, similar to what we do in the proof of Theorem 8, 10, we may w.l.o.g. consider transformers without truncation (i.e., $\mathsf{R} = \infty$) in our construction of step 1—4 in (25).

**Proof of Step 1** We prove this step by constructing a transformer that implements the following two steps:

Step 1a For each $t' < t$ with $t' = (k'-1)H + h'$ for some $h' \in [H]$, we add $s_{k',h'}, (a_{k',h'}, r_{k',h'})$ from $\mathbf{h}^b_{2(t'-1)+k'}$ and $\mathbf{h}^a_{2t'-1+k'}$ to $\mathbf{h}^d_{2t'+k'}$.

Step 1b Compute $N_{\mathsf{h}}(s, a, s'), N_{\mathsf{h}}(s, a)$ for $\mathsf{h} \in [H]$ and assign them to the current token $\mathbf{h}^d_{2(t-1)+k}$.

For step 1a, we can construct a two-layer attention-only transformer with $\mathbf{Q}^{(1)}_{1,2,3}, \mathbf{K}^{(1)}_{1,2,3}, \mathbf{V}^{(1)}_{1,2,3}$ such that for all $i \le 2(t-1) + k$

$$\mathbf{Q}^{(1)}_1 \mathbf{h}^{(0)}_i = \begin{bmatrix} \overline{k}(i) + 1 - v_i \\ \mathsf{B} \\ 1 \\ i \end{bmatrix}, \quad \mathbf{K}^{(1)}_1 \mathbf{h}^{(0)}_i = \begin{bmatrix} -\mathsf{B} \\ \overline{k}(i) \\ i + 3 \\ -1 \end{bmatrix}, \quad \mathbf{V}^{(1)}_1 \mathbf{h}^{(0)}_{2(t'-1)+k'} = \begin{bmatrix} \mathbf{0} \\ \mathbf{0}_{A+1} \\ s_{k',h'} \\ \mathbf{0} \end{bmatrix},$$

$$\mathbf{V}^{(1)}_1 \mathbf{h}^{(0)}_{2t'-1+k'} = \begin{bmatrix} \mathbf{0} \\ a_{k',h'} \\ r_{k',h'} \\ \mathbf{0}_S \\ \mathbf{0}, \end{bmatrix}$$

where we choose $\mathsf{B} = 4$ and $\mathbf{V}\mathbf{h}^{(0)}$ are supported on some entries in $\mathbf{h}^{(0),d}$. Moreover, we choose $\mathbf{Q}^{(1)}_3 = \mathbf{Q}^{(1)}_2 = \mathbf{Q}^{(1)}_1, \mathbf{V}^{(1)}_2 = \mathbf{V}^{(1)}_3 = -\mathbf{V}^{(1)}_1$ and $\mathbf{K}^{(1)}_2, \mathbf{K}^{(1)}_3$ such that

$$\mathbf{K}^{(1)}_2 \mathbf{h}^{(0)}_i = \begin{bmatrix} -\mathsf{B} \\ \overline{k}(i) \\ i + 2 \\ -1 \end{bmatrix}, \quad \mathbf{K}^{(1)}_3 \mathbf{h}^{(0)}_i = \begin{bmatrix} -\mathsf{B} \\ \overline{k}(i) \\ i + 1 \\ -1. \end{bmatrix}$$

We verify that $\|\mathbf{Q}^{(1)}_\star\|_{\mathrm{op}}, \|\mathbf{K}^{(1)}_\star\|_{\mathrm{op}} = 4, \|\mathbf{V}^{(1)}_\star\|_{\mathrm{op}} = 1$ for $\star \in [3]$. Summing up the heads, we obtain the following update on a subset of coordinates in $\mathbf{h}^{(0),d}_{2t'+k'}$:

$$\begin{aligned}
\mathbf{0}_{S+A+1} \to \mathbf{0}_{S+A+1} &+ \sum_{j=1}^{3} \sum_{i=1}^{2t'+k'} \sigma(\langle \mathbf{Q}^{(1)}_j \mathbf{h}^{(0)}_{2t'+k'}, \mathbf{K}^{(1)}_j \mathbf{h}^{(0)}_i \rangle) \mathbf{V}_j \mathbf{h}^{(0)}_i \\
&= \frac{1}{2t' + k'} [(\mathbf{V}^{(1)}_1 \mathbf{h}^{(0)}_{2t'+k'-2} + 2\mathbf{V}^{(1)}_1 \mathbf{h}^{(0)}_{2t'+k'-1} + 3\mathbf{V}^{(1)}_1 \mathbf{h}^{(0)}_{2t'+k'}) \\
&\quad - (\mathbf{V}^{(1)}_1 \mathbf{h}^{(0)}_{2t'+k'-1} + 2\mathbf{V}^{(1)}_1 \mathbf{h}^{(0)}_{2t'+k'}) - \mathbf{V}^{(1)}_1 \mathbf{h}^{(0)}_{2t'+k'})] \\
&= \frac{1}{2t' + k'} (\mathbf{V}^{(1)}_1 \mathbf{h}^{(0)}_{2t'+k'-2} + \mathbf{V}^{(1)}_1 \mathbf{h}^{(0)}_{2t'+k'-1}) \\
&= \frac{1}{2t' + k'} \begin{bmatrix} a_{k',h'} \\ r_{k',h'} \\ s_{k',h'} \end{bmatrix}.
\end{aligned}$$

Note that $\langle \mathbf{Q}^{(1)} \mathbf{h}^{(0)}_i, \mathbf{K}^{(1)} \mathbf{h}^{(0)}_j \rangle \le 0$ for $i = 2t' - 1 + k'$ (i.e., all tokens that embed the action and reward) since $v_i = 0$, it follows that no update happens on the tokens in which we embed the action and reward (i.e., the corresponding part of $\mathbf{h}^d$ remains zero). Moreover, it should be noted that no update happens on tokens with $h = 1$.

We then use another attention layer to multiply the updated vectors by a factor of $2t' + k'$, namely, to perform the map

$$\frac{1}{2t' + k'} \begin{bmatrix} a_{k',h'} \\ r_{k',h'} \\ s_{k',h'} \end{bmatrix} \mapsto \begin{bmatrix} a_{k',h'} \\ r_{k',h'} \\ s_{k',h'} \end{bmatrix},$$

where the output vector is supported on coordinates different from the input vectors. This can be achieved by choosing $\|\mathbf{Q}_1^{(2)}\|_{\text{op}} \leq (2H+1)K, \|\mathbf{K}_1^{(2)}\|_{\text{op}} \leq (2H+1)K, \|\mathbf{V}_1^{(2)}\|_{\text{op}} \leq 1$ such that

$$\mathbf{Q}_1^{(2)}\mathbf{h}_i^{(1)} = \begin{bmatrix} i^2 \\ -(2H+1)Ki^2 \\ 1 \\ \mathbf{0} \end{bmatrix}, \quad \mathbf{K}_1^{(2)}\mathbf{h}_j^{(1)} = \begin{bmatrix} 1 \\ 1 \\ (2H+1)Kj^2 \\ \mathbf{0} \end{bmatrix}, \quad \mathbf{V}_1^{(2)}\mathbf{h}_{2t'+k'}^{(1)} = \frac{1}{2t'+k'}\begin{bmatrix} \mathbf{0} \\ a_{k',h'} \\ r_{k',h'} \\ s_{k',h'} \\ \mathbf{0} \end{bmatrix},$$

(26)

and noting that $\left\langle \mathbf{Q}_1^{(2)}\mathbf{h}_i^{(1)}, \mathbf{Q}_1^{(2)}\mathbf{h}_j^{(1)} \right\rangle = i$ when $j = i$ and otherwise 0.

For step 1b, we show that it can be implemented using a two-layer attention-only transformer.

To compute $N_\mathsf{h}(s, a, s')$, in the first layer we construct $M = 10HS^2A$ heads with the query, key, value matrices $\{\mathbf{Q}_{\mathsf{ijkh},s}^{(1)}\}_{s=1}^{10}, \{\mathbf{K}_{\mathsf{ijkh},s}^{(1)}\}_{s=1}^{10}, \{\mathbf{V}_{\mathsf{ijkh},s}^{(1)}\}_{s=1}^{10}$ such that for all $i \leq 2(t-1) + k$ and i, k $\in [S]$, j $\in [A]$, h $\in [H]$

$$\mathbf{Q}_{\mathsf{ijkh},1}^{(1)}\mathbf{h}_i^{(0)} = \begin{bmatrix} \mathsf{B}(v_i - 1) \\ \mathsf{Be_i} \\ \mathsf{Be_j} \\ \mathsf{Be_k} \\ 1 \\ 1 \\ \mathsf{h} \end{bmatrix}, \quad \mathbf{K}_{\mathsf{ijkh},1}^{(1)}\mathbf{h}_i^{(0)} = \begin{bmatrix} 1 \\ s_{\overline{k}(i),\overline{h}(i)-1} \\ a_{\overline{k}(i),\overline{h}(i)-1} \\ s_{\overline{k}(i),\overline{h}(i)} \\ -3\mathsf{B} \\ 1 - \overline{h}(i) \\ 1 \end{bmatrix}, \quad \mathbf{V}_{\mathsf{ijkh},1}^{(1)}\mathbf{h}_i^{(0)} = -\begin{bmatrix} \mathbf{0} \\ \mathsf{e}_{\mathsf{ijk}}^{N_\mathsf{h}} \\ \mathbf{0} \end{bmatrix},$$

where we choose $\mathsf{B} = 2H$ and $\mathsf{e}_{\mathsf{ijk}}^{\mathsf{h}}$ denotes the one-hot vector supported on the $(\mathsf{i,j,k})$-entry in $N_\mathsf{h}(s, a, s')$. We similarly construct

$$\mathbf{Q}_{\mathsf{ijkh},2}^{(1)}\mathbf{h}_i^{(0)} = \begin{bmatrix} \mathsf{B}(v_i - 1) \\ \mathsf{Be_i} \\ \mathsf{Be_j} \\ \mathsf{Be_k} \\ 1 \\ 1 \\ \mathsf{h} \end{bmatrix}, \quad \mathbf{K}_{\mathsf{ijkh},2}^{(1)}\mathbf{h}_i^{(0)} = \begin{bmatrix} 1 \\ s_{\overline{k}(i),\overline{h}(i)-1} \\ a_{\overline{k}(i),\overline{h}(i)-1} \\ s_{\overline{k}(i),\overline{h}(i)} \\ -3\mathsf{B} \\ -\overline{h}(i) \\ 1 \end{bmatrix}, \quad \mathbf{V}_{\mathsf{ijkh},2}^{(1)}\mathbf{h}_i^{(0)} = \begin{bmatrix} \mathbf{0} \\ \mathsf{e}_{\mathsf{ijk}}^{N_\mathsf{h}} \\ \mathbf{0} \end{bmatrix},$$

$$\mathbf{Q}_{\mathsf{ijkh},3}^{(1)}\mathbf{h}_i^{(0)} = \begin{bmatrix} \mathsf{B}(v_i - 1) \\ \mathsf{Be_i} \\ \mathsf{Be_j} \\ \mathsf{Be_k} \\ 1 \\ 1 \\ -\mathsf{h} \end{bmatrix}, \quad \mathbf{K}_{\mathsf{ijkh},3}^{(1)}\mathbf{h}_i^{(0)} = \begin{bmatrix} 1 \\ s_{\overline{k}(i),\overline{h}(i)-1} \\ a_{\overline{k}(i),\overline{h}(i)-1} \\ s_{\overline{k}(i),\overline{h}(i)} \\ -3\mathsf{B} \\ \overline{h}(i) - 1 \\ 1 \end{bmatrix}, \quad \mathbf{V}_{\mathsf{ijkh},3}^{(1)}\mathbf{h}_i^{(0)} = -\begin{bmatrix} \mathbf{0} \\ \mathsf{e}_{\mathsf{ijk}}^{N_\mathsf{h}} \\ \mathbf{0} \end{bmatrix},$$

$$\mathbf{Q}_{\mathsf{ijkh},4}^{(1)}\mathbf{h}_i^{(0)} = \begin{bmatrix} \mathsf{B}(v_i - 1) \\ \mathsf{Be_i} \\ \mathsf{Be_j} \\ \mathsf{Be_k} \\ 1 \\ 1 \\ -\mathsf{h} \end{bmatrix}, \quad \mathbf{K}_{\mathsf{ijkh},4}^{(1)}\mathbf{h}_i^{(0)} = \begin{bmatrix} 1 \\ s_{\overline{k}(i),\overline{h}(i)-1} \\ a_{\overline{k}(i),\overline{h}(i)-1} \\ s_{\overline{k}(i),\overline{h}(i)} \\ -3\mathsf{B} \\ \overline{h}(i) - 2 \\ 1 \end{bmatrix}, \quad \mathbf{V}_{\mathsf{ijkh},4}^{(1)}\mathbf{h}_i^{(0)} = \begin{bmatrix} \mathbf{0} \\ \mathsf{e}_{\mathsf{ijk}}^{N_\mathsf{h}} \\ \mathbf{0} \end{bmatrix},$$

$$\mathbf{Q}_{\mathsf{ijkh},5}^{(1)}\mathbf{h}_i^{(0)} = \begin{bmatrix} \mathsf{B}(v_i - 1) \\ \mathsf{Be_i} \\ \mathsf{Be_j} \\ \mathsf{Be_k} \\ 1 \end{bmatrix}, \quad \mathbf{K}_{\mathsf{ijkh},5}^{(1)}\mathbf{h}_i^{(0)} = \begin{bmatrix} 1 \\ s_{\overline{k}(i),\overline{h}(i)-1} \\ a_{\overline{k}(i),\overline{h}(i)-1} \\ s_{\overline{k}(i),\overline{h}(i)} \\ -3\mathsf{B} \end{bmatrix}, \quad \mathbf{V}_{\mathsf{ijkh},5}^{(1)}\mathbf{h}_i^{(0)} = \begin{bmatrix} \mathbf{0} \\ \mathsf{e}_{\mathsf{ijk}}^{N_\mathsf{h}} \\ \mathbf{0} \end{bmatrix}.$$

Summing up the first five heads, we verify that such attention updates the token with $\mathbf{h}_i^a = \mathbf{0}$ and has the form

$$\mathbf{0} \to \mathbf{0} + \frac{1}{i}\widetilde{N}_{\mathsf{h}}(\mathsf{i},\mathsf{j},\mathsf{k})\mathbf{e}_{\mathsf{ijk}}^{N_h}$$

on $\mathbf{h}_i^d$, where $\widetilde{N}_{\mathsf{h}}(\mathsf{i},\mathsf{j},\mathsf{k})$ denote the number of visits to the state-action-next-state tuple $(\mathsf{i},\mathsf{j},\mathsf{k})$ at time step $\mathsf{h}$ before token $i$. For $\star \in [5]$, we choose $\mathbf{V}_{\mathsf{ijkh},\star+5}^{(1)} = -\mathbf{V}_{\mathsf{ijkh},\star+5}^{(1)}$ and $\mathbf{Q}_{\mathsf{ijkh},\star+5}^{(1)}, \mathbf{K}_{\mathsf{ijkh},\star+5}^{(1)}$ be such that

$$\mathbf{Q}_{\mathsf{ijkh},\star+5}^{(1)}\mathbf{h}_i^{(0)} = \begin{bmatrix} \mathbf{Q}_{\mathsf{ijkh},\star}^{(1)}\mathbf{h}_i^{(0)} \\ \mathsf{B} \\ -\overline{k}(i) \end{bmatrix}, \quad \mathbf{K}_{\mathsf{ijkh},\star+5}^{(1)}\mathbf{h}_i^{(0)} = \begin{bmatrix} \mathbf{K}_{\mathsf{ijkh},\star}^{(1)}\mathbf{h}_i^{(0)} \\ \overline{k}(i) \\ \mathsf{B} \end{bmatrix}$$

which adds positional embedding about the current episode $\overline{k}(i)$. We verify that summing up the sixth to the tenth heads gives the update

$$\mathbf{0} \to \mathbf{0} + \frac{1}{i}(N_{\mathsf{h}}(\mathsf{i},\mathsf{j},\mathsf{k}) - \widetilde{N}_{\mathsf{h}}(\mathsf{i},\mathsf{j},\mathsf{k}))\mathbf{e}_{\mathsf{ijk}}^{N_h}$$

on $\mathbf{h}_i^d$ for $i \leq 2(t-1) + k$ with $\mathbf{h}_i^a = \mathbf{0}$. Therefore, combining all the heads together we have the update

$$\mathbf{0} \to \mathbf{0} + \frac{1}{i}N_{\mathsf{h}}(\mathsf{i},\mathsf{j},\mathsf{k})\mathbf{e}_{\mathsf{ijk}}^{N_h} \quad \text{for all } \mathsf{i},\mathsf{k} \in [S], \mathsf{j} \in [A], \mathsf{h} \in [H]$$

on $\mathbf{h}_i^d$ for $i \leq 2(t-1) + k$ with $\mathbf{h}_i^a = \mathbf{0}$, in particular when $i = 2(t-1) + k$. Moreover, notice that the matrices $\{\mathbf{Q}_{\mathsf{ijkh},s}^{(1)}\}_{s=1}^{10}, \{\mathbf{K}_{\mathsf{ijkh},s}^{(1)}\}_{s=1}^{10}$ can be constructed with the operator norm less than $10\mathsf{B} = 10H$, and $\{\mathbf{V}_{\mathsf{ijkh},s}^{(1)}\}_{s=1}^{10}$ with the operator norm equals 1.

Following a similar construction, we can also compute $N_{\mathsf{h}}(s,a), N_{\mathsf{h}}(s,a)r_{\mathsf{h}}(s,a)$ for all $\mathsf{h}, s, a, s'$ on different supports of coordinates in $\mathbf{h}_i^d$ via adding additional $M = \mathrm{O}(HSA)$ heads to the attention-only layer.

Next, we construct the second attention layer to multiply the token vector by the index number $i$ as in the proof of Step 1a. The construction is similar to that in Eq. (26) and we omit it here. Moreover, note that Step 1b can be implemented with the embedding dimension $D \leq \mathrm{O}(HS^2A)$ as we need $\mathrm{O}(1)$ dimensions for each quadruple $(\mathsf{i},\mathsf{j},\mathsf{k},\mathsf{h})$. Combining Step 1a, 1b concludes the proof of Step 1.

**Proof of Step 2** After Step 1, for the current token $i = 2(t-1) + k$, we have $N_{\mathsf{h}}(s,a,s'), r_{\mathsf{h}}(s,a), N_{\mathsf{h}}(s,a), N_{\mathsf{h}}(s,a)r_{\mathsf{h}}(s,a)$ lie in $\mathbf{h}_i^d$ for all $\mathsf{h} \in [H]$. Given these vectors that store the number of visits and rewards, note that

$$r_{\mathsf{h}}(s,a) = \frac{N_{\mathsf{h}}(s,a)r_{\mathsf{h}}(s,a)}{N_{\mathsf{h}}(s,a) \vee 1}, \quad \text{when } N_{\mathsf{h}}(s,a) \geq 1,$$

$$b_{\mathsf{h}}(s,a) = 2H\sqrt{\frac{\log(SAT/\delta)}{N_{\mathsf{h}}(s,a) \vee 1}},$$

$$\widehat{P}_{\mathsf{h}}(s,a,s') = \frac{N_{\mathsf{h}}(s,a,s')}{N_{\mathsf{h}}(s,a) \vee 1}.$$

Therefore, we may compute $\widehat{P}_{\mathsf{h}}, b_{\mathsf{h}}$ via using a transformer layer to implement the functions $f_1(x,y) = \frac{x}{y \vee 1}, f_2(y) = 2H\sqrt{\frac{\log(SAT/\delta)}{y \vee 1}}, f_3(x,y) = \frac{x}{y \vee 1} + H\mathbf{1}_{y=0}$ for $x, y \in \{0\} \cup [K]$. We demonstrate the computation of $\widehat{P}_{\mathsf{h}}(s,a,s')$ (i.e., the computation of $f_1(x,y)$) here. We start with constructing an attention layer with $M = \mathrm{O}(HS^2A)$ heads such that it implements $x \mapsto x^2$ for $x = N_{\mathsf{h}}(s,a,s'), N_{\mathsf{h}}(s,a)$. For $N_{\mathsf{h}}(s,a,s')$, this can be done by choosing $\|\mathbf{Q}_{\mathsf{ijkh}}^{(1)}\|_{\mathrm{op}} \leq K, \|\mathbf{K}_{\mathsf{ijkh}}^{(1)}\|_{\mathrm{op}} \leq K, \|\mathbf{V}_{\mathsf{ijkh}}^{(1)}\|_{\mathrm{op}} = 1$ such that

$$\mathbf{Q}_{\mathsf{ijkh}}^{(1)}\mathbf{h}_i^{(0)} = \begin{bmatrix} K \\ -i \\ N_{\mathsf{h}}(\mathbf{e}_{\mathsf{i}},\mathbf{e}_{\mathsf{j}},\mathbf{e}_{\mathsf{k}}) \end{bmatrix}, \quad \mathbf{K}_{\mathsf{ijkh}}^{(1)}\mathbf{h}_j^{(0)} = \begin{bmatrix} j \\ K \\ N_{\mathsf{h}}(\mathbf{e}_{\mathsf{i}},\mathbf{e}_{\mathsf{j}},\mathbf{e}_{\mathsf{k}}) \end{bmatrix}, \quad \mathbf{V}_{\mathsf{ijkh}}^{(1)}\mathbf{h}_j^{(0)} = \begin{bmatrix} \mathbf{0} \\ j \\ \mathbf{0} \end{bmatrix},$$

where $\mathbf{e_i}, \mathbf{e_k}$ denote the $i, j$-th states and $\mathbf{e}_{sj}$ denotes the $k$-th action. Similarly, we can construct $HSA$ additional heads to compute $N_{\mathsf{h}}(s, a)^2$ for all possible $s, a$.

Next, we compute the exact values of $\widehat{P}(s, a, s')$ using an MLP layer. Namely, we construct $\mathbf{W}_1^{(1)} = \mathbf{W}_{12}^{(1)}\mathbf{W}_{11}^{(1)}, \mathbf{W}_2^{(1)} = \mathbf{W}_{23}^{(1)}\mathbf{W}_{22}^{(1)}\mathbf{W}_{21}^{(1)}$ such that for all $\mathsf{h}, s, a, s'$, on the corresponding vector component we have

$$
\mathbf{W}_{11}^{(1)}\mathbf{h}_i^{(0)} = \begin{bmatrix} 1 \\ N_{\mathsf{h}}(s, a, s')^2 \\ \vdots \\ (N_{\mathsf{h}}(s, a, s') - K)^2 \\ N_{\mathsf{h}}(s, a)^2 \\ \vdots \\ (N_{\mathsf{h}}(s, a) - K)^2 \end{bmatrix} = \begin{bmatrix} 1 \\ N_{\mathsf{h}}(s, a, s')^2 \\ \vdots \\ N_{\mathsf{h}}(s, a, s')^2 + K^2 - 2KN_{\mathsf{h}}(s, a, s') \\ N_{\mathsf{h}}(s, a)^2 \\ \vdots \\ N_{\mathsf{h}}(s, a)^2 + K^2 - 2KN_{\mathsf{h}}(s, a) \end{bmatrix},
$$

$$
\mathbf{W}_{12}^{(1)}\mathbf{W}_{11}^{(1)}\mathbf{h}_i^{(0)} = \begin{bmatrix} 1 - N_{\mathsf{h}}(s, a, s')^2 - N_{\mathsf{h}}(s, a)^2 \\ \vdots \\ 1 - (N_{\mathsf{h}}(s, a, s') - x)^2 - (N_{\mathsf{h}}(s, a) - y)^2 \\ \vdots \\ 1 - (N_{\mathsf{h}}(s, a, s') - K)^2 - (N_{\mathsf{h}}(s, a) - K)^2 \end{bmatrix},
$$

where $x, y \in \{0\} \cup [K]$. Moreover, we construct $\mathbf{W}_{21}^{(1)}$ so that on the entries corresponding to $\mathsf{h}, s, a, s'$ it implements

$$
\mathbf{W}_2^{(1)}\sigma(\mathbf{W}_1^{(1)}\mathbf{h}_i^{(0)}) = \left[ \sum_{x,y=0}^{K} \sigma(1 - (N_{\mathsf{h}}(s, a, s') - x)^2 - (N_{\mathsf{h}}(s, a) - y)^2) \cdot \frac{x}{y \vee 1} \right] = \left[ \frac{N_{\mathsf{h}}(s, a, s')}{N_{\mathsf{h}}(s, a) \vee 1} \cdot \right]
$$

It can be verified that we can find such $\mathbf{W}_1^{(1)}, \mathbf{W}_2^{(1)}$ with

$$
\|\mathbf{W}_1^{(1)}\|_{\mathrm{op}} \le \|\mathbf{W}_{11}^{(1)}\|_{\mathrm{op}}\|\mathbf{W}_{12}^{(1)}\|_{\mathrm{op}} \le \mathrm{O}(K^2) \cdot \mathrm{O}(K) = \mathrm{O}(K^3),
$$

$\|\mathbf{W}_2^{(1)}\|_{\mathrm{op}} \le \mathrm{O}(K)$, and the number of hidden neurons $D' = \mathrm{O}(K^2HS^2A)$. Simlarly, we can compute $f_2(\cdot)$ (or $f_3(\cdot)$) exactly following the same construction but with a different $\mathbf{W}_2^{(1)}$ that records all possible values of $f_2(\cdot)$ (or $f_3(\cdot)$). Combining the upper bounds on the operator norm of the weight matrices, we further have $\|\boldsymbol{\theta}\| \le \tilde{\mathcal{O}}(HS^2A + K^3 + KH)$.

**Proof of Step 3** Given $\widehat{V}_{H+1} = \widetilde{V}_{H+1} = \mathbf{0}$, we show the there exists an transformer with

$$
L = 2, \quad \max_{\ell \in [L]} M^{(\ell)} \le 2SA, \quad \max_{\ell \in [L]} D'^{(\ell)} \le 3SA, \quad \|\boldsymbol{\theta}\| \le \mathrm{O}(H + SA)
$$

that implements one step of value iteration

$$
\widehat{Q}_{\mathsf{h}}(s, a) = \max\{\min\{H, r_{\mathsf{h}}(s, a) + b_{\mathsf{h}}(s, a) + \sum_{s' \in \mathcal{S}} \widehat{P}_{\mathsf{h}}(s' \mid s, a)\widehat{V}_{\mathsf{h}+1}(s')\}, 0\},
$$

$$
\widehat{V}_{\mathsf{h}}(s) = \max_{a \in \mathcal{A}} \widehat{Q}_{\mathsf{h}}(s, a)
$$

for some $\mathsf{h} \in [H]$. Namely, we start with constructing an-attention layer with $M = 2SA$ and $\{\mathbf{Q}_{ijh,s}^{(1)}\}_{s=1}^2, \{\mathbf{K}_{ijh,s}^{(1)}\}_{s=1}^2, \{\mathbf{V}_{ijh,s}^{(1)}\}_{s=1}^2$ such that for all $i \le 2(t - 1) + k$

$$
\mathbf{Q}_{ij,1}^{(1)}\mathbf{h}_i^{(0)} = \begin{bmatrix} \mathsf{B} \\ -i \\ \widehat{V}_{\mathsf{h}+1}(\cdot) \end{bmatrix}, \quad \mathbf{K}_{ij,1}^{(1)}\mathbf{h}_i^{(0)} = \begin{bmatrix} i \\ \mathsf{B} \\ \widehat{P}_{\mathsf{h}+1}(\cdot|s, a) \end{bmatrix}, \quad \mathbf{V}_{ij,1}^{(1)}\mathbf{h}_i^{(0)} = \begin{bmatrix} \mathbf{0} \\ i\mathbf{e}_{ij}^{Q_{\mathsf{h}}} \\ \mathbf{0}, \end{bmatrix}
$$

$$
\mathbf{Q}_{ij,2}^{(1)}\mathbf{h}_i^{(0)} = \begin{bmatrix} \mathsf{B} \\ -i \\ -\widehat{V}_{\mathsf{h}+1}(\cdot) \end{bmatrix}, \quad \mathbf{K}_{ij,2}^{(1)} = \mathbf{K}_{ij,1}^{(1)}, \quad \mathbf{V}_{ij,2}^{(1)} = -\mathbf{V}_{ij,2}^{(1)}
$$

where $\mathsf{B} = 3H$ and $\mathbf{e}_{ij}^{Q_\mathsf{h}} \in \mathbb{R}^{SA}$ is a vector supported on some coordinates in $\mathbf{h}_i^d$ reserved for $\widehat{Q}_\mathsf{h}$. Moreover, we have $\|\mathbf{Q}_{ijh,s}^{(1)}\|_{\mathrm{op}}, \|\mathbf{K}_{ijh,s}^{(1)}\|_{\mathrm{op}} \leq \mathsf{B}$, $\|\mathbf{V}_{ijh,s}^{(1)}\|_{\mathrm{op}} = 1$. Since

$$\left| \left\langle \widehat{V}_{\mathsf{h}+1}(\cdot), \widehat{P}_{\mathsf{h}+1}(\cdot | s, a) \right\rangle \right| \leq \left\| \widehat{V}_{\mathsf{h}+1}(\cdot) \right\|_\infty \cdot \left\| \widehat{P}_{\mathsf{h}+1}(\cdot | s, a) \right\|_1 \leq H$$

as $\widehat{V}_{\mathsf{h}+1}(s) \in [0, H]$ and $\left\| \widehat{P}_{\mathsf{h}+1}(\cdot | s, a) \right\|_1 = 1$, it follows that summing up two heads gives the update for $i \leq 2(t-1) + k$

$$\mathbf{0} \mapsto \mathbf{0} + \left[ \sigma(\left\langle \mathbf{Q}_{ij,1}^{(1)} \mathbf{h}_i^{(0)}, \mathbf{K}_{ij,1}^{(1)} \mathbf{h}_j^{(0)} \right\rangle) - \sigma(\left\langle \mathbf{Q}_{ij,1}^{(1)} \mathbf{h}_i^{(0)}, \mathbf{K}_{ij,1}^{(1)} \mathbf{h}_j^{(0)} \right\rangle) \right] \mathbf{e}_{ij}^{Q_\mathsf{h}}$$
$$= \left\langle \mathbf{Q}_{ij,1}^{(1)} \mathbf{h}_i^{(0)}, \mathbf{K}_{ij,1}^{(1)} \mathbf{h}_j^{(0)} \right\rangle \mathbf{e}_{ij}^{Q_\mathsf{h}}.$$

Denote the resulting token vector by $\mathbf{h}_i^{(1)}$. Moreover, we can construct a two-layer MLP with

$$\|\mathbf{W}_1^{(1)}\|_{\mathrm{op}} = \mathrm{O}(H), \quad \|\mathbf{W}_2^{(1)}\|_{\mathrm{op}} \leq 3, \quad D' = 3SA$$

such that for any state-action pair $(s, a) \in \mathcal{S} \times \mathcal{A}$ on the corresponding coordinates

$$\mathbf{W}_1^{(1)} \mathbf{h}_i^{(1)} = \begin{bmatrix} \vdots \\ -[r_\mathsf{h}(s, a) + b_\mathsf{h}(s, a) + \sum_{s' \in \mathcal{S}} \widehat{P}_\mathsf{h}(s' | s, a) \widehat{V}_{\mathsf{h}+1}(s')] \\ r_\mathsf{h}(s, a) + b_\mathsf{h}(s, a) + \sum_{s' \in \mathcal{S}} \widehat{P}_\mathsf{h}(s' | s, a) \widehat{V}_{\mathsf{h}+1}(s') - H \\ r_\mathsf{h}(s, a) + b_\mathsf{h}(s, a) + \sum_{s' \in \mathcal{S}} \widehat{P}_\mathsf{h}(s' | s, a) \widehat{V}_{\mathsf{h}+1}(s') \\ \vdots \end{bmatrix}$$

and
$$\mathbf{W}_2^{(1)} \sigma(\mathbf{W}_1^{(1)} \mathbf{h}_i^{(1)}) = \sigma(-[r_\mathsf{h}(s, a) + b_\mathsf{h}(s, a) + \sum_{s' \in \mathcal{S}} \widehat{P}_\mathsf{h}(s' | s, a) \widehat{V}_{\mathsf{h}+1}(s')])$$
$$- \sigma(r_\mathsf{h}(s, a) + b_\mathsf{h}(s, a) + \sum_{s' \in \mathcal{S}} \widehat{P}_\mathsf{h}(s' | s, a) \widehat{V}_{\mathsf{h}+1}(s') - H)$$
$$+ \sigma(r_\mathsf{h}(s, a) + b_\mathsf{h}(s, a) + \sum_{s' \in \mathcal{S}} \widehat{P}_\mathsf{h}(s' | s, a) \widehat{V}_{\mathsf{h}+1}(s'))$$
$$= \max\{\min\{H, r_\mathsf{h}(s, a) + b_\mathsf{h}(s, a) + \sum_{s' \in \mathcal{S}} \widehat{P}_\mathsf{h}(s' | s, a) \widehat{V}_{\mathsf{h}+1}(s')\}, 0\} = \widehat{Q}_h(s, a).$$

Denote the resulting token vector by $\mathbf{h}_i^{(2)}$. Next, we construct a second MLP layer with

$$\|\mathbf{W}_1^{(2)}\|_{\mathrm{op}} \leq 2, \quad \|\mathbf{W}_2^{(2)}\|_{\mathrm{op}} \leq \sqrt{A}, \quad D' = AS$$

such that for any $s \in \mathcal{S}$ on the corresponding coordinates we have

$$\mathbf{W}_1^{(2)} \mathbf{h}_i^{(2)} = \begin{bmatrix} \vdots \\ \widehat{Q}_\mathsf{h}(s, a_1) \\ \widehat{Q}_\mathsf{h}(s, a_2) - \widehat{Q}_\mathsf{h}(s, a_1) \\ \vdots \\ \widehat{Q}_\mathsf{h}(s, a_A) - \widehat{Q}_\mathsf{h}(s, a_{A-1}) \\ \vdots \end{bmatrix},$$

where $a_j$ denotes the $j$−th action, and

$$\mathbf{W}_2^{(2)} \sigma(\mathbf{W}_1^{(2)} \mathbf{h}_i^{(2)}) = \sigma(\widehat{Q}_\mathsf{h}(s, a_1)) + \sum_{j=2}^{A} \sigma(\widehat{Q}_\mathsf{h}(s, a_j) - \widehat{Q}_\mathsf{h}(s, a_{j-1}))$$
$$= \max_{a \in \mathcal{A}} \widehat{Q}_\mathsf{h}(s, a) = \widehat{V}_\mathsf{h}(s).$$

Using the upper bounds on the operator norm of the weight matrices, we further have $\|\boldsymbol{\theta}\| \leq \mathrm{O}(SA + H)$. Combining the steps concludes the construction in Step 3.

**Proof of Step 4**   we start with constructing an-attention layer with $M = 2HA$ and $\{\mathbf{Q}_{\mathsf{jh},s}^{(1)}\}_{s=1}^2$, $\{\mathbf{K}_{\mathsf{jh},s}^{(1)}\}_{s=1}^2$, $\{\mathbf{V}_{\mathsf{jh},s}^{(1)}\}_{s=1}^2$ such that for all the current token $i = 2(t-1) + k$ and $j \le i$

$$\mathbf{Q}_{\mathsf{jh},1}^{(1)}\mathbf{h}_i^{(0)} = \begin{bmatrix} s_{\overline{k}(i),\overline{h}(i)} \\ -i \\ \mathsf{B} \end{bmatrix}, \quad \mathbf{K}_{\mathsf{jh},1}^{(1)}\mathbf{h}_j^{(0)} = \begin{bmatrix} \widehat{Q}_{\mathsf{h}}(\cdot, a_j) \\ \mathsf{B} \\ j \end{bmatrix}, \quad \mathbf{V}_{\mathsf{jh},1}^{(1)}\mathbf{h}_i^{(0)} = \begin{bmatrix} \mathbf{0} \\ i\mathbf{e}_{\mathsf{jh}} \\ \mathbf{0}, \end{bmatrix}$$

$$\mathbf{Q}_{\mathsf{jh},2}^{(1)}\mathbf{h}_i^{(0)} = \begin{bmatrix} -s_{\overline{k}(i),\overline{h}(i)} \\ -i \\ \mathsf{B} \end{bmatrix}, \quad \mathbf{K}_{\mathsf{jh},2}^{(1)} = \mathbf{K}_{\mathsf{jh},1}^{(1)}, \quad \mathbf{V}_{\mathsf{jh},2}^{(1)} = -\mathbf{V}_{\mathsf{jh},1}^{(1)},$$

where we choose $\mathsf{B} = 2H$ and $\mathbf{V}_{\mathsf{jh},1}^{(1)}\mathbf{h}_i^{(0)}$ is a one-hot vector supported on some entry of $\mathbf{h}_i^d$. We verify that summing up the heads gives the update

$$\mathbf{0} \mapsto \widehat{Q}_{\mathsf{h}}(s_{k,h}, a_{\mathsf{j}})\mathbf{e}_{\mathsf{jh}}$$

for all $\mathsf{h} \in [H], \mathsf{j} \in [A]$. Moreover, we have $\|\mathbf{Q}_{\mathsf{jh},s}^{(1)}\|_{\mathrm{op}} \le 2H, \|\mathbf{K}_{\mathsf{jh},s}^{(1)}\|_{\mathrm{op}} \le 2H, \|\mathbf{V}_{\mathsf{jh},s}^{(1)}\|_{\mathrm{op}} \le 1$ for $s = 1, 2$. Through this attention-only layer, we extract the values $\widehat{Q}_{\mathsf{h}}(s_{k,h}, a_j)$ for all $\mathsf{h} \in [H]$ from the Q-function.

Similar to the proof of Step 1b, we construct a second attention-only layer with attention heads $\{\mathbf{Q}_{\mathsf{jh},s}^{(2)}\}_{s=1}^2, \{\mathbf{K}_{\mathsf{jh},s}^{(2)}\}_{s=1}^2, \{\mathbf{V}_{\mathsf{jh},s}^{(2)}\}_{s=1}^2$ that

$$\mathbf{Q}_{\mathsf{jh},1}^{(2)}\mathbf{h}_i^{(1)} = \begin{bmatrix} 1 \\ -\overline{h}(i) \\ -i \\ \mathsf{B} \end{bmatrix}, \quad \mathbf{K}_{\mathsf{jh},1}^{(2)}\mathbf{h}_j^{(1)} = \begin{bmatrix} \mathsf{h} \\ 1 \\ \mathsf{B} \\ j \end{bmatrix}, \quad \mathbf{V}_{\mathsf{jh},1}^{(2)}\mathbf{h}_i^{(1)} = -\begin{bmatrix} \mathbf{0} \\ \widehat{Q}_{\mathsf{h}}(s_{k,h}, a_{\mathsf{j}})\mathbf{e}_{\mathsf{j}} \\ \mathbf{0} \end{bmatrix},$$

$$\mathbf{Q}_{\mathsf{jh},2}^{(2)} = \mathbf{Q}_{\mathsf{jh},1}^{(2)}, \quad \mathbf{K}_{\mathsf{jh},2}^{(2)}\mathbf{h}_j^{(1)} = \begin{bmatrix} \mathsf{h} - 1 \\ 1 \\ \mathsf{B} \\ j \end{bmatrix}, \quad \mathbf{V}_{\mathsf{jh},2}^{(2)} = -\mathbf{V}_{\mathsf{jh},2}^{(2)},$$

where $\mathbf{V}_{\mathsf{jh},1}^{(2)}\mathbf{h}_i^{(1)}$ are supported on some entry of $\mathbf{h}_i^d$ for $s = 1, 2$. Summing up the heads gives the update

$$\mathbf{0} \mapsto -\sum_{s=\overline{h}(i)+1}^{H} \frac{1}{i}\widehat{Q}_s(s_{k,h}, a_{\mathsf{j}}).$$

Similarly, we can construct attention heads $\{\mathbf{Q}_{\mathsf{jh},s}^{(2)}\}_{s=3}^4, \{\mathbf{K}_{\mathsf{jh},s}^{(2)}\}_{s=3}^4, \{\mathbf{V}_{\mathsf{jh},s}^{(2)}\}_{s=3}^4$ that implements

$$\mathbf{0} \mapsto -\frac{1}{i}\sum_{s=1}^{\overline{h}(i)-1} \widehat{Q}_s(s_{k,h}, a_{\mathsf{j}}).$$

Moreover, we construct $\mathbf{Q}_{\mathsf{jh},5}^{(2)}, \mathbf{K}_{\mathsf{jh},5}^{(2)}, \mathbf{V}_{\mathsf{jh},5}^{(2)}$ with

$$\mathbf{Q}_{\mathsf{jh},5}^{(2)}\mathbf{h}_i^{(1)} = \begin{bmatrix} 1 \\ -i \\ \mathsf{B} \end{bmatrix}, \quad \mathbf{K}_{\mathsf{jh},5}^{(2)}\mathbf{h}_j^{(1)} = \begin{bmatrix} 1 \\ \mathsf{B} \\ j \end{bmatrix}, \quad \mathbf{V}_{\mathsf{jh},1}^{(2)}\mathbf{h}_i^{(1)} = \begin{bmatrix} \mathbf{0} \\ \widehat{Q}_{\mathsf{h}}(s_{k,h}, a_{\mathsf{j}})\mathbf{e}_{\mathsf{j}} \\ \mathbf{0}, \end{bmatrix}$$

that implements

$$\mathbf{0} \mapsto \frac{1}{i}\sum_{s=1}^{H} \widehat{Q}_s(s_{k,h}, a_{\mathsf{j}}).$$

Therefore, summing up the $M = 5HA$ heads we obtain the update

$$\mathbf{0}_A \mapsto \frac{1}{i}\widehat{Q}_h(s_{k,h}, \cdot).$$

Note that $\|\mathbf{Q}_{\mathsf{jh},s}^{(1)}\|_{\mathrm{op}} \leq 4H$, $\|\mathbf{K}_{\mathsf{jh},s}^{(1)}\|_{\mathrm{op}} \leq 4H$, $\|\mathbf{V}_{\mathsf{jh},s}^{(1)}\|_{\mathrm{op}} \leq 1$ for $s \in [5]$.

Finally, we apply an attention-only layer to implement the multiplication by a factor of $i/\tau$ using a similar construction as in Eq. (26) with $\|\mathbf{Q}_1^{(3)}\|_{\mathrm{op}} = \mathrm{O}(HK)$, $\|\mathbf{K}_1^{(3)}\|_{\mathrm{op}} = \mathrm{O}(HK)$, $\|\mathbf{V}_1^{(3)}\|_{\mathrm{op}} = \mathrm{O}(1/\tau)$, and assign the resulting vector $\widehat{Q}(s_{k,h}, \cdot)/\tau$ to $\mathbf{h}_i^c$. Combining the three attention-only layers completes Step 4.

### G.4 PROOF OF THEOREM 13

By Theorem 6 and 12, it suffices to show the regret of soft UCB-VI satisfies

$$\mathbb{E}[KV_{\mathsf{M}}(\pi^*) - \mathfrak{R}_{\mathsf{M},\mathsf{Alg}_{\mathrm{sUCBVI}(\tau)}}(T)] \leq \tilde{\mathcal{O}}(H^2\sqrt{SAK} + H^3S^2A)$$

for all MDP instances $\mathsf{M}$, where $\tau = 1/K$ and $\tilde{\mathcal{O}}(\cdot)$ hides logarithmic dependencies on $(H, K, S, A)$.

Throughout the proof, we may drop the dependence on $\mathsf{M}$ for notational simplicity when there is no confusion. For each episode $k \in [K]$, let $N_{\mathsf{h}}^k, \widehat{P}_{\mathsf{h}}^k, \widehat{Q}_{\mathsf{h}}^k, \widehat{V}_{\mathsf{h}}^k, b_{\mathsf{h}}^k$ denote the corresponding quantities $N_{\mathsf{h}}, \widehat{P}_{\mathsf{h}}, \widehat{Q}_{\mathsf{h}}, \widehat{V}_{\mathsf{h}}, b_{\mathsf{h}}$ introduced in UCB-VI (see Section G.2).

For a policy $\pi$ and time step $\mathsf{h} \in [H]$, we define the Q-function $Q_{\mathsf{h}}^\pi$ and the value function $V_{\mathsf{h}}^\pi$

$$Q_{\mathsf{h}}^\pi(s,a) := \mathbb{E}[\sum_{t=\mathsf{h}}^{H} r(s_t, a_t) \mid s_{\mathsf{h}} = s, a_{\mathsf{h}} = a, \pi],$$

$$V_{\mathsf{h}}^\pi(s) := \mathbb{E}[\sum_{t=\mathsf{h}}^{H} r(s_t, a) \mid s_{\mathsf{h}} = s, \pi].$$

We use $\pi^k = (\pi_1^k, \ldots, \pi_H^k), \pi_{\mathsf{sm}}^k = (\pi_{\mathsf{sm},1}^k, \ldots, \pi_{\mathsf{sm},\mathsf{h}}^k, \ldots \pi_{\mathsf{sm},H}^k)$ to denote the policies given by UCB-VI and soft UCB-VI in the $k$-th episode, respectively. Note that we have $V_{\mathsf{M}}(\pi) = \mathbb{E}_{s \sim \mu_1}[V_{\mathsf{h}}^\pi(s)]$ and cumulative the regret

$$\mathbb{E}[KV_{\mathsf{M}}(\pi^*) - \mathfrak{R}_{\mathsf{M},\mathsf{Alg}_{\mathrm{sUCBVI}}(\tau)}(T)]] = \mathbb{E}\Big[\sum_{k=1}^{K}[V_1^{\pi^*}(s_{k,1}) - V_1^{\pi_{\mathsf{sm}}^k}(s_{k,1})]\Big]$$

where the expectation is taken over the collected data

$$D_T = \{(s_{k,\mathsf{h}}, a_{k,\mathsf{h}}, r_{k,\mathsf{h}})\}_{k \in [K], \mathsf{h} \in [H]} \sim \mathbb{P}_{\mathsf{M}}^{\mathrm{sUCBVI}(\tau)}.$$

For any function $f = f(s, a)$, we abuse the notation $f(s, \pi(\cdot)) := \mathbb{E}_{a \sim \pi}[f(s, a)]$. Lastly, we define

$$\varepsilon_{\mathrm{sfmax}} = \max_{k \in [K], \mathsf{h} \in [H], s \in \mathcal{S}}[\widehat{Q}_{\mathsf{h}}^k(s, \pi_{\mathsf{h}}^k(\cdot)) - \widehat{Q}_{\mathsf{h}}^k(s, \pi_{\mathsf{sm},\mathsf{h}}^k(\cdot))].$$

We claim the following which we will prove later

$$\varepsilon_{\mathrm{sfmax}} \leq A\tau. \tag{27}$$

The proof follows from similar arguments as in the proof of Theorem 1 in Azar et al. (2017) (see also Theorem 7.6 in Agarwal et al. (2019)). Hence we only provide a sketch of proof here. First, from the proof of Theorem 7.6 in Agarwal et al. (2019), it can be shown that

$$\widehat{V}_{\mathsf{h}}^k(s) \geq V_{\mathsf{h}}^{\pi^*}(s)$$

for any $k, \mathsf{h}, s$ with probability at least $1-\delta$. Thus with probability at least $1-\delta$ for all $\mathsf{h} \in [H], k \in [K]$

$$V_{\mathsf{h}}^{\pi^*}(s_{k,\mathsf{h}}) - V_{\mathsf{h}}^{\pi_{\mathsf{sm}}^k}(s_{k,\mathsf{h}})$$
$$\leq \widehat{V}_{\mathsf{h}}^k(s_{k,\mathsf{h}}) - V_{\mathsf{h}}^{\pi_{\mathsf{sm}}^k}(s_{k,\mathsf{h}})$$
$$= \widehat{Q}_{\mathsf{h}}^k(s_{k,\mathsf{h}}, \pi_{\mathsf{h}}^k(\cdot)) - \widehat{Q}_{\mathsf{h}}^k(s_{k,\mathsf{h}}, \pi_{\mathsf{sm},\mathsf{h}}^k(\cdot)) + \widehat{Q}_{\mathsf{h}}^k(s_{k,\mathsf{h}}, \pi_{\mathsf{sm},\mathsf{h}}^k(\cdot)) - Q_{\mathsf{h}}^{\pi_{\mathsf{sm}}^k}(s_{k,\mathsf{h}}, \pi_{\mathsf{sm},\mathsf{h}}^k(\cdot))$$
$$\leq \widehat{Q}_{\mathsf{h}}^k(s_{k,\mathsf{h}}, \pi_{\mathsf{sm},\mathsf{h}}^k(\cdot)) - Q_{\mathsf{h}}^{\pi_{\mathsf{sm}}^k}(s_{k,\mathsf{h}}, \pi_{\mathsf{sm},\mathsf{h}}^k(\cdot)) + \varepsilon_{\mathrm{sfmax}}$$

$$= \widehat{Q}_\mathsf{h}^k(s_{k,\mathsf{h}}, a_{k,\mathsf{h}}) - Q_\mathsf{h}^{\pi_\mathsf{sm}^k}(s_{k,\mathsf{h}}, a_{k,\mathsf{h}}) + \mathrm{MD}_{k,\mathsf{h}}^{(1)} + \varepsilon_\mathrm{sfmax},$$

where the first equality uses $\widehat{V}_\mathsf{h}^k(s_{k,\mathsf{h}}) = \arg\max_a \widehat{Q}_\mathsf{h}^k(s_{k,\mathsf{h}}, a_{k,\mathsf{h}}) = \widehat{Q}_\mathsf{h}^k(s_{k,\mathsf{h}}, \pi_\mathsf{h}^k(\cdot))$, and in the last line

$$\mathrm{MD}_{k,\mathsf{h}}^{(1)} := [\widehat{Q}_\mathsf{h}^k(s_{k,\mathsf{h}}, \pi_{\mathsf{sm},\mathsf{h}}^k(\cdot)) - Q_\mathsf{h}^{\pi_\mathsf{sm}^k}(s_{k,\mathsf{h}}, \pi_{\mathsf{sm},\mathsf{h}}^k(\cdot))] - [\widehat{Q}_\mathsf{h}^k(s_{k,\mathsf{h}}, a_{k,\mathsf{h}}) - Q_\mathsf{h}^{\pi_\mathsf{sm}^k}(s_{k,\mathsf{h}}, a_{k,\mathsf{h}})].$$

Note that for any fixed $\mathsf{h} \in [H]$, $\{\mathrm{MD}_{k,\mathsf{h}}^{(1)}\}_{k=1}^K$ is a bounded martingale difference sequence. Following the proof of Theorem 7.6 in Agarwal et al. (2019), we further have

$$V_\mathsf{h}^{\pi^*}(s_{k,\mathsf{h}}) - V_\mathsf{h}^{\pi_\mathsf{sm}^k}(s_{k,\mathsf{h}})$$
$$\leq \widehat{Q}_\mathsf{h}^k(s_{k,\mathsf{h}}, a_{k,\mathsf{h}}) - Q_\mathsf{h}^{\pi_\mathsf{sm}^k}(s_{k,\mathsf{h}}, a_{k,\mathsf{h}}) + \mathrm{MD}_{k,\mathsf{h}}^{(1)} + \varepsilon_\mathrm{sfmax}$$
$$\leq \left(1 + \frac{1}{H}\right)\left[\widehat{V}_{\mathsf{h}+1}^k(s_{k,\mathsf{h}+1}) - V_{\mathsf{h}+1}^{\pi_\mathsf{sm}^k}(s_{k,\mathsf{h}+1})\right] + 2b_\mathsf{h}^k(s_{k,\mathsf{h}}, a_{k,\mathsf{h}})$$
$$+ \frac{c_0 L_0 H^2 S}{N_\mathsf{h}^k(s_{k,\mathsf{h}}, a_{k,\mathsf{h}})} + \mathrm{MD}_{k,\mathsf{h}}^{(2)} + \mathrm{MD}_{k,\mathsf{h}}^{(1)} + \varepsilon_\mathrm{sfmax},$$

with probability at least $1 - c\delta$ for some universal constant $c > 0$, where $L_0 = \log(SAKH/\delta)$, $c_0 > 0$ is some universal constant and

$$\mathrm{MD}_{k,\mathsf{h}}^{(2)} := P_\mathsf{h}(\cdot \mid s_{k,\mathsf{h}}, a_{k,\mathsf{h}}) \cdot (V_{\mathsf{h}+1}^{\pi^*} - V_{\mathsf{h}+1}^{\pi_\mathsf{sm}^k}) - (V_{\mathsf{h}+1}^{\pi^*}(s_{k,\mathsf{h}+1}) - V_{\mathsf{h}+1}^{\pi_\mathsf{sm}^k}(s_{k,\mathsf{h}+1}))$$

is a bounded martingale difference sequence for any fixed $\mathsf{h} \in [H]$. Using the recursive formula and the fact that $(1 + 1/H)^H < e$, we obtain

$$\mathbb{E}\Big[\sum_{k=1}^K [V_1^{\pi^*}(s_{k,1}) - V_1^{\pi_\mathsf{sm}^k}(s_{k,1})]\Big]$$
$$\leq c\mathbb{E}\left[\sum_{k=1}^K \sum_{\mathsf{h}=1}^H \left[2b_\mathsf{h}^k(s_{k,\mathsf{h}}, a_{k,\mathsf{h}}) + \frac{c_0 L_0 H^2 S}{N_\mathsf{h}^k(s_{k,\mathsf{h}}, a_{k,\mathsf{h}})} + \mathrm{MD}_{k,\mathsf{h}}^{(2)} + \mathrm{MD}_{k,\mathsf{h}}^{(1)}\right]\right] + \mathbb{E}[K\sum_{\mathsf{h}=0}^{H-1}(1+\frac{1}{H})^\mathsf{h}\varepsilon_\mathrm{sfmax}]$$
$$\leq c\mathbb{E}\left[\sum_{k=1}^K \sum_{\mathsf{h}=1}^H \left[2b_\mathsf{h}^k(s_{k,\mathsf{h}}, a_{k,\mathsf{h}}) + \frac{c_0 L_0 H^2 S}{N_\mathsf{h}^k(s_{k,\mathsf{h}}, a_{k,\mathsf{h}})} + \mathrm{MD}_{k,\mathsf{h}}^{(2)} + \mathrm{MD}_{k,\mathsf{h}}^{(1)}\right]\right] + cKHA\tau$$
$$\leq \tilde{\mathcal{O}}(H^2\sqrt{SAK} + H^3 S^2 A) + cKHA\tau$$
$$\leq \tilde{\mathcal{O}}(H^2\sqrt{SAK} + H^3 S^2 A),$$

where $c > 0$ is some universal constant, $\tilde{\mathcal{O}}(\cdot)$ hides logarithmic dependencies on $(H, K, S, A)$, and the last line follows again from the proof of Theorem 7.6 in Agarwal et al. (2019), and the assumption that $\tau = 1/K$. We omit the detailed derivations here as they are similar to those in Azar et al. (2017); Agarwal et al. (2019). Therefore, we conclude the proof of the first part of Theorem 13. Moreover, the second part of Theorem 13 (i.e., the upper bound on $\log\mathcal{N}_\Theta$) follows immediately from Lemma 16 and Eq. (10).

**Proof of Eq.** (27) By definition of $\widehat{Q}_\mathsf{h}^k$ and $\pi_\mathsf{h}^k, \pi_{\mathsf{sm},\mathsf{h}}^k$, we have

$$\widehat{Q}_\mathsf{h}^k(s, \pi_\mathsf{h}^k(\cdot)) - \widehat{Q}_\mathsf{h}^k(s, \pi_{\mathsf{sm},\mathsf{h}}^k(\cdot)) = \max_a \widehat{Q}_\mathsf{h}^k(s, a) - \sum_a \frac{\exp(\widehat{Q}_\mathsf{h}^k(s, a)/\tau)}{\sum_a \exp(\widehat{Q}_\mathsf{h}^k(s, a)/\tau)} \cdot \widehat{Q}_\mathsf{h}^k(s, a)$$
$$= \sum_a \frac{\exp(\widehat{Q}_\mathsf{h}^k(s, a)/\tau)}{\sum_a \exp(\widehat{Q}_\mathsf{h}^k(s, a)/\tau)} \cdot [\max_a \widehat{Q}_\mathsf{h}^k(s, a) - \widehat{Q}_\mathsf{h}^k(s, a)]$$
$$\leq \sum_a \frac{\exp(\widehat{Q}_\mathsf{h}^k(s, a)/\tau)}{\exp(\max_a \widehat{Q}_\mathsf{h}^k(s, a)/\tau)} \cdot [\max_a \widehat{Q}_\mathsf{h}^k(s, a) - \widehat{Q}_\mathsf{h}^k(s, a)]$$
$$\leq A \cdot [\sup_{t\geq 0} t\exp(-t/\tau)] \leq A\tau.$$

