# OpenReview forum: "Transformers as Decision Makers: Provable In-Context Reinforcement Learning via Supervised Pretraining"
_ICLR.cc/2024/Conference — ICLR 2024 poster_

### Official Review · Reviewer_E6sG · 2023-10-30

**Soundness:** 3 good
**Presentation:** 3 good
**Contribution:** 3 good
**Rating:** 6
**Confidence:** 3

**Summary:**

The paper examines the theoretical basis of in-context reinforcement learning (ICRL), specifically focusing on the supervised pretraining approach. This analysis builds upon previous studies [1, 2] and sheds light on the specific capabilities of algorithms trained through supervised pretraining. Theoretical findings demonstrate that transformers can approximate well-known algorithms such as LinUCB for bandit problems and UCB-VI for tabular MDPs under certain conditions. Empirical simulations further confirm the authors' findings that trained transformers effectively imitate bandit algorithms.

[1] Laskin M, Wang L, Oh J, et al. In-context reinforcement learning with algorithm distillation[J]. arXiv preprint arXiv:2210.14215, 2022.

[2] Lee J N, Xie A, Pacchiano A, et al. Supervised Pretraining Can Learn In-Context Reinforcement Learning[J]. arXiv preprint arXiv:2306.14892, 2023.

**Strengths:**

1. This work offers a timely investigation into the theoretical understanding of ICRL. Based on previous findings, the proposed framework advances this line of research by providing valuable analysis with guarantees on sample complexity.
2. The analysis is overall well-executed.  It is good to have a clear and unified overview of the previous work.
3. The paper is easy to follow.

**Weaknesses:**

1. Although this work theoretically covers tabular MDPs, it lacks the necessary experimental results to support the analysis.
2. I am unsure if the proposed framework can offer insights on improving pretraining approaches for ICRL. I recommend that the authors address this in their manuscript.

**Questions:**

1. Can you provide further explanation on the importance of calculating loss on the entire trajectory in Equation 3?

---

> ### Author Response · Authors · 2023-11-21
> **Response to Reviewer E6sG**
>
> We sincerely appreciate the reviewer's detailed review and insightful feedback on our paper. In the following, we hope to address some points raised in the review.
>
> > “I am unsure if the proposed framework can offer insights on improving pretraining approaches for ICRL. I recommend that the authors address this in their manuscript.”
>
> Thanks for the suggestion. In Appendix A.1 in our revision, we have included a discussion of some practical implications of our results, as well as potential ways to further improve ICRL practice.
>
> As examples, our results highlight that the training labels (expert actions) in the offline dataset matter a lot. When it is itself a proper algorithm that only depends on the past observations, we can learn ${\sf Alg}_E$ (cf. Theorem 9). By contrast, when ${\sf Alg}_E$ is the ground truth optimal action $a^\star$ (which involves knowledge of the underlying MDP), we can learn the posterior average of this algorithm given past observations, which corresponds to the Thompson sampling algorithm as in Decision-Pretrained Transformers (cf. Theorem 11).
>
> > “Can you provide further explanation on the importance of calculating loss on the entire trajectory in Equation 3?”
>
> The reason we calculate the loss over the entire trajectory is that we want the learned algorithm to imitate the expert algorithm at every time step. Unlike in-context learning for supervised learning problems (Garg et al. 2022) in which the loss only needs to be calculated on a new test data point since an i.i.d. assumption on the in-context data is assumed, in ICRL the actions chosen by the expert algorithm may depend on the time $t$ and previous actions and rewards. Therefore, to fully imitate the expert at every time step, the loss needs to be calculated on the entire trajectory.

---

> > ### Author Response · Authors · 2023-11-22
> > **Follow-up**
> >
> > Thank you again for your review! As the author-reviewer discussion period ends today, please let us know if our response has adequately addressed your questions and concerns, and we'd be happy to answer if you have additional questions.

---

### Official Review · Reviewer_fr59 · 2023-10-30

**Soundness:** 4 excellent
**Presentation:** 3 good
**Contribution:** 3 good
**Rating:** 8
**Confidence:** 4

**Summary:**

**Update after rebuttal:** I consider all issues raised by all reviewers sufficiently addressed by the authors. I stand by my original recommendation to accept the paper and have raised my confidence accordingly.

The paper theoretically investigates the conditions under which Transformers, pre-trained with offline RL datasets, can be trained to perform In-Context RL (ICRL). To this end, the paper distinguishes between an offline algorithm, used to collect trajectories, and an expert algorithm, which is evaluated on the offline trajectories and the goal of the transformer is to learn to imitate the expert algorithm. The main result of the paper is a generalization bound that depends on model capacity (somewhat expected) and on the distribution ratio between the offline algorithm and the expert algorithm to imitate; implying that ICRL is possible if the model capacity is large enough and the online algorithm performs “close enough” to the expert algorithm. The paper then theoretically analyzes three RL algorithms in detail (LinUCB, ThompsonSampling, and UCB-VI), concluding that Transformers are able to learn these and then perform them in-context. The theoretical analysis is complemented with simulations that support the theory.

**Strengths:**

The paper is quite technical, but well written; leaning towards formalism and clean definitions over extensive prose (which I consider a plus in this case, if the paper length were not limited by 9 pages I would recommend adding additional explanation / interpretation of the formal statements and results). The analysis is timely and original, looks very plausible to me (though I did not check the proofs in detail), and sheds some much needed light on the properties of Transformers that are often portrayed as borderline mystical. I think the work lays some much needed groundwork and provides a theoretical framework for analysis of further RL algorithms; I thus have no concerns regarding the paper's significance to the ICLR community.

**Pros:**
 * Timely problem, approached from a solid theoretical angle. The result is a sophisticated theoretical framework for analyzing RL algorithms and their learnability by Transformers via offline supervised pre-training.
 * Generalization error bound regarding the imitation performance of the Transformer w.r.t. the expert algorithm. Additionally, when assuming realizability, the supervised pre-trained Transformer (log-loss minimization on offline trajectories) is proven to be able to imitate the expert algorithm, given that the offline policy used to collect data “is close” to the expert algorithm (quantified in terms of a distribution ratio).
 * Analysis of 3 concrete RL algorithms, showing theoretically that Transformers can learn to imitate them in-context (with some extra conditions).
 * Simple simulations to confirm the theoretical results for the 3 algorithms.

**Weaknesses:**

The paper is interesting and results look good - my main criticism is that the paper stops somewhat short of situating the results within a wider agenda and clearly stating open issues and limitations of the current results. Overall I currently think that the paper is ready for publication and interesting to a significant part of the ICLR audience. I therefore suggest acceptance at the moment, and will update my verdict based on the other reviews and the authors’ response (it is possible that I missed or overlooked some critical details in the theory and required assumptions for it to hold; therefore I have lowered my confidence accordingly).

**Cons:**
 * The paper currently lacks a strong discussion of limitations. The theory requires some assumptions, and while having the generalization bound is very good it remains somewhat unclear how hard/easy it is to make the bound tight in practice (a lot comes down of course to the mismatch between offline and expert algorithm; how easy is it in practice to keep this mismatch low? What can even be easily said about the distribution ratio in practice?). How do the theoretical statements in the paper lead to non-vacuous practical take-aways (after all requiring low distribution ratio between offline and expert algorithm, and sufficient model capacity are not the most surprising requirements)? Or do additional problems need to be solved to make such statements?
 * Another issue that is missing from the discussion (and perhaps even the analysis) is partial observability. If the expert algorithm depends on unobservables, I believe it would be possible to create settings that lead to a failure to imitate the expert; essentially leading to “Self-Delusions” - see [1]. Taking a wild guess here, but this would probably imply additional assumptions around identifiability? This point becomes even more important when considering humans as “expert policies”.
 * This is perhaps more a question than a weakness with actionable improvement (so consider this point addressed if it can easily be clarified/answered). I am wondering about the relationship between the transformer learning an **amortized algorithm** (i.e. the trained net behaves as if it were running the algorithm under the hood, but only on-distribution because it is not actually running the algorithm but has learned a function that produces the same outputs) **vs. actually implementing something close to the expert algorithm**. It seems to me that the paper suggests the latter (which is why it is important that Transformers can implement accelerated gradient descent and matrix square roots algorithmically). Naively I would have guessed that the transformer learns an amortized solution rather than the actual algorithm. On-distribution the two are indistinguishable, but typically the amortized solution generalizes less well. There are some experiments that could tease apart the two possibilities (but I’d also be interested in a general comment by the authors; most importantly whether anything in the paper would break with an amortized solution). Sketches for experiments (I do not necessarily expect to see them performed, but they might help clarify the discussion):
    * Let the offline algorithm not fully cover the state space. At test time input these parts of the state-space (e.g. a coordinate input in a gridworld that is much larger than anything the Transformer has ever seen; but same dynamics, same reward function, etc). An amortized solution should struggle in this case (and potentially break down catastrophically), whereas learning the correct algorithm should not.
    * Change the reward function at test-time (e.g. picking up bananas in a gridworld has positive reward and apples have negative reward during pre training, but at test-time the rewards are reversed). The amortized solution should struggle in this case.

[1] Shaking the foundations: delusions in sequence models for interaction and control, Ortega et al. 2021.

**Questions:**

1) (How) Could ICRL be used to surpass the performance of the expert (e.g. going from human chess as the expert to superhuman chess)? Most RL algorithms have fairly simple parameters to increase performance (train longer, lower discount, deeper backups, more unrolls, …) - how could this look like for ICRL. Or, alternatively, is it possible make a theoretical statement showing that ICRL performance is capped at exactly the level of the expert (even if we used ICRL to generate a new dataset of supervised trajectories where it slightly surpasses expert performance)?

2) After Eq (2): “all the algorithm outputs [...] along the trajectory can be computed in a single forward propagation.” Does that also imply having masked (causal) attention?

---

> ### Author Response · Authors · 2023-11-21
> **Response to Reviewer fr59**
>
> We are grateful to the reviewer for their thorough review and valuable feedback on our paper. In the following, we hope to address each point raised in the review in the order presented.
>
> > The paper currently lacks a strong discussion of limitations… mismatch between offline and expert algorithm…  non-vacuous practical take-aways?
>
> Thanks for the suggestion. We have added a discussion of the limitations as well as practical implications in our revision, which is currently in Appendix A.1 (due to the space limit) with a pointer in the Conclusion section in the main text.
> As one example of a non-vacuous practical take-away, our results highlight that the training labels (expert actions) in the offline dataset matter a lot. When it is itself a proper algorithm that only depends on the past observations, we can learn ${\sf Alg}_E$ (cf. Theorem 9). By contrast, when ${\sf Alg}_E$ is the ground truth optimal action $a^\star$ (which involves knowledge of the underlying MDP), we can learn the posterior average of this algorithm given past observations, which corresponds to the Thompson sampling algorithm as in Decision-Pretrained Transformers (cf. Theorem 11). A discussion about this can also be found in “Three special cases of expert algorithms” (Page 3) where the “reduced algorithm” $\overline{\sf Alg}_E$ denotes this posterior average.
>
> > … partial observability. If the expert algorithm depends on unobservables, I believe it would be possible to create settings that lead to a failure to imitate the expert; essentially leading to “Self-Delusions” - see [1]. Taking a wild guess here, but this would probably imply additional assumptions around identifiability?
>
> We appreciate the reviewer raising the question of partial observability. It would be helpful if the reviewer could clarify what they mean by "partial observability" in this context.
>
> We agree that when the expert algorithm depends on variables that are unobserved by the learner, the transformer will fail to perfectly imitate the expert. Our Theorem 6 formalizes this notion: we show that the transformer will imitate the expectation of the expert algorithm conditional on the observables, rather than the exact expert algorithm itself. In this setting, the full environment (the decision process instance) is unobservable to the learner.
>
> However, the "self-delusions" discussed in [1] seem to refer to a different type of "partial observability" than what we are considering above. It would be helpful if the reviewer could clarify what they mean by partial observability in this context. Specifically, is the reviewer referring to the formalism of “partially observable Markov decision processes”? We want to ensure we fully address the reviewer's concern, so additional details would allow us to refine our discussion accordingly.
>
> [1] Shaking the foundations: delusions in sequence models for interaction and control, Ortega et al. 2021.
>
> > … **amortized algorithm** vs **expert algorithm** …
>
> As posited by the reviewer, our *statistical theory* only ensures that the pretrained transformer learns an “amortized algorithm” that is only guaranteed to match the expert algorithm in-distribution (same training/test distribution for the bandit/MDP instances), even though our *algorithmic construction* gives the existence of a transformer that actually approximates the expert algorithm in the whole input space.
>
> In some of our early experiments (not included in the paper), we have indeed found that the learned transformers do not generalize that well to out-of-distribution instances such as shift of reward distribution. Similar phenomena have been observed in other types of in-context learning problems, e.g. Garg et al. (2022), among many others. We have added a brief discussion about this in Appendix A.1 in our revision.
>
> > (How) Could ICRL be used to surpass the performance of the expert (e.g. going from human chess as the expert to superhuman chess)? … Or, alternatively, is it possible make a theoretical statement showing that ICRL performance is capped at exactly the level of the expert (even if we used ICRL to generate a new dataset of supervised trajectories where it slightly surpasses expert performance)?
>
> In our setting, we can indeed only expect the learned ICRL algorithm to at most match the expert (i.e., $\overline{\sf Alg}_E$) but not surpass it. However, as the reviewer pointed out, that does not rule out surpassing the expert, e.g. by online RL style training where the transformer can interact with the environment in an online fashion. We would like to leave that as future work.
>
> > After Eq (2): “all the algorithm outputs [...] along the trajectory can be computed in a single forward propagation.” Does that also imply having masked (causal) attention?
>
> Yes, we use the masked (causal) attention in this work (see Definition 1).

---

> > ### Author Response · Authors · 2023-11-22
> > **Follow-up**
> >
> > Thank you again for your review! As the author-reviewer discussion period ends today, please let us know if our response has adequately addressed your questions and concerns, and we'd be happy to answer if you have additional questions.

---

> > ### Comment · Reviewer_fr59 · 2023-11-23
> > **Thanks you for the clarifications and additional details**
> >
> > Thank you for the detailed response. I consider the issues raised in my 'Cons' sufficiently addressed. Please see my response to your general response for more details on my final verdict (as well as my updated review).
> >
> > Re partial observability: I really like the authors' response to my ambiguous question - primarily, I think it is a good idea to have one or two sentences along these lines in the limitations section or general discussion. The main point is that imitation of any expert algorithm has fundamental limitations when the imitator lacks access to certain "knowledge". This can play out in multiple ways (as pointed out by the authors' response) - but no amount of scaling (more data, larger transformers, etc) can solve the problem. This is a general limitation of imitation, not a criticism of the paper. What I had in mind in particular is the case where the expert algorithm observes parts of the state that the imitator does not observe, or, equivalently, the expert has an unobservable internal state (this might sound a bit exotic, but I would argue that this certainly applies in the case of RLHF for instance). Formally, this falls under POMDPs, where the belief-state of the expert is not observed by the imitator, leading to a difficult and potentially ill-defined inference problem on the imitator's side. Again, this is not a particular shortcoming of the paper, but an important problem with imitation in general that the paper is not immune to - and pointing it out briefly to readers might be helpful as the problem is often overlooked in in-context decision-making and RL.
> >
> > Re amortization: thanks for clarifying. I really like the first sentence of the response above (differentiating between statistical guarantee and algorithmic construction). The algorithmic construction is very nice and definitely adds value - the only (small) caveat observed many times before (e.g. w.r.t. Turing completeness of neural networks) is that showing the existence of a set of parameters to implement a particular algorithm with a neural network type can sometimes be a bit vacuous in practice, since *finding* that set of parameters via SGD can reliably fail empirically; thus effectively limiting the representational capacity for due to issues with optimization/search.
> >
> > I appreciate the authors detailed comments, and want to point out that, also given my initial rating of the paper, I do not consider the issues raised by me as substantial for acceptance - I raised them out of interest, and do think that the answers (slightly) add to improving an already good paper.

---

### Official Review · Reviewer_HUY7 · 2023-10-31

**Soundness:** 3 good
**Presentation:** 3 good
**Contribution:** 2 fair
**Rating:** 6
**Confidence:** 1

**Summary:**

This paper provides a theoretical framework that analyzes supervised pretraining for ICRL. It provides two proofs: 1. the supervised-pretrained transformer will imitate the conditional expectation of the expert algorithm given the observed trajectory; 2. transformers with ReLU attention can efficiently approximate near-optimal online reinforcement learning algorithms.

**Strengths:**

1. Their original theory proved the feasibility of using a supervised pretraining transformer on in-context reinforcement learning and provides a quantitative analysis framework.
2. The theoretical analysis demonstrates the important role of supervised pretraining Transformers in in-context RL.

**Weaknesses:**

1. Since my research does not focus on the theory of RL. Therefore, I give some general comments.
2. I am not sure the conclusion is a little bit narrow. The conclusion is limited in a supervised paradigm, using pretraining, and in an ICRL setting. It seems that these conditions restrict the conclusions to be general.

**Questions:**

1.Would different pre-trained method lead to different results?

2.Can learning-from-scratch DT figure out similar conclusion? What’s the difference?

3.Will your conclusions help to inspire the empirical study? Such as figuring out better DT-based offline RL algorithms when using D4RL datasets?

---

> ### Author Response · Authors · 2023-11-21
> **Response to Reviewer HUY7**
>
> We are grateful to the reviewer for their thorough review and valuable feedback on our paper. In the following, we hope to address each point raised in the review in the order presented.
>
> > I am not sure the conclusion is a little bit narrow. The conclusion is limited in a supervised paradigm, using pretraining, and in an ICRL setting. It seems that these conditions restrict the conclusions to be general.
>
> Regarding ICRL: We believe the ICRL paradigm (especially with transformers) has received lots of recent attention, and many works keep improving it including Algorithm Distillation (AD; Laskin et al. 2022), Decision-Pretrained Transformers (DPT; Lee et al. 2023), and more recent ones such as AMAGO [1]. In general, we believe ICRL is a promising way of using transformers to do reinforcement learning, and offers an alternative to other paradigms such as Decision Transformers.
> Regarding supervised pretraining: we believe this is a natural way for transformers to learn to do ICRL, and is actually an advantage (easier and more stable to train) compared with alternatives such as reward maximization using RL training.
>
> [1] AMAGO: Scalable In-Context Reinforcement Learning for Adaptive Agents, Grigsby et al., 2023.
>
> > Would different pre-trained method lead to different results?
>
> Right, other pretraining methods may lead to different results. While we use a supervised MLE loss, there exist other training methods such as cumulative reward maximization, e.g. in Duan et al., (2016). However, that method does not give similar guarantees in our setting. We have added a discussion about this in Appendix A.1 in our revision.
>
> > Can learning-from-scratch DT figure out similar conclusion? What’s the difference?
>
> By “DT” does the reviewer mean Decision Transformers, and “learning-from-scratch DT” as some adaptation of DT to the ICRL setting? (Since the original paradigm of DT only learns a goal-conditioned policy for a fixed environment and cannot do ICRL in new environments). In that case, the “Learning-from-scratch DT” in our opinion is very different from the original DT and not very different from our setup other than perhaps goal-conditioning.
>
> > Will your conclusions help to inspire the empirical study? Such as figuring out better DT-based offline RL algorithms when using D4RL datasets?
>
> We believe our conclusions may provide several intuitions for empirical study on ICRL:
> * The training labels (expert actions) in the offline dataset matter a lot. For example, AD (using actions generated from some expert algorithm) learns the nominal algorithm, DPT (using the optimal actions $a_i^*$) learns Thompson sampling algorithm instead.
> * The distribution ratio between the offline algorithm and the expert algorithm may affect the generalization properties of the learned algorithm. As shown in both our theories (Theorem 6) and simulations (Figure 4), it is important to have a small distribution ratio between the offline algorithm $A_0$ and the expert algorithm $A_E$ in order to achieve good online performance, otherwise the performance may substantially degrade. This suggests that it is good to always incorporate trajectories generated purely from the expert algorithm (“on-policy ICRL”) into the offline dataset, when possible.

---

> > ### Author Response · Authors · 2023-11-22
> > **Follow-up**
> >
> > Thank you again for your review! As the author-reviewer discussion period ends today, please let us know if our response has adequately addressed your questions and concerns, and we'd be happy to answer if you have additional questions.

---

### Official Review · Reviewer_ycZf · 2023-11-03

**Soundness:** 3 good
**Presentation:** 3 good
**Contribution:** 2 fair
**Rating:** 5
**Confidence:** 3

**Summary:**

This paper addresses the theory of supervised pretraining for in-context reinforcement learning (ICRL) via transformers.
- It first proves that supervised pre-training will imitate the expert algorithm, with a generalization error that scales with the model capacity and a distribution ratio between the expert algorithm and the context algorithm that is used to generate offline trajectories.
- It then transformers can approximate RL algorithms including LinUCB (w ridge regression), Thompson sampling (w ridge regression), and UCB-VI.

**Update after rebuttal and reviewer-AC discussion**

After reading the authors' rebuttal and discussing it with the other reviewers/AC, I am partly convinced about the connection to Transformer, though a stronger connection is more desirable. I still had reservations about how useful the bound is, but this is okay as the first step towards the theory and inspiring the community to continue working on it. I am maintaining my score, but I think this is generally a decent work and it would be okay to see it getting accepted.

**Strengths:**

1. The theory of supervised pretraining for in-context reinforcement learning via transformers is an interesting and important problem. This paper is among the first to explore that area.

2. The paper shows a comprehensive understanding of the existing literature via the related work.

3. The paper shows technical depth in the proofs of Theorems.

4. The paper is technically sound and well-written.

**Weaknesses:**

1. My main concern is that the two main sets of results are not as dependent/relevant to Transformers.
- For Theorem 6 (supervised pretraining can imitate the expert algorithm) -- this seems to be a general result that is not specific to Transformers. In fact, any supervised pretraining models that satisfy the approximate realizability assumption should suffice Theorem 6. Is this true?
- For the theorems in Section 4, the paper mostly constructs a specific Transformer structure (e.g., 3-head attention + MLP with a specified QVK) and says that this certain Transformer can mimic the gradient update in the no-regret RL/bandit algorithms. My understanding is that the Transformer is an overkill, and this can generally apply to NNs that have similar structures. Is this true?


2. The idea of proofs of Theorems in Section 4 seems essentially similar to that of Bai et al. 2023, though with nuances.


3. If I understand correctly, the distribution ratio $\mathcal{R}$ in Equation (6) can be arbitrarily large. This significantly limits the usefulness of the bound. In fact, in many cases (except the Algorithm distillation case), this ratio should be sufficiently large as Alg_E and Alg_0 are usually different, and the difference is accumulated over the entire trajectory.


4. Minor issues:
- $M$ is used as both an MDP environment and the number of heads
- Not sure if I missed this -- $n$ Equation (5) and onwards has not been formally introduced. Is this the number of trajectories/samples?

**Questions:**

Please see my questions as in the above.

---

> ### Author Response · Authors · 2023-11-21
> **Response to Reviewer ycZf**
>
> We would like to thank the reviewer for taking the time to review our paper and provide helpful feedback! We truly appreciate your comments and suggestions, and believe they can make our work better.
> In the following, we hope to address each of the points made in the review, in the order as they came out.
>
> > … Two main sets of results are not as dependent/relevant to Transformers… any supervised pretraining models that satisfy the approximate realizability assumption should suffice Theorem 6. Is this true?
>
> Theorem 6 indeed does not depend on transformers, and it holds for any model class that satisfies the approximate realizability assumption. We believe this is not a concern but rather a benefit, and this is expected as the result only involves statistical analysis that does not rely on the structure of the model class.
>
> > For the theorems in Section 4, the paper mostly constructs a specific Transformer structure (e.g., 3-head attention + MLP with a specified QVK) and says that this certain Transformer can mimic the gradient update in the no-regret RL/bandit algorithms. My understanding is that the Transformer is an overkill, and this can generally apply to NNs that have similar structures. Is this true?
>
> We agree with the overall statement that the constructions in Section 4 can be done on other NN architectures that have the similar structures. However, we emphasize that in order for the construction to be *efficient* (with a small number of layers and parameters), our proof heavily uses the specific structure of attention, specifically its ability for token $t$ to efficiently retrieve information from token $1$ to $t-1$, as well as the inner product (between query and key). It is unclear to us whether other sequential architectures such as RNNs/LSTMs can do these as efficiently.
>
> > The idea of proofs of Theorems in Section 4 seems essentially similar to that of Bai et al. 2023, though with nuances.
>
> For our proof ideas for Section 4, we believe the only core similarity with Bai et al. (2023) is that we also use some form of gradient descent as an iterative algorithm to approximate more sophisticated procedures such as matrix inverse. Other than that, our constructions involve several new ingredients, such as accelerated gradient descent (Theorem 8), approximate matrix square root (Theorem 10), and dynamic programming (Theorem 12), as well as lower-level operations such as token copying. In addition to transformer constructions, Section 4 also involves RL arguments for propagating the approximation error from the single-step log-probabilities to the regret, and for bounding the regret of the soft UCBVI algorithm (Theorem 13).
>
> > If I understand correctly, the distribution ratio R in Equation (6) can be arbitrarily large. This significantly limits the usefulness of the bound. In fact, in many cases (except the Algorithm distillation case), this ratio should be sufficiently large as Alg_E and Alg_0 are usually different, and the difference is accumulated over the entire trajectory.
>
> We agree that the distribution ratio R can be large when Alg_E and Alg_0 are different. We believe how to get rid of the dependence on R, e.g. under additional structural conditions for the RL problem or the algorithms is an important question, which we would like to leave as future work.
>
> > Minor issues: $M$ is used as both an MDP environment and the number of heads… $n$ in Equation (5) and onwards has not been formally introduced. Is this the number of trajectories/samples?
>
> Thanks for noticing these. We have fixed the $M$ in our updated version. $n$ is the number of pretraining trajectories ($nT$ is the number of samples), which we introduced in the first paragraph of Section 3.

---

> ### Author Response · Authors · 2023-11-22
> **Follow-up**
>
> Thank you again for your review! As the author-reviewer discussion period ends today, please let us know if our response has adequately addressed your questions and concerns, and we'd be happy to answer if you have additional questions.

---

### Author Response · Authors · 2023-11-21
**Revision uploaded**

We thank all reviewers again for their valuable feedback on our paper. We have incorporated the reviewers’ suggestions and uploaded a revised version of our paper. In particular, we have added limitations and additional discussions section in Appendix A.1. For clarity, all changes (other than typo fixes) are marked in blue.



Best,

Authors

---

> ### Comment · Reviewer_fr59 · 2023-11-23
> **Thank you for the revision and responses**
>
> Reading all other reviews and the authors' response I personally believe that all issues raised, including all issues raised by ycZf, have been well addressed by the authors. I have therefore raised my confidence accordingly.
>
> I agree with the authors that the independence of Theorem 6 on the transformer architecture is an advantage, and I disagree with ycZf that this makes the title a mismatch or implies that the results are only weakly related to transformers - the overwhelming fraction of modern work on in-context RL is done via transformers, which makes it fair, in my opinion, to focus the paper on transformers, even though some of the main results hold for other neural network architectures. Similarly, the results in Section 4 benefit from a focus on architectures that have attention (aka transformers) - while weak (and as the authors state, inefficient) versions of the results could be obtained for other neural network architectures, transformers are *the* important special case that give Section 4 (practical) relevance.
>
> Issue 3 raised by ycZf, regarding the practicality of the bound in Eq. (6), is somewhat warranted, I think: of course it would be nice to have a tighter bound - but given the nature of in-context RL, I would be surprised to see any theory that could come up with a much tighter bound. I would argue that Eq. (6) is a proper formalization of a very naturally and intuitively expected condition. I therefore consider Eq. (6) highly valuable as a formal expression of a fundamental condition for in-context RL to work. Adding additional structural assumptions (as pointed out by the reviewers) is interesting and of high practical relevance, but will not invalidate the more general Eq. (6). In addition, finding other practical ways to keep the bound low are important (but again: this seems a very natural thing to do in in-context RL; analogously in offline-RL there is only so much that can be done about distributional shift; under arbitrary shifts performance bounds are expected to become arbitrarily large). I would be happy to discuss this further in the reviewer discussion, and could see how one might make a point that Eq. (6) is perhaps a contribution of lower significance w.r.t. immediately coming up with an improved ICRL method - I personally would argue that that is not the aim of the paper.

---

### Meta-Review · Area_Chair_b8Cj · 2023-12-06

**Metareview:**

This paper provides a valuable and comprehensive analysis of in-context RL. While previous works mainly focused on empirical analyses, this paper provides a good supplement through a theoretical analysis. The analysis is competent and focuses on three of the most important in-context RL algorithms used in practice. The paper can be improved by adding more discussions. For instance:

* The main general result is Theorem 6. It seems to be proved using standard tools (covering number in Definition 4) applied to more complex spaces than before.

* Theorem 6 is applied to transformers in Paragraph 2 of Section 3.2. However, the paragraph does not make clear at all how the structure of the transformer would be reflected in Theorem 6.

This comment and several related ones were discussed among the reviewers. The general agreement is that these are just writing issues that can be easily fixed. Thus our recommendation is to accept this paper.

**Justification For Why Not Higher Score:**

This paper is theoretical and would be better presented as a poster.

**Justification For Why Not Lower Score:**

Theoretical investigation of in-context RL is timely.

---

### Decision · Program_Chairs · 2024-01-16

Accept (poster)